# PAPER2CODE: AUTOMATING CODE GENERATION FROM SCIENTIFIC PAPERS IN MACHINE LEARNING

**Minju Seo**[1,3], **Jinheon Baek**[†1], **Seongyun Lee**[1], **Sung Ju Hwang**[†1,2]
[1]KAIST, [2]DeepAuto.ai, [3]LG AI Research
{minjuseo, jinheon.baek, seongyun, sungju.hwang}@kaist.ac.kr

## ABSTRACT

Despite the rapid growth of machine learning research, corresponding code implementations are often unavailable, making it slow and labor-intensive for researchers to reproduce results and build upon prior work. In the meantime, recent Large Language Models (LLMs) excel at understanding scientific documents and generating high-quality code. Inspired by this, we introduce PaperCoder, a multi-agent LLM framework that transforms machine learning papers into operational code repositories. PaperCoder operates in three stages: planning, where it constructs a high-level roadmap, designs the system architecture with diagrams, identifies file dependencies, and generates configuration files; analysis, which focuses on interpreting implementation-specific details; and generation, where modular, dependency-aware code is produced. Moreover, each phase is instantiated through a set of specialized agents designed to collaborate effectively across the pipeline. We then evaluate PaperCoder on generating code implementations from machine learning papers based on both model-based and human evaluations, particularly from the authors of those papers, with author-released repositories as ground truth if available. Our results demonstrate the effectiveness of PaperCoder in creating high-quality, faithful implementations. Furthermore, it consistently shows strengths in the recently released PaperBench benchmark, surpassing strong baselines by substantial margins. Code is available at: `https://github.com/going-doer/Paper2Code`.

## 1 INTRODUCTION

Reproducibility lies at the heart of scientific progress, which enables researchers to validate findings, build upon prior work, and ultimately push the boundaries of knowledge (Collaboration, 2015; Baker, 2016; Pineau et al., 2021). However, reproducing scientific results remains an enduring challenge. This is often due to incomplete documentation, missing experimental details, lack of access to data or proprietary tools, and, especially in machine learning research, the absence of corresponding code: for example, only average 19.5% of the papers accepted to top-tier machine learning conferences in 2024 provide their code implementations shown in Figure 1. As a result, researchers frequently invest substantial effort in reverse-engineering methods and experimental results from papers, a process that is both time-consuming and labor-intensive, subsequently slowing down the overall pace of science.

Meanwhile, recent Large Language Models (LLMs) have shown outstanding capabilities in understanding and generating both natural language and programming code (Dubey et al., 2024; OpenAI, 2024; Reid et al., 2024), with performances increasingly approaching or even surpassing that of domain experts in some scenarios. In addition, this progress has sparked growing interest in leveraging LLMs to accelerate scientific workflows, particularly in the early stages of ideation for new and valid research hypotheses (Lu et al., 2024; Li et al., 2024; Yang et al., 2024; Si et al., 2024; Yamada et al., 2025; Schmidgall et al., 2025; Baek et al., 2025). Furthermore, some of these studies, as well as others focusing on later stages of automating experimental validations and improvements (Huang et al., 2024; Zhang et al., 2024; Trirat et al., 2024; Chan et al., 2025), demonstrate the potential of LLMs to generate code and even carry out experiments end-to-end; however, they typically assume and heavily rely on access to pre-existing implementations, partial code snippets, or well-defined

---

[†]Equal Advising.

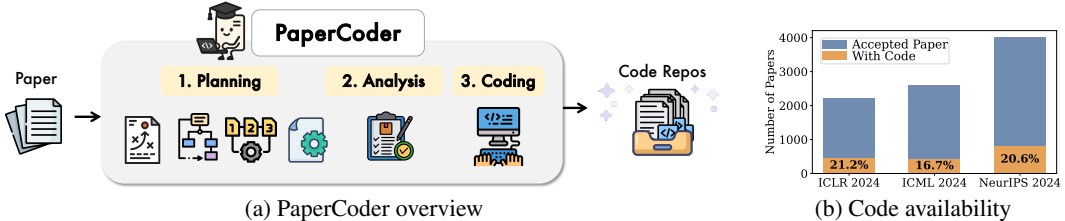

(a) PaperCoder overview        (b) Code availability

Figure 1: (a) PaperCoder, which aims to transform given scientific papers into code repositories, consisting of planning, analysis, and coding steps. (b) Code availability, where blue bars indicate the total number of accepted papers and orange regions show those with officially released code (See Appendix B.1 for calculation details).

APIs. As such, it remains questionable whether generating faithful implementations solely from papers (without access to prior code, APIs, or additional supplementary materials) can be achievable.

To answer this question, we introduce PaperCoder, a multi-agent LLM-powered framework, designed to automatically generate faithful code repositories in machine learning directly from and contextualized with research papers, which differs from prior work that requires partial implementations from human inputs. Specifically, PaperCoder aims to emulate the typical life cycle of human developers and researchers in writing the repository-level code, by decomposing the task into three structured stages: planning, analysis, and generation. First, during the planning stage, the proposed framework constructs a high-level roadmap to identify core components to implement, draws the overall system architecture with class and sequence diagrams to model structural relationships between modules, identifies file dependencies with their execution orders to guide correct build and execution flows, and generates configuration files to enable flexible customization of experimental workflows by human researchers. This is followed by the analysis stage, performing a fine-grained interpretation of each file and function with respect to their intended functionality, such as required inputs and outputs, interactions with other modules, and any algorithmic or architectural constraints derived from the source paper. Finally, in the generation stage, the framework synthesizes the entire code base based on the execution order determined earlier, along with the artifacts produced in the previous stages.

To validate the effectiveness of PaperCoder, we conduct extensive evaluations on a subset of recent machine learning papers from ICLR, ICML, and NeurIPS referred to as our proposed Paper2Code benchmark (in short, Paper2CodeBench). Also, we incorporate the recent benchmark (Starace et al., 2025) in our evaluation suite, enabling fine-grained evaluations of code implementations. Then, on a battery of tests conducted not only with automated model-based evaluations (covering both reference-free and reference-based settings, conditional on the availability of author-released ground-truth repositories) but also with expert human evaluations (based on authors of original papers), PaperCoder demonstrates substantial improvements over baselines, generating more valid and faithful code repositories that could meaningfully support human researchers in reproducing prior work. Specifically, 88% of the generated repositories by PaperCoder are rated as the best over baselines, and 92% of human judges report that the generated repositories are indeed helpful. Also, analyses show that each component of PaperCoder (consisting of planning, analysis, and generation) contributes to the performance gains, but also that the generated codebases can be executed, sometimes with only minor modifications (averaging 0.81% of total code lines) in cases where execution errors occur.

## 2 RELATED WORK

**Large Language Models for Code**     LLMs have shown impressive capabilities in text understanding and generation (OpenAI, 2024; Dubey et al., 2024; Reid et al., 2024) and widely utilized for specialized domains (beyond general tasks), such as mathematics, science, and coding (Prabhakar et al., 2025; Wang et al., 2024b; Trinh et al., 2024). Particularly, code-specialized LLMs (Hui et al., 2024; DeepSeek-AI et al., 2024; 2025) have received significant attention thanks to remarkable performance on various software engineering tasks (Xia et al., 2024), including software design and development (Qian et al., 2024; Hong et al., 2024), requirements elicitation (Mu et al., 2023), and formal specification generation (Luo et al., 2024). Our work aligns closely with this line of research, exploring and expanding upon the capabilities and applications of (code-specialized) LLMs.

**Repository-Level Coding**  Early work on code generation typically focuses on single-file tasks, whose objective is to generate short code snippets to solve isolated tasks, such as (algorithmic-level) programming competition problems (Chen et al., 2021; Austin et al., 2021; Hendrycks et al., 2021; Li et al., 2022). However, as LLMs have advanced in comprehending and generating code with the long-context reasoning ability, recent studies have increasingly shifted their attention toward more challenging repository-level coding tasks, which involve generating multi-file repositories that jointly account for architectural design, modular structure, and inter-file dependencies (Liu et al., 2024; Jain et al., 2024; Tang et al., 2024). In particular, several recent efforts explore this emerging paradigm (Zhang et al., 2023; Ouyang et al., 2025), adopting multi-agent or role-based frameworks to emulate realistic development workflows. For instance, ChatDev instantiates LLMs into role-playing agents that collaborate through structured dialogues (Qian et al., 2024), while MetaGPT implements a waterfall-style development pipeline with specialized agents (Hong et al., 2024). Beyond prior work, we explore the underexplored task of transforming full, complex papers into repository-level code.

**LLM-Powered Scientific Research**  LLMs have been adopted to support the scientific process from ideation to experimental validation (Popper, 1959; Qi et al., 2023; Li et al., 2024; Yang et al., 2024; D'Arcy et al., 2024; Liang et al., 2024; Baek et al., 2025; Weng et al., 2025); thereby, helping researchers overcome existing challenges and ultimately accelerate scientific discovery (Lehr et al., 2024; Lu et al., 2024; Yamada et al., 2025). Specifically, in fields such as computer science (where code-based experimentation is central), LLMs have been used to design, refine, and extend code implementations. However, many recent efforts in this space assume access to and build on top of the original codebase (Huang et al., 2024; Trirat et al., 2024; Xiang et al., 2025; Chan et al., 2025), which significantly limits their applicability in real-world scenarios since such implementations are oftentimes unavailable (See Figure 1). To address this, concurrent to our work, Starace et al. (2025) introduces a benchmark dataset called PaperBench, evaluating the capability of existing agentic AI systems in reproducing papers with fine-grained metrics. Notably, on top of PaperBench (which emphasizes evaluation), we further complement and extend this line by focusing on methodological aspects of how to transform scientific papers into repository-level code implementations.

## 3 METHOD

In this section, we start with describing the task of repository-level code generation from machine learning papers, and propose PaperCoder, a multi-agent, multi-stage framework designed to tackle it.

### 3.1 REPOSITORY-LEVEL CODE GENERATION FROM MACHINE LEARNING PAPERS

The goal of our repository-level code generation task is to automatically produce a repository that faithfully implements methods and experiments described in machine learning papers (especially for cases where authors do not release their code), to support reproducibility and accelerate scientific progress (Pineau et al., 2021; Magnusson et al., 2023). Formally, we define this task as a function (or a model) $M$ that maps a paper $R$ to a corresponding code repository $C$, as follows: $M(R) = C$. Here, $C$ is composed of multiple files $\{c_1, c_2, ..., c_n\}$, each responsible for implementing different components of the methods and experiments in $R$, but together they should form a cohesive pipeline.

The most straightforward approach to instantiating $M$ is to instruct the LLM to generate the entire code repository, conditioned on the given paper, as follows: $M(R) := \text{LLM}(\mathcal{T}(R))$, where $\mathcal{T}$ is the prompt template that specifies the intended behavior of the LLM for the target task (including task descriptions, detailed instructions, and any other relevant context). Yet, generating a complete, modular, and faithful repository in a single pass is extremely challenging, even for powerful LLMs, due to the inherent complexity of scientific papers and their corresponding implementations, the long-context limitations of current models, and the difficulty in maintaining consistent global structure and cross-file dependencies. Therefore, we propose to decompose the overall task into smaller subtasks, each handled by a specialized agent tailored to a specific aspect of paper-to-code transformation.

### 3.2 PAPERCODER: LLM-POWERED MULTI-AGENT FRAMEWORK FOR PAPER-TO-CODE

We now introduce PaperCoder, a structured, multi-agent framework for generating code repositories directly from machine learning papers (without access to pre-existing artifacts or implementations,

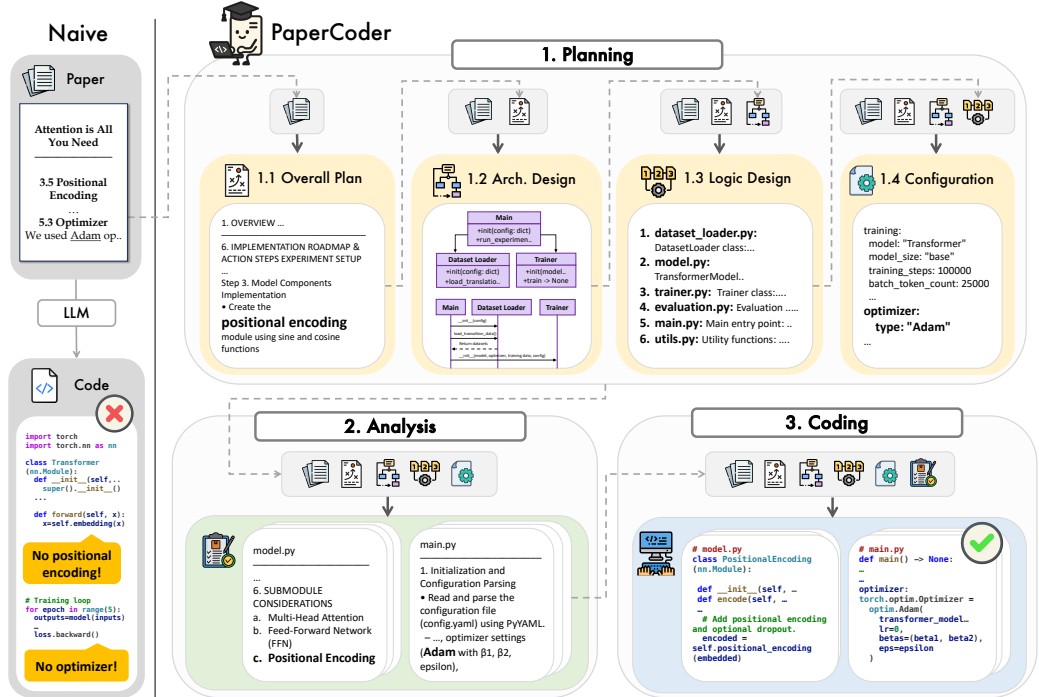

Figure 2: (Left) The naive approach, which directly generates an entire code repository from a paper. (Right) Our PaperCoder framework, which is operationalized by decomposing the task into three stages: (1) Planning, where a high-level implementation plan is constructed from the paper, including overall plan, architectural design, logic design, and configuration file; (2) Analysis, where the plan is translated into detailed file-level specifications; and (3) Coding, where the final codes are generated to implement the methods and experiments of the paper.

such as skeleton code). Specifically, inspired by typical software development workflows, PaperCoder decomposes the task into three coordinated stages: Planning, Analysis, and Coding, each orchestrated by specialized LLM agents. Formally, given a paper $R$, the overall process can be defined as follows:

$$\textbf{Planning: } P = M_{\text{plan}}(R), \quad \textbf{Analysis: } A = M_{\text{analysis}}(R, P), \quad \textbf{Coding: } C = M_{\text{code}}(R, P, A),$$

where $P$, $A$, and $C$ represent the high-level implementation plan, the detailed function-level analysis, and the final code repository, respectively. The overall pipeline of PaperCoder is shown in Figure 2.

### 3.2.1 PLANNING

It is worth noting that, in contrast to implementation specifications designed explicitly for software development, papers are written to communicate ideas and findings to humans. As a result, they often contain high-level motivations, persuasive narratives, and auxiliary details that are crucial for human understanding but noisy, loosely specified, or ambiguous from a software engineering perspective. To mitigate this, we introduce a planning phase that transforms unstructured textual content into implementation-level abstractions. Also, we decompose the planning process into four sequential subcomponents (to simplify the task and reduce cognitive load of LLM-powered agents at each step): 1) overall plan, 2) architecture design, 3) logic design, and 4) configuration generation. Formally, we define this as: $M_{\text{plan}}(R) \rightarrow P = \{o, d, l, g\}$, where $o$ is the overall plan, $d$ is the architecture design, $l$ is the logic design, and $g$ is the configuration file, with each stage using the outputs of the previous ones as contextual input. We then describe how each subcomponent is instantiated below.

**Overall Plan** The first step is to extract a high-level summary of the core components and functionalities described throughout the paper, to identify the specific methods and experiments to be implemented. In other words, this high-level overview includes model components, training objectives, data processing steps, and evaluation protocols (distributed across the entire paper), which can form the foundation for all subsequent steps, formalized as follows: $M_{\text{plan}}^{(1)}(R) := \text{LLM}(\mathcal{T}_{\text{plan}}^{(1)}(R)) \rightarrow o$.

**Architecture Design** Based on the extracted overall plan alongside the input paper, the next step is to define the repository-level architecture, which includes identifying files, organizing them into modules, and defining their relationships, to ensure a coherent and maintainable structure. Specifically, the LLM-powered agent is prompted to generate a file list, which outlines the overall file structure of the repository; a class diagram, which details static representations of files (such as core classes and their attributes); and a sequence diagram, which models the dynamic interactions. Formally, similar to overall plan, this process can be defined as follows: $M_{\text{plan}}^{(2)}(R, o) := \text{LLM}(\mathcal{T}_{\text{plan}}^{(2)}(R, o)) \rightarrow d$.

**Logic Design** While the previous architecture design focuses on what to build, the logic design phase specifies how these components should be instantiated in practice by considering their dependencies in terms of overall execution flow. This step is crucial because individual modules often depend on shared utilities, configurations, or data loaders that are defined in other parts of the repository, and without an explicitly defined execution order, the code generation can result in failure or inconsistency (e.g., generating file B before file A when B imports modules from A). To address this, the logic design stage not only produces an ordered file list that dictates the sequence in which the files should be implemented and executed, but also further elaborates on the logic within each file; thereby, providing more fine-grained specifications. Formally, $M_{\text{plan}}^{(3)}(R, o, d) := \text{LLM}(\mathcal{T}_{\text{plan}}^{(3)}(R, o, d)) \rightarrow l$.

**Configuration Generation** In the last stage of planning, PaperCoder synthesizes a configuration file (`config.yaml`) that includes key hyperparameters, model settings, and other runtime options based on prior outputs alongside the given paper. We note that, in addition to grounding the code generation process with the explicit configuration details, it enables researchers to easily review and adjust experimental configurations without modifying the source code. Formally, $M_{\text{plan}}^{(4)}(R, o, d, l) :=$ $\text{LLM}(\mathcal{T}_{\text{plan}}^{(4)}(R, o, d, l)) \rightarrow g$. We provide prompts used to elicit each planning output in Appendix D.

### 3.2.2 ANALYSIS

Following the planning stage, which defines the overall structure and execution flow of the repository, the analysis phase focuses on interpreting and specifying the implementation-level details for modules within each file. In other words, unlike planning that answers what components to build and how they relate, this phase addresses the question of how each component should be operationalized and concretely implemented at the file level, which includes the definition of functional goals, input-output behaviors, intra- and inter-file dependencies, and algorithmic specifications derived from the original paper. Specifically, given the input paper $R$ and planning outputs $P = \{o, d, l, g\}$, the analysis agent iteratively processes each file $f_i$ (identified during planning) and generates a detailed analysis $a_i$ describing what needs to be implemented in that file. Formally, $\{M_{\text{analysis}}(R, P, f_i)\}_{i=1}^{n=|F|}$ where $M_{\text{analysis}}(R, P, f_i) := \text{LLM}(\mathcal{T}_{\text{analysis}}(R, P, f_i)) \rightarrow a_i$, with $F$ as the set of identified files, e.g., $f_i \in F$.

### 3.2.3 CODING

The final stage is the coding phase, where the complete code repository is produced. In particular, each file is generated based on all the available contextual information accumulated from the previous stages, including the overall plan, architecture design, logic design, configuration file, and file-specific analyses, as well as the original paper. Additionally, to ensure consistency across different files, we generate them sequentially according to the execution order (i.e., the ordered file list determined during the logic design stage). To be formal, for each file $f_i$, the corresponding code $c_i$ is generated as follows: $M_{\text{code}}(R, P, f_i, a_i, \{c_1, ..., c_{i-1}\}) := \text{LLM}(\mathcal{T}_{\text{code}}(R, P, f_i, a_i, \{c_1, ..., c_{i-1}\})) \rightarrow c_i$, resulting in the complete code repository $C = \{c_i\}_{i=1}^{n=|F|}$. We note that this iterative formulation can ensure that $i$-th code is generated with full awareness of its dependencies and the evolving state of the repository.

## 4 EXPERIMENT

We now describe the experimental setup and the experimental results with reproducibility analyses.

## 4.1 EXPERIMENTAL SETUP

**Datasets** To evaluate our PaperCoder, we construct a new benchmark (**Paper2CodeBench**). Specifically, we collect the accepted papers from recent machine learning venues (such as ICLR, ICML, and NeurIPS 2024) with the OpenReview API[1], and filter them based on the availability of code with its total number of tokens less than 70,000, to ensure the full repository remains within reasonable processing limits of modern LLMs for generation and evaluation. Also, to maintain the quality, we perform model-based evaluation (Liu et al., 2023) with GPT-4o on all the collected repositories and select the top 30 from each venue, resulting in a total of 90 papers listed in Tables 20, 21, and 22. Moreover, we additionally consider 21 papers for human evaluation (See Table 23). In addition to Paper2CodeBench, we also use the recently released **PaperBench Code-Dev** (Starace et al., 2025), which consists of 20 papers from ICML 2024 with paper-specific rubrics annotated by humans. In particular, those rubrics are used to judge the correct implementation based on LLM-based evaluation.

**Baselines and Our Model** We target the novel problem of Paper2Code, and there are no baselines designed for it to enable direct comparison. Nevertheless, we consider several related approaches proposed to implement repository-level code (or the entire software) from natural language inputs (such as software requirements), in addition to the ablated variants of our full PaperCoder framework, as follows: **ChatDev** (Qian et al., 2024) is a multi-agent framework for software development, where several role-specific LLM-powered agents collaborate via structured dialogues; **MetaGPT** (Hong et al., 2024) similarity adopts a role-based multi-agent paradigm, but its process is organized by the principle of Standardized Operating Procedures (SOPs); **Abstract** is a variant of our PaperCoder, which uses only the paper abstract for implementation; **Paper**, while using the full paper, performs one-shot code generation; **PaperCoder (Ours)** is our full framework, structured into three stages of planning, analysis, and code generation. Additionally, for the PaperBench Code-Dev, we consider baselines suggested by it: **Basic Agent** is the agentic architecture that can run a predefined set of tools with the ReAct-style approach (Yao et al., 2023), built upon the agent from Inspect AI[2], and **Iterative Agent** that extends Basic Agent, iteratively instructing the model to complete the next subtask.

**Evaluation Setup** Recall that, as shown in Figure 1, the official code implementations of many papers are not available; however, manually annotating their corresponding code implementations to evaluate the quality of automatically generated code repositories is highly labor-intensive and challenging. To address this and ultimately perform the evaluation at scale, we design two evaluation protocols: reference-based (when ground-truth code is available) and reference-free (when it is not), following the recent trends in using LLMs as a judge (Zheng et al., 2023; Fu et al., 2024; Liu et al., 2023). In addition to this, we also perform human evaluations with the authors of the original papers, to ensure reliable judgments and to assess the quality of our model-based evaluations by measuring their correlation with human scores. We discuss each evaluation protocol in detail below.

- *Reference-Based Evaluation*. We use the official author-released repository as the gold standard only if it is available, since it most accurately reflects the implementations intended by the authors, including the components they consider essential to their main ideas. Specifically, we prompt the model (such as o3-mini-high[3]) to judge the quality of the generated repository with respect to the gold repository, alongside the input paper as context (See Appendix D for the detailed prompt). The model then identifies components (to be implemented), categorizes them into three severity levels (high, medium, and low), and critiques how well each component is implemented. After that, it returns the overall score on a 5-point Likert scale. We note that, to ensure the reliability of the model-based evaluation, we sample multiple outputs (e.g., 8) and report the average score.

- *Reference-Free Evaluation*. For cases where the official author-released code is not available, we introduce the reference-free evaluation protocol that leverages only the paper to assess the quality of its generated repository. Similar to the reference-based evaluation, the evaluation model is prompted to identify key components, categorize them by severity, and critique their implementations in the generated code, but they are performed solely based on the information provided in the paper. The rest of the evaluation process, such as sampling and score averaging, follows the same setup.

---

[1]https://docs.openreview.net/reference/api-v2

[2]https://inspect.ai-safety-institute.org.uk/agents.html#sec-basic-agent

[3]Unless otherwise stated, we use o3-mini-high due to strong code understanding and reasoning capability.

Table 1: Results on our Paper2CodeBench, where we report average scores and standard deviations (in parentheses) grouped by conferences. Oracle denotes the evaluation results with the official repository released by the paper authors. Also, on the right side, we report statistics on the number of tokens, files, and functions, averaged over all implementations. Bold indicates the best scores, statistically significant than baselines ($p \leq 0.05$).

| | Reference-Based Evaluation | | | Reference-Free Evaluation | | | Statistics | | |
|---|---|---|---|---|---|---|---|---|---|
| | ICLR | ICML | NeurIPS | ICLR | ICML | NeurIPS | # of Tokens | # of Files | # of Funcs |
| ChatDEV | 2.70 (0.63) | 2.97 (0.58) | 2.96 (0.69) | 4.00 (0.65) | 4.12 (0.53) | 4.01 (0.74) | 6150.54 | 6.99 | 23.82 |
| MetaGPT | 2.48 (0.48) | 2.75 (0.70) | 2.95 (0.87) | 3.52 (0.60) | 3.63 (0.75) | 3.59 (0.92) | 5405.21 | 3.24 | 18.08 |
| Abstract | 2.28 (0.42) | 2.43 (0.49) | 2.35 (0.62) | 3.03 (0.64) | 3.01 (0.60) | 2.99 (0.78) | 3376.99 | 1.28 | 12.62 |
| Paper | 3.08 (0.66) | 3.28 (0.67) | 3.22 (0.80) | 4.15 (0.63) | 4.30 (0.53) | 4.08 (0.84) | 3846.33 | 1.79 | 14.84 |
| PaperCoder | **3.68** (0.52) | **3.72** (0.54) | **3.83** (0.50) | **4.73** (0.32) | **4.73** (0.44) | **4.77** (0.38) | 14343.38 | 6.97 | 35.22 |
| Oracle | N/A | N/A | N/A | 4.84 (0.26) | 4.80 (0.32) | 4.83 (0.38) | 32149.04 | 28.00 | 122.03 |

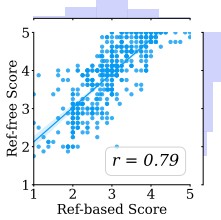

Figure 3: Correlation between model-based evaluations: reference-based and reference-free.

Table 2: Results with human evaluation. For model-based evaluations (both reference-based and reference-free), 5-point Likert evaluation scores are converted to rankings for comparability with human ranking results. Human rankings are also converted to scores of 5 (top repository), 3 (middle repository), and 1 (bottom repository).

| | Score ($\uparrow$) | | | Ranking ($\downarrow$) | | |
|---|---|---|---|---|---|---|
| | Ref-based | Ref-free | Human | Ref-based | Ref-free | Human |
| Abstract | 2.26 (0.37) | 2.94 (0.61) | 2.68 (0.56) | 2.96 (0.20) | 2.96 (0.00) | 2.70 (0.56) |
| Paper | 3.00 (0.54) | 3.91 (0.63) | 2.76 (1.20) | 1.92 (0.41) | 1.88 (0.38) | 2.09 (0.60) |
| PaperCoder (Ours) | **3.66** (0.43) | **4.55** (0.51) | **4.60** (1.00) | **1.08** (0.28) | **1.08** (0.28) | **1.22** (0.52) |
| ChatDEV | 2.68 (0.60) | 3.82 (0.37) | 2.12 (1.17) | 2.58 (0.50) | 2.23 (0.59) | 2.43 (0.59) |
| MetaGPT | 2.61 (0.54) | 3.39 (0.67) | 2.12 (1.17) | 2.38 (0.58) | 2.46 (0.51) | 2.43 (0.59) |
| PaperCoder (Ours) | **3.66** (0.43) | **4.55** (0.51) | **4.76** (0.88) | **1.04** (0.20) | **1.04** (0.20) | **1.13** (0.46) |

- *Human Evaluation*. While model-based evaluation offers a scalable and automated way of assessment, we also conduct human evaluations to validate our PaperCoder based on expert-grounded evaluation. Specifically, to ensure informed and accurate judgment, each participant is assigned a paper for which they are the first author. Also, they are presented with multiple implementations generated by different approaches, and asked to rank them. We offer more details in Appendix A.2.

Lastly, for evaluation on the PaperBench Code-Dev benchmark (Starace et al., 2025), we follow their evaluation setup, measuring the score over the paper-specific rubrics with LLM-based evaluation.

## 4.2 EXPERIMENTAL RESULTS AND ANALYSIS

**Main Results** Table 1 presents main results on Paper2CodeBench, in which PaperCoder consistently outperforms all baselines. We hypothesize that this performance gap stems from its top-down behavior, analyzing full papers thoughtfully before generation, unlike prior approaches that typically follow a bottom-up strategy, which begins with and expands short requiremental descriptions (via role-playing or SOP). In other words, the top-down approach, operationalized through the sequence of planning, analysis, and coding, is effective in handling long-form scientific documents, which are often loosely structured from a software engineering perspective. Also, when compared to the non-comparable Oracle setting (which performs evaluations on the author-released repositories), PaperCoder achieves performance that is on par, without statistically significant differences, demonstrating its effectiveness in faithfully implementing code whose quality is closer to the implementation by authors.

**Correlation between Reference-Based and Reference-Free Evaluation** Recall that the reference-free evaluation protocol is designed for cases where the ground-truth repository is not available, and to investigate whether it works as a reliable proxy for the reference-based evaluation protocol, we measure their rank correlation on all samples from Paper2CodeBench. Then, as shown in Figure 3, there is a strong positive correlation between them, achieving a Pearson correlation coefficient of $r = 0.79$. This result supports that the reference-free evaluation can serve as a reliable proxy for the reference-based evaluation, ultimately functioning as a standalone metric to assess the code quality.

Table 3: PaperBench Code-Dev results. We report the averaged performance over three runs with standard deviations.

| | Replication Score (%) | |
|---|---|---|
| Model | o3-mini-high | claude-3.5-sonnet |
| BasicAgent | $5.1 \pm 0.8$ | $35.4 \pm 0.8$ |
| IterativeAgent | $16.4 \pm 1.4$ | $27.5 \pm 1.6$ |
| PaperCoder | $\mathbf{45.14} \pm 0.3$ | $\mathbf{51.14} \pm 1.4$ |

Table 4: Results based on both model-based and human evaluations with varying backbone LLMs for PaperCoder.

| | | DS-Coder | Qwen-Coder | DS-Distill-Qwen | o3-mini-high |
|---|---|---|---|---|---|
| Score (↑) | Ref-based | 1.47 (0.46) | 1.78 (0.28) | 2.05 (0.25) | **3.66** (0.43) |
| | Ref-free | 1.62 (0.54) | 2.09 (0.22) | 2.31 (0.24) | **4.55** (0.51) |
| | Human | 1.32 (0.58) | 2.71 (1.12) | 3.29 (0.98) | **4.68** (0.80) |
| Ranking (↓) | Ref-based | 3.46 (0.00) | 2.92 (0.88) | 2.25 (0.65) | **1.00** (0.20) |
| | Ref-free | 3.50 (0.00) | 2.88 (0.83) | 2.12 (0.54) | **1.00** (0.25) |
| | Human | 3.74 (0.45) | 2.74 (0.86) | 2.30 (0.70) | **1.22** (0.60) |

Table 6: Ablation results on the subset of Paper2CodeBench with scores and standard deviations.

| | Ref-based | Ref-free |
|---|---|---|
| Paper | 3.28 (0.67) | 4.30 (0.53) |
| + Overall Plan | 3.40 (0.57) | 4.34 (0.58) |
| + Arch. Design | 3.13 (0.68) | 4.07 (0.74) |
| + Logic Design | 3.60 (0.52) | 4.50 (0.57) |
| + Config File | 3.66 (0.45) | 4.45 (0.53) |
| + Analysis (Ours) | **3.72** (0.54) | **4.73** (0.44) |

Table 7: Results of the PaperCoder and PaperCoder with Self-Refine, under the reference-based evaluation protocol.

| | PaperCoder | w/ Self-Refine |
|---|---|---|
| Overall Plan | 4.67 | 4.87 (+0.20) |
| Arch. Design | 3.20 | 3.96 (+0.76) |
| Logic Design | 4.09 | 4.38 (+0.29) |
| Config File | 2.93 | 3.93 (+1.00) |
| Analysis | 4.18 | 4.32 (+0.14) |
| Code | 3.39 | 3.89 (+0.50) |

Figure 4: Model-based evaluation results by paper presentation types.

**Human Evaluation Results** In addition to automatic evaluations, we conduct human evaluations and report the results in Table 2. From this, we confirm that PaperCoder achieves the best ranking, consistent with model-based evaluations, which reaffirms its effectiveness. Also, to ensure whether the model-based evaluations are a reasonable proxy to judge the implementation quality, we measure their correlations with human evaluation scores. As shown in Table 5, we observe strong rank correlations

Table 5: Rank correlation coefficient between human and model-based evaluations (with GPT-4o or o3-mini).

| | GPT-4o | o3-mini-high |
|---|---|---|
| Ref-based | 0.74 | **0.78** |
| Ref-free | 0.71 | **0.73** |

across both reference-based and reference-free settings, which suggests that model-based evaluation can reliably approximate human judgment. Also, based on this result, we use o3-mini-high as the default evaluation model. Lastly, we ensure the quality and reliability of human evaluations by measuring the inter-annotator agreement based on Cohen's kappa coefficient, which exhibits a high score of 0.79, indicating strong consistency.

**Results on PaperBench Code-Dev** In addition to our Paper2CodeBench, we further validate the effectiveness of PaperCoder on another PaperBench Code-Dev dataset, which enables fine-grained evaluations for code implementations. As Table 3 shows, PaperCoder achieves the highest replication scores across two different LLMs of o3-mini-high and Claude 3.5 Sonnet, substantially outperforming baselines designed for PaperBench Code-Dev. These results further demonstrate the generalizability and robustness of PaperCoder across diverse evaluation benchmarks and models.

**Analysis on Different LLMs** Extending the model variations results on PaperBench Code-Dev, we conduct an auxiliary analysis with DS-Coder (DeepSeek-Coder-V2-Lite-Instruct; DeepSeek-AI et al., 2024), Qwen-Coder (Qwen2.5-Coder-7B-Instruct; Hui et al., 2024), DS-Distill-Qwen (DeepSeek-R1-Distill-Qwen-14B; DeepSeek-AI et al., 2025), and o3-mini-high (the high reasoning-effort variant of o3-mini) on Paper2CodeBench. As summarized in Table 4, the proprietary model (o3-mini-high) consistently outperforms all other backbones across all evaluation settings. Among other open-source models, DS-Distill-Qwen performs the best, followed by Qwen-Coder and DS-Coder. These results suggest the importance of selecting a capable backbone to instantiate PaperCoder, particularly one with strong reasoning capabilities. Also, based on this, we primarily use o3-mini-high as the basis.

**Ablation Studies** To see how much each component of PaperCoder contributes to the performance gain, we conduct ablation studies on the subset of Paper2CodeBench (composed of ICML papers). Specifically, we start with the method that uses only the full paper and incrementally add components in the order they are executed (such as overall plan, architecture design, logic design, configuration generation, and analysis), reported in Table 6. From this, we observe that the performance steadily improves as additional components are incorporated. Meanwhile, a performance drop occurs when the architecture design module is added; however, while this might seem surprising at first, it is in fact expected: architecture design alone does not specify the execution or implementation order of files, which leads to confusion during the code generation (see Figure 7). However, this issue is addressed once the subsequent logic design module explicitly defines file dependencies and establishes a clear

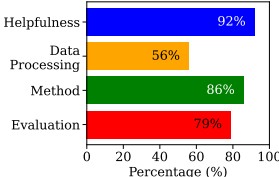

Figure 5: Fine-grained analyses on code by PaperCoder.

Table 8: Replication scores on 10 papers from PaperBench, including execution and result match.

| Model | Score (%) |
|---|---|
| BasicAgent | 2.60 |
| IterativeAgent | 11.22 |
| PaperCoder | **28.46** |

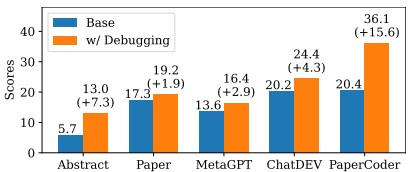

Figure 6: Results on the author-written rubric for papers from Paper2CodeBench (human evaluated), with gains in parentheses.

generation order. Overall, integrating all modules yields the highest performance, confirming the effectiveness of our fully structured, multi-stage pipeline with various modules proposed.

**Experiment with Refinement** We confirm in Table 6 that the planning and analysis stages play a pivotal role in guiding subsequent analysis and coding, and we further test whether refining earlier outputs can improve downstream performance. Specifically, we augment the planning and analysis phases with verification-and-refinement steps (See Figures 23 to 32 for prompts), following Self-Refine (Madaan et al., 2023), and evaluate a total of 30 papers subsampled from Paper2CodeBench (10 from each conference). As shown in Table 7, refinement of planning and analysis improves their own outputs but also leads to measurable gains in the subsequent stages, reducing downstream errors.

**Correlation on Paper Type** To see whether the acceptance category (or presentation format) of papers correlates with the quality of their corresponding implementations by PaperCoder, we analyze it by separating papers into oral/spotlight and poster categories on Paper2CodeBench (which includes 14 oral or spotlight papers and 76 poster papers). As shown in Figure 4, scores are slightly higher for oral/spotlight papers on model-based evaluations with GPT-4o and o3-mini, suggesting that papers with higher recognition might reflect clearer writing, probably leading to faithful code generation. For further analysis on how the completeness of papers impacts the results, please refer to Table 12.

**Fine-Grained Analysis of Generated Repositories** To more thoroughly evaluate the quality and practical utility of the generated code, we conduct a set of fine-grained human analyses according to its usability for reproduction and its component-wise implementation quality. Specifically, we ask annotators whether the top-ranked repository from PaperCoder would make reproducing the original work easier than starting from scratch, and 92% agree, highlighting its practical value. Also, we conduct a component-level analysis to assess which parts of the papers are most effectively translated into code, by asking human annotators to identify key elements for Data Processing, Method, and Evaluation, then measure how many are actually implemented. As shown in Figure 5, the coverage reaches 80% for Method and 79% for Evaluation. Notably, among the errors observed, many of them originate from the Data Processing stage, where papers often under-specify details about data formats, preprocessing steps, or loading procedures. Lastly, to investigate why human annotators prefer PaperCoder over its baselines and ablated variants (with 22 out of 25 selecting the repositories from PaperCoder), we ask them to provide the reasons for their choices, and the majority of which are completeness, clean structure, and faithfulness to the original papers, summarized in Table 16.

### 4.3 ADDITIONAL ANALYSIS ON REPRODUCTION FROM IMPLEMENTED CODE REPOSITORY

While our focus is on generating faithful implementations that can aid research, we further examine whether these implementations can fully reproduce the original experimental results end-to-end.

**Analysis on Executability** It is worth noting that making the repository-level code executable and fully reproducible in one go is extremely challenging (even for humans), as demonstrated by Starace et al. (2025). Also, our goal is to provide a faithful starting point that meaningfully aids reproduction efforts (Figure 5), rather than aiming for perfect reproduction. Nevertheless, to assess how close our generated repositories are to being directly executable, we perform manual execution evaluations on five papers. Specifically, when execution fails, we manually debug and refine the code and adapt the input data as needed to enable successful runs. We then find that, on average, only 0.81% of the code lines require minor modification, such as updating deprecated API or correcting data type mismatches, for successful execution (see examples in Figures 8 to 12 with statistics in Table 15), which highlights that our generated repositories are near-executable with minimal human intervention.

**Analysis on Reproducibility**   An equally important, though not our primary focus, question is whether the generated repositories can reproduce the results intended by the original authors. To examine this, we sample 10 papers from PaperBench and another 10 from the human evaluation set of Paper2CodeBench. Also, we automatically invoke LLM-assisted debugging (only when execution errors occur), where the model was provided with error messages, source code, and relevant training data (if needed) to resolve issues. First, for PaperBench, we use the full rubric provided, including the aspects of result match as well as code development and execution, with o3-mini serving as the judge. Then, as shown in Table 8, PaperCoder achieves the highest score. Also, for Paper2CodeBench, we adopt the rubric defined by the paper authors, covering Data Processing, Method, and Evaluation, with o4-mini as the judge, and as shown in Figure 6, PaperCoder outperforms all baselines regardless of whether debugging is used. These results show that its repositories are not only executable with minimal (and automatically debuggable) intervention but also more faithfully reproduce the papers.

**Case Study**   We further conduct a manual case study on five repositories, where annotators check whether the returned outputs match the reported results. As described in Table 18 with Appendix A.5, four reproduce results (at least partially), while one fails due to issues in loss function design.

## 5   CONCLUSION

In this work, we introduced PaperCoder, a framework that automatically generates code repositories from research papers in machine learning through a structured, three-stage pipeline. Specifically, we defined a high-level roadmap, system architecture, execution logic, and configuration via the planning stage, which are then enhanced through detailed per-file analysis, followed by the sequential code generation informed by artifacts from prior stages. To validate PaperCoder, we performed evaluations on two benchmarks: our Paper2CodeBench, comprising recent papers from top-tier machine learning venues, and (recently released) PaperBench Code-Dev, providing fine-grained evaluation protocols, on which PaperCoder consistently outperforms existing baselines on both model-based and human evaluations. Furthermore, additional analyses demonstrate its robustness and practicality: it remains effective across different LLM backbones, shows practical executability with only 0.81% of the lines requiring minor fixes, and benefits from each stage in the pipeline. We envision PaperCoder as one important step toward accelerating scientific progress by aiding the reproduction of research papers.

## ETHICS STATEMENT

Our work aims to generate faithful code repositories from scientific papers in machine learning, and we believe it has a substantial positive impact in contributing to open science and facilitating rapid experimentation. However, we also acknowledge potential risks and misuse of our framework. For example, some papers intentionally refrain from releasing implementations due to security concerns, such as those involving jailbreaking or exploitation techniques. Yet, our method could potentially be used to reproduce such sensitive implementations. To address such risks, in real-world production, it would be necessary to develop and incorporate safeguards (such as harmful content filters, protective prompting, and secure execution environments) to ensure responsible and safe use of our framework.

## REPRODUCIBILITY STATEMENT

We provide the public GitHub link to the code to reproduce our work. Detailed instructions for running the experiments are included in the accompanying README files, and furthermore, all necessary details to reproduce our experiments are described in Section 4.1 and in Appendix A.1.

## ACKNOLEDGEMENTS

This work was supported by the Institute for Information & communications Technology Planning & Evaluation (IITP) grant funded by the Korea government (MSIT) (RS-2019-II190075, Artificial Intelligence Graduate School Program (KAIST)), the National Research Foundation of Korea (NRF) grant funded by the Korea government (MSIT) (No. RS-2023-00256259), the Institute of Information & Communications Technology Planning & Evaluation (IITP) with a grant funded by the Ministry of Science and ICT (MSIT) of the Republic of Korea in connection with the Global AI Frontier Lab International Collaborative Research (No. RS-2024-00469482 & RS-2024-00509279), and the InnoCORE program of the Ministry of Science and ICT (No. N10250156).

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

# APPENDIX

## A  ADDITIONAL EXPERIMENTAL DESIGNS

### A.1  IMPLEMENTATION DETAILS

All experiments are conducted using `o3-mini` with high reasoning effort version (`o3-mini-high`) as the default backbone, released on January 31, 2025. To collect paper metadata and content, we use `openreview_scraper`[4] with the OpenReview API[5] and Semantic Scholar API[6]. For document processing, we convert papers into structured JSON format using the `s2orc-doc2json` library (Lo et al., 2020)[7]. Notably, with `o3-mini-high` to generate repositories for 90 papers, the total API cost of PaperCoder amounts to $76.65, resulting in an average cost of approximately $0.90 per paper.

### A.2  HUMAN EVALUATION PROCESS

Given the complexity of the task (requiring comprehension of scientific papers and their associated implementations), we recruit participants who have at least one peer-reviewed paper and a degree in computer science. We note that they were compensated at a rate of $15 per hour. For annotation, they were provided with a 4-page document, which includes task instructions, annotation examples, and 10 generated repositories grouped into three sets, as follows: (Group 1) Model Variants of Our Method that includes repositories generated by our system using different backbone models (e.g., `o3-mini` vs. three open-source alternatives); (Group 2) Naive Baselines that includes repositories generated using only the Paper or the Abstract as input; and (Group 3) Related Works that includes repositories generated by existing software development frameworks, such as MetaGPT and ChatDev. Each repository was anonymized using a `repo X` naming format to prevent bias regarding the generation method. Following the question guidelines in the document, annotators reviewed and evaluated the repositories generated by different methods and models. Also, on average, evaluating 10 repositories for a single paper took approximately 45 minutes. Table 41 shows a detailed annotation example.

### A.3  REFERENCE-BASED EVALUATION

In the reference-based evaluation setup, the repository may exceed the context length of (even frontier) LLMs. Following Starace et al. (2025), when this occurs, we prompt the model to select the most relevant files for evaluation. The selected subset is then used as the reference for scoring. We use the gpt-4o-2024-11-20 as the evaluation model.

### A.4  PAPERBENCH CODE-DEV EVALUATION

While PaperCoder is designed to generate only the source code, the PaperBench Code-Dev benchmark used for evaluation requires an additional script file called `reproduce.sh`. To meet this requirement, we further prompt the coding agent to generate it and evaluate the code with it.

### A.5  ADDITIONAL DETAILS ON EXECUTION AND REPRODUCIBILITY EXPERIMENTS

To assist the reproduction of repositories from PaperCoder, we perform LLM-assisted automatic debugging. Specifically, we primarily use o4-mini for debugging, with GPT-5 used as a fallback when identical errors persist. Furthermore, all executions are performed in a Docker environment with an NVIDIA GeForce RTX 2080 GPU, and for experiments requiring larger memory, an NVIDIA RTX A6000. Lastly, due to hardware constraints, we adjust certain hyperparameters (e.g., batch size or learning rate), and in rare cases, subsampled the training data to enable successful execution. We provide the prompts in Figure 22, and statistics on the number of modified lines in Table 19.

---

[4]https://github.com/pranftw/openreview_scraper
[5]https://docs.openreview.net/reference/api-v2
[6]https://www.semanticscholar.org/product/api
[7]https://github.com/allenai/s2orc-doc2json

Table 9: Code availability across major machine learning conferences. We report the total number of accepted papers, the number of papers with publicly available code (identified via GitHub URLs in ArXiv abstracts), and the corresponding percentage for each venue. The last row shows the average across all three conferences.

| Conference | # of Accepted | w/ Code | Percentage (%) |
|---|---|---|---|
| ICLR 2024 | 2207 | 467 | 21.2 |
| ICML 2024 | 2610 | 435 | 16.7 |
| NeurIPS 2024 | 4006 | 825 | 20.6 |
| Average | 2941 | 576 | 19.5 |

Table 10: Average Replication Scores (%) on PaperBench Code-Dev. For all OpenAI models, the reasoning effort is set to high, and we take results for BasicAgent and IterativeAgent from Starace et al. (2025). For PaperCoder, we report the average and standard deviation over three runs, except for o1 and o3 due to costs.

| Model | Replication Score (%) | Cost per Paper ($) |
|---|---|---|
| BasicAgent (o3-mini) | $5.1 \pm 0.8$ | N/A |
| BasicAgent (o1) | $19.5 \pm 1.2$ | N/A |
| BasicAgent (claude-3-5-sonnet) | $35.4 \pm 0.8$ | N/A |
| IterativeAgent (o3-mini) | $16.4 \pm 1.4$ | N/A |
| IterativeAgent (o1) | $43.3 \pm 1.1$ | 400.00 |
| IterativeAgent (claude-3-5-sonnet) | $27.5 \pm 1.6$ | N/A |
| PaperCoder (o3-mini) | $45.14 \pm 0.3$ | 0.69 |
| PaperCoder (o1) | 38.31 | 8.81 |
| PaperCoder (o3) | 60.86 | 8.99 |
| PaperCoder (claude-3-5-sonnet) | $51.14 \pm 1.4$ | 3.61 |

# B  ADDITIONAL EXPERIMENTAL RESULTS AND ANALYSIS

## B.1  CODE AVAILABILITY

To estimate the proportion of accepted papers that release official code repositories, we collect data from three major machine learning conferences in 2024: ICLR, ICML, and NeurIPS. Specifically, we first retrieve the list of accepted papers from each conference using the OpenReview API[8] via `openreview_scraper`[9]. While OpenReview abstracts sometimes include repository links, they are more commonly found in ArXiv[10] abstracts. Therefore, we additionally use the Semantic Scholar API[11] to obtain ArXiv abstracts corresponding to the accepted papers. We then check whether the abstract includes a GitHub URL as an indicator of released code. Table 9 summarizes the number of accepted papers, the number with publicly available repositories, and the corresponding percentages for each conference. On average, only 19.5% of accepted papers in them provide official code.

## B.2  ADDITIONAL ANALYSIS ON EXECUTABILITY

As discussed in Section 4.3, we observe that the fixes required for execution are overwhelmingly simple, which are minor syntax issues, missing imports, or small adjustments to variable names rather than logic- or architecture-level revisions, as shown in Figure 8 to 12 with statistics in Table 15.

Table 11: Developer time comparison.

| Method | Developer Time |
|---|---|
| CoLoR | 5 |
| cognitive-behaviors | 6 |
| RADA | 27 |
| self-instruct | 53 |
| geval | 22 |

To quantify this more concretely, we further estimate developer time by multiplying the number of modified lines by a difficulty factor (1 - 3):

- **1**: simple syntax or typo fixes, variable renaming, comment adjustments
- **2**: fixes requiring local reasoning (e.g., adjusting conditions or API usage)
- **3**: nontrivial issues requiring deeper debugging (asynchronous or memory-related errors)

The results of this estimation are then reported in the developer time column of Table 11, which demonstrates that correcting the errors does not require a significant amount of time.

---

[8]https://docs.openreview.net/reference/api-v2

[9]https://github.com/pranftw/openreview_scraper

[10]https://arxiv.org/

[11]https://www.semanticscholar.org/product/api

In addition to human debugging, we also perform LLM-assisted repair to examine whether the generated code can be fixed automatically without human intervention. We use `o4-mini-high` and `GPT-5` during the repair stage. For each repair round, we record (i) the number of iterations required, and (ii) the error categories identified using `GPT-5.1`. The results, included in Table 14, then show that this automatic repair strategy resolves all the issues mostly within a small number of iterations and that the errors are primarily syntactic or import-related rather than logic-level.

### B.3 PAPERBENCH CODE-DEV RESULTS

We conduct additional experiments using various reasoning models, as shown in Table 10. Overall, our method achieves strong replication scores across models. Notably, when using o3, PaperCoder records the highest score of 60.86%. These results suggest that the latest and larger models, particularly those with stronger reasoning and coding capabilities, tend to yield better performance.

### B.4 IMPACT OF PAPER CONTENT ON CODE GENERATION

To examine the extent to which the clarity and specificity of the paper content influence code generation quality, we remove the Methodology section from each paper and use PaperCoder to generate the corresponding code repository. Specifically, this experiment is conducted with 30 papers (10 from each conference) in Paper2CodeBench, with `o3-mini-high` as the backbone LLM. As shown in Table 12, the average score drops from 4.26 to 3.75 without the Methodology section, indicating that when detailed specifications are absent, the generated code quality degrades substantially, which supports the importance of precise and explicit descriptions for faithful paper-to-code generation, as well as for human readers seeking to understand and reproduce the work.

Table 12: Comparison of reference-based average scores between the full paper content and the paper content without the methodology section on the subsampled Paper2CodeBench. Values in parentheses indicate the standard deviation.

|  | Full (Original) | w/o Methodology |
|---|---|---|
| Ref-based Average Score | 4.26 (0.28) | 3.75 (0.55) |

### B.5 MOST COMMON TYPES OF ERRORS AND FAILURE MODES

To analyze failure cases, we execute the generated repositories on Paper2CodeBench (without debugging) and inspect the resulting errors. We note that each error is automatically categorized by prompting `o4-mini-high` with the raw error message and mapping its response to a canonical taxonomy. As summarized in Table 13, the most frequent causes are MissingDependency, ImportError, and ModuleNotFoundError, in that order. This pattern suggests that environment and packaging issues dominate over algorithmic or logic errors in practice.

Table 13: Categories of error types observed when running Paper2CodeBench. Categories are analyzed using o4-mini-high, and Count indicates the number of papers belonging to each category.

| Category | Count | Category | Count |
|---|---|---|---|
| MissingDependency | 23 | ConfigurationError | 5 |
| ImportError | 14 | SyntaxError | 4 |
| ModuleNotFoundError | 14 | Success | 4 |
| ValueError | 6 | OSError | 4 |
| FileNotFoundError | 6 | TypeError | 2 |
| RuntimeError | 6 | AttributeError | 2 |

### B.6 ADDRESSING ENVIRONMENT SETUP

Our framework focuses on faithfully reconstructing the methodological pipeline, rather than fully automating environment configuration. In practice, environment setup tends to be far easier and more reliable for users to adjust manually, whereas reconstructing the core algorithmic logic from natural language is substantially more challenging and central to our goal. For this reason, PaperCoder prioritizes method-level faithfulness over full automation of environment specification, which is highly paper-specific and often varies across systems. Nevertheless, to further handle this environmental issue, we have conducted an additional experiment on 30 papers (10 from each conference in Paper2CodeBench), augmenting PaperCoder with prompts (shown in Figure 33) explicitly aimed at inferring and repairing environment requirements. Then, in these cases, we do not observe dependency-related failures, suggesting that dependency issues in Table 13 are not from the intrinsic weaknesses of the framework. We view incorporating a dedicated environment-construction or "DevOps" agent as an interesting extension for future work.

### B.7 Analysis of Performance Across Paper Categories

Examining performance across different paper categories helps reveal where code generation is easier or more challenging. To achieve this, we categorize 90 papers in Paper2CodeBench using o4-mini-high, and then report the average reference-based scores per category in Figure 13. First, we observe that the scores range from 3.38 to 4.21 (a maximum gap of about 0.83). Specifically, theory and interpretability/explainability achieve the highest scores (4.21 and 3.97), while reinforcement learning/control and dataset-focused papers yield the lowest (3.38 each). These results suggest that there are measurable variations across different categories of papers when implementing them with PaperCoder, with some types of papers being easier for PaperCoder to implement than others.

### B.8 Discussion Against Data Contamination in Paper2CodeBench

We evaluate target papers from ICLR/ICML/NeurIPS 2024, because the primary models used in our experiments (OpenAI o3-mini and GPT-4o) have a knowledge cutoff of October 2023, meaning these 2024 paper/code repositories fall after the training window of models. While we cannot fully rule out all forms of indirect exposure, the temporal gap substantially reduces the likelihood of contamination, and thus, data contamination is unlikely to meaningfully affect the results.

### B.9 Cross-Family Validation for Paper2CodeBench Model Evaluation

While human evaluation might be the most reliable form of assessment, conducting full-scale human evaluation for all baselines and papers would be prohibitively expensive (Zheng et al., 2023). Therefore, we adopt the standard LLM-as-a-judge strategy for primary assessment. Also, as shown in Table 5, o3-mini-high correlates strongly with human judgments (0.78 in the reference-based setting and 0.73 in the reference-free setting), suggesting that the LLM-based evaluation could be a reliable proxy for human assessments. Nevertheless, to further address the concern on the circularity and bias, specifically that an evaluator from the same model family might favor code generated by the same model family, we perform a cross-family evaluation with Gemini-Flash 2.5 on 30 randomly selected papers of the Paper2Code benchmark. The resulting Spearman correlation of 0.73 between Gemini-Flash 2.5 and o3-mini-high indicates that the evaluation remains consistent across different model families, providing strong evidence that our results are not an artifact of model-family alignment.

## C   Limitations and Future Work

While PaperCoder demonstrates strong performance in reproducing machine learning papers (where code implementations are particularly helpful and usually necessary for validating research ideas), its current scope is limited to this domain. Beyond this, we believe accelerating the reproduction of scientific discovery to other domains where code is not the primary medium for validation, such as theoretical mathematics, is an exciting direction for future work. In addition, the current version of PaperCoder processes only textual inputs, and extending it to process visual inputs (such as figures in papers) with recent OCR models capable of extracting figures and tables (in addition to text) (Wang et al., 2024a; He et al., 2024; Wei et al., 2025; Li et al., 2025; Zhang et al., 2025; Niu et al., 2025) is an interesting avenue. Lastly, as with other repository-level code generation approaches, improving executability remains an important (but still challenging) direction for future work.

Table 14: The number of repair iterations (with the LLM) and error categories. We use `o4-mini-high` and `GPT-5` for the automatic repair, and `GPT-5.1` for identifying the error categories.

| RepoName | Iteration | Error Category |
|---|---|---|
| CoLoR | 1 | Configuration Update |
| | 2 | Code Refactor / Robustness Enhancement |
| | 3 | Bug Fix |
| | 4 | Performance Adjustment / Behavior Control Update |
| Cognitive-Behaviors | 1 | Configuration Update |
| | 2 | Bug Fix / Type Compatibility Correction |
| | 3 | Autograd Control / Memory Optimization |
| | 4 | API Modernization / Correct Mask Handling / Behavior Tuning (Token Budget) |
| | 5 | Bug Fix / Robustness Improvement |
| | 6 | Representation Handling Feature + Refactor |
| | 7 | Configuration Robustness / Type Validation |
| | 8 | Training Efficiency Optimization / Memory Reduction / Autograd Hygiene |
| | 9 | Memory & Performance Optimization / Behavioral Correction |
| | 10 | Configuration Robustness / Type Validation |
| | 11 | Algorithmic Refactor / Performance & Memory Optimization |
| | | API Modernization / Robustness Improvement |
| RADA | 1 | Configuration Update |
| | 2 | API Modernization / Compatibility Update |
| Self-Instruct | 1 | Configuration Update |
| | 2 | API Migration / Compatibility Update |
| | 3 | API Modernization / SDK Refactor |
| | 4 | API Rollback / Correct Endpoint Restoration |
| | 5 | API Consistency Fix |
| | 6 | API Field Access Modernization |
| G-EVAL | 1 | API Modernization / SDK Migration |
| | 2 | Debug Cleanup / Schema Update |

Table 15: Executability analysis results on the repositories: we sample five papers and generate corresponding repositories using PaperCoder. For each repository, we report the number of lines modified during debugging, the total number of lines, and the percentage of modified lines.

| Repo Name | CoLoR | cognitive-behaviors | RADA | Self-Instruct | G-EVAL | Average |
|---|---|---|---|---|---|---|
| Modified lines (*.py) | 2 | 0 | 10 | 26 | 10 | 8 |
| Modified lines (config.yaml) | 3 | 6 | 7 | 1 | 4 | 3.5 |
| Total lines | 1132 | 2060 | 1609 | 1334 | 1374 | 1251.5 |
| Percentage | 0.44 | 0.29 | 1.06 | 2.02 | 1.02 | **0.81** |

Table 16: Qualitative analysis of top-ranked repositories. We categorize the reasons why human annotators select the repositories generated by our PaperCoder framework as their top choice into six (described in the first row). We also show an example of the response in Table 17.

| Completeness | Clean Structure | Faithfulness to Paper | Ease of Use | Code Quality | Unique Strengths |
|---|---|---|---|---|---|
| 16 | 13 | 8 | 6 | 7 | 4 |

Table 17: Example responses from human annotators for the reasons for the top-ranked repositories in the human evaluation. We ask the human evaluator (who is the first author of the paper to be reproduced by PaperCoder) the following question: "Among the top-ranked repositories in each group, which one do you think is the best? If the repositories are the same, you may select any of them. Please briefly explain your reasoning."

| RepoName | Response for the top-ranked repositories |
|---|---|
| Janus | The code includes everything needed to implement the paper. The code is clean and easy to understand (not overloaded with comments). It's not overly packaged, and since config files are provided, it's convenient for running various experiments. |
| VideoRAG | Each component—data processing, retrieval, frame selection, generation, and evaluation—is clearly separated into its own module, making the system easy to maintain and extend. Moreover, the most critical modules are fully implemented, covering the essential functionality required by the framework. |
| T1 | This repository successfully implements the following: Tool-based filtering, Calculate score of each generation after filtering using verifier model, Training code for distillation. |

Table 18: Analysis of the results from the reproducibility case study.

| Repo Name | Analysis of Reproducibility |
|---|---|
| CoLoR | Execution was successful, but the ORPO loss was likely mis-specified, causing the compression model to fail in training as intended. This issue stems from the overly simplified description of the loss function in the paper. |
| cognitive-behaviors | Successfully reproduced SFT and RL training processes but encountered a minor error in parsing model responses during evaluation. |
| RADA | Implementation closely matched the paper, but missing details prevented full reproduction of the reported results, leading to identical samples. |
| Self-Instruct | Executed smoothly and accurately reflected the procedure described in the paper. |
| G-EVAL | Implemented only the Coherence metric, though the original paper included Coherence, Consistency, Fluency, and Relevance. The Coherence implementation was faithful and correct. |

Table 19: Comparison of the total lines, modified lines, and percentages when applying automatic debugging on 10 papers from Paper2CodeBench used for human evaluation.

| | Abstract | Paper | MetaGPT | ChatDEV | PaperCoder |
|---|---|---|---|---|---|
| Modified lines (*.py, *.sh, *.yaml) | 30 | 705 | 226 | 275 | 780 |
| Total lines | 3517 | 3047 | 8225 | 4185 | 16189 |
| Percentage | 0.85 | 23.14 | 2.75 | 6.57 | 4.82 |

**+ Arch. Design**

```python
# main.py

# (... omitted ...)

class DatasetLoader:

# (... omitted ...)

    def load_data(self) -> Tuple[DataLoader,
        DataLoader, DataLoader]:
        """Returns DataLoaders for train, validation
        , and test splits."""
        train_dataset = TimeSeriesDataset(
            dataset_name=self.dataset_name, mode="
        train",
            lookback=self.lookback,
        prediction_horizon=self.prediction_horizon,
            num_samples=self.num_train
        )
        val_dataset = TimeSeriesDataset(
            dataset_name=self.dataset_name, mode="
        val",
            lookback=self.lookback,
        prediction_horizon=self.prediction_horizon,
            num_samples=self.num_val
        )
        test_dataset = TimeSeriesDataset(
            dataset_name=self.dataset_name, mode="
        test",
            lookback=self.lookback,
        prediction_horizon=self.prediction_horizon,
            num_samples=self.num_test
        )
        batch_size: int = self.config["training"].
        get("batch_size", 32)
        train_loader = DataLoader(train_dataset,
        batch_size=batch_size, shuffle=True)
        val_loader = DataLoader(val_dataset,
        batch_size=batch_size, shuffle=False)
        test_loader = DataLoader(test_dataset,
        batch_size=batch_size, shuffle=False)
        return train_loader, val_loader, test_loader

# (... omitted ...)

class Main:

# (... omitted ...)

    def run_experiment(self) -> None:
        for dataset_name in dataset_names:
            for run in range(self.num_runs):
                loader_instance = DatasetLoader(self
        .config, dataset_name)
                train_loader, val_loader,
        test_loader = loader_instance.load_data()
```

```python
# dataset_loader.py

# (... omitted ...)

class DatasetLoader:

# (... omitted ...)

    def load_data(self) -> Tuple[DataLoader,
        DataLoader, DataLoader]:
        batch_size: int = int(self.config.get("
        training", {}).get("batch_size", 32))
        train_loader: DataLoader = DataLoader(
        train_dataset, batch_size=batch_size, shuffle=
        True, num_workers=0)
        val_loader: DataLoader = DataLoader(
        val_dataset, batch_size=batch_size, shuffle=
        False, num_workers=0)
        test_loader: DataLoader = DataLoader(
        test_dataset, batch_size=batch_size, shuffle=
        False, num_workers=0)
        return train_loader, val_loader, test_loader
```

**+ Logic Design**

```python
# main.py

# (... omitted ...)

class Main:

# (... omitted ...)

    def run_experiment(self) ->
        None:
        num_runs: int = self.config.
        get("num_runs", 5)
        seed_base: int = self.
        config.get("seed", 42)

        for run in range(1,
        num_runs + 1):
            dataset_loader =
        DatasetLoader(self.config)
            data_loaders =
        dataset_loader.load_data()
            train_loader =
        data_loaders["train"]
            val_loader =
        data_loaders["val"]
            test_loader =
        data_loaders["test"]
```

```python
# dataset_loader.py

# (... omitted ...)

class DatasetLoader:

# (... omitted ...)

    def load_data(self) -> Tuple[
        DataLoader, DataLoader,
        DataLoader]:
        """
        Loads the time series data,
        segments it, splits it into
        training, validation, and
        test sets,
        and returns the
        corresponding DataLoader
        objects.

        Returns:
            Tuple[DataLoader,
        DataLoader, DataLoader]: (
        train_loader, val_loader,
        test_loader)
        """
        # Load CSV using pandas
        df: pd.DataFrame = pd.
        read_csv(self.data_file)

# (... omitted ...)
```

Figure 7: Case study on ablation experiments. Without a dependency-aware generation order, the Architecture Design stage causes main.py to duplicate functions from dataset_loader.py. In contrast, adding the Logic Design phase resolves this issue by aligning the file-generation order.

**CoLoR**

```
1  # config.yaml
2  model:
3    base_model: "microsoft/Phi-3-mini-4k-instruct"
4    alternative_models:
5  +    - "mistralai/Mistral-7B-Instruct-v0.3"
6  +    - "meta-llama/Llama-3.2-3B-Instruct"
```

```
1  # trainer.py
2  - self.optimizer = AdamW(self.model.model.parameters(), lr=lr)
3  + self.optimizer = AdamW(self.model.model.parameters(), lr=float(lr))
```

```
1  # model.py
2  - self.model = AutoModelForCausalLM.from_pretrained(base_model)
3  + self.model = AutoModelForCausalLM.from_pretrained(
4  +     base_model, trust_remote_code=True
5  + )
```

Figure 8: Case study on reproducing a paper, called Efficient Long-Context Language Model Retrieval with Compression (Seo et al., 2025) (repository: CoLoR), displaying the files modified during manual debugging.

**cognitive-behaviors**

```
1   # config.yaml
2   -     peak_learning_rate: 1e-5
3   +   peak_learning_rate: 0.00001
4
5   -    actor_learning_rate: 1e-6
6   -    critic_learning_rate: 1e-5
7   +   actor_learning_rate: 0.000001
8   +   critic_learning_rate: 0.00001
9
10  model:
11    base_models:
12  -     - "Llama-3.2-3B"
13  -     - "Qwen-2.5-3B"
14  +     - "meta-llama/Llama-3.2-3B"
15  +     - "Qwen/Qwen2.5-1.5B"
16
17    additional_models:
18  -     - "Llama-3.1-70B"
19  +     - "meta-llama/Llama-3.1-3B"
```

Figure 9: Case study on reproducing a paper, called Cognitive Behaviors that Enable Self-Improving Reasoners, or, Four Habits of Highly Effective STaRs (Gandhi et al., 2025) (repository: cognitive-behavior), displaying the files modified during manual debugging.

RADA

```
1   # config.yaml
2   t5_base:
3   -     model_name: "t5-base"
4   +     model_name: "google-t5/t5-base"
5
6   llama2_7b:
7   -     model_name: "Llama2-7B"
8   +     model_name: "meta-llama/Llama-2-7b"
9
10  - augmentation_model: "Llama2-7B-Chat"
11  + augmentation_model: "meta-llama/Llama-2-7b-chat"
12
13  retrieval:
14  -   embedding_model: "distilbert-base-nli-stsb-mean-tokens"
15  +   embedding_model: "sentence-transformers/distilbert-base-nli-mean-tokens"
16
17  + seed_data_path: data/seed_data.json
18  + external_data_path: data/external_data.csv
```

```
1   # main.py
2   -     generator: Generator = Generator(llm_model=augmentation_model)
3   +     generator: Generator = Generator(
4   +         llm_model=augmentation_model,
5   +         generation_params={
6   +             "max_length": 2048,
7   +             "temperature": 0.7,
8   +             "top_k": 50,
9   +             "top_p": 0.95,
10  +             "num_return_sequences": 1
11  +         }
12  +     )
```

Figure 10: Case study on reproducing a paper, called Retrieval-Augmented Data Augmentation for Low-Resource Domain Tasks (Seo et al., 2024) (repository: RADA), displaying the files modified during manual debugging.

self-instruct

```
1   # config.yaml
2   - engine: davinci
3   + engine: gpt-4.1-nano
```

```
1   # dataset_loader.py
2   - import openai
3   + from openai import OpenAI
4   + client = OpenAI(api_key=os.environ.get("OPENAI_API_KEY"))
5
6   - response = openai.Completion.create(
7   -     engine=self.engine,
8   -     prompt=prompt,
9   -     max_tokens=150,
10  + response = client.chat.completions.create(
11  +     model=self.engine,
12  +     messages=[{"role": "user", "content": prompt}],
13  +     max_completion_tokens=150,
14    )
15  - raw_text = response.choices[0].text.strip()
16  + raw_text = response.choices[0].message.content.strip()
17
18  - answer = response.choices[0].text.strip().lower()
19  + answer = response.choices[0].message.content.strip().lower()
20
21  - generated_text = response.choices[0].text.strip()
22  + generated_text = response.choices[0].message.content.strip()
```

Figure 11: Case study on reproducing a paper, called Self-Instruct: Aligning Language Models with Self-Generated Instructions (Wang et al., 2023) (repository: self-instruct), displaying the files modified during manual debugging.

```
geval
```

```
1  # config.yaml
2  -    name: text-davinci-003
3  +    name: gpt-4.1-nano
4
5  + summ_eval_name: data/summeval_1.csv
6  + dialogue_name: data/topical_chat.csv
7  + hallucination_name: data/qags.csv
```

```
1  # config.py
2  -                    "name": "gpt-4",
3  +                    "name": "gpt-4o-mini",
```

```
1  # llm_evaluator.py
2  + from openai import OpenAI
3  + client = OpenAI(api_key=os.environ.get("OPENAI_API_KEY"))
4
5  - self.samples: int = int(params.get("samples", 20)) if self.model_name.lower() == "gpt-4" else 1
6  + self.samples: int = int(params.get("samples", 20)) if "gpt-4" in self.model_name.lower() else 1
7
8  - if self.model_name.lower() == "gpt-4":
9  -     response = openai.ChatCompletion.create(
10 + if "gpt-4" in self.model_name.lower():
11 +     response = client.chat.completions.create(
12
13 -     max_tokens=self.max_tokens,
14 +     max_completion_tokens=self.max_tokens,
15
16 -     text: str = response["choices"][0]["message"]["content"].strip()
17 +     text = response.choices[0].message.content.strip()
18
19 -     choice["message"]["content"].strip() for choice in response.get("choices", [])
20 +     choice.message.content.strip() for choice in response.choices
21
22 - if score_normalization and self.model_name.lower() == "gpt-4":
23 + if score_normalization and "gpt-4" in self.model_name.lower():
```

Figure 12: Case study on reproducing a paper, called G-Eval: NLG Evaluation using GPT-4 with Better Human Alignment(Liu et al., 2023) (repository: geval), displaying the files modified during manual debugging.

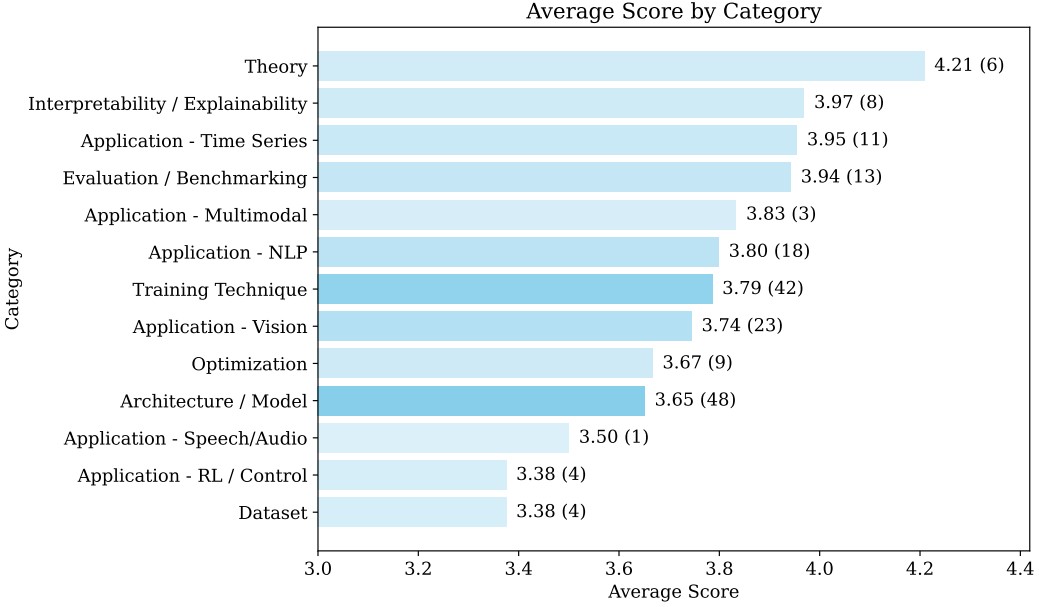

Figure 13: Average scores (measured by reference-based evaluation) per category on Paper2CodeBench. The numbers to the right of each bar indicate the average score, along with the number of papers in parentheses. Bar transparency is proportional to the count, highlighting categories with more or fewer papers.

# D  PROMPTS

---

**Prompt for generating the overall plan in the planning stage**

**[System]**

You are an expert researcher and strategic planner with a deep understanding of experimental design and reproducibility in scientific research.
You will receive a research paper in JSON format.
Your task is to create a detailed and efficient plan to reproduce the experiments and methodologies described in the paper.
This plan should align precisely with the paper's methodology, experimental setup, and evaluation metrics.

Instructions:

1. Align with the Paper: Your plan must strictly follow the methods, datasets, model configurations, hyperparameters, and experimental setups described in the paper.
2. Be Clear and Structured: Present the plan in a well-organized and easy-to-follow format, breaking it down into actionable steps. 3. Prioritize Efficiency: Optimize the plan for clarity and practical implementation while ensuring fidelity to the original experiments.

**[User]**
## Paper
{paper_json}

## Task
1. We want to reproduce the method described in the attached paper.
2. The authors did not release any official code, so we have to plan our own implementation.
3. Before writing any Python code, please outline a comprehensive plan that covers:
- Key details from the paper's **Methodology**.
- Important aspects of **Experiments**, including dataset requirements, experimental settings, hyperparameters, or evaluation metrics.
4. The plan should be as **detailed and informative** as possible to help us write the final code later.

## Requirements
- You don't need to provide the actual code yet; focus on a **thorough, clear strategy**.
- If something is unclear from the paper, mention it explicitly.

## Instruction
The response should give us a strong roadmap, making it easier to write the code later.

---

Figure 14: Prompt for generating the overall plan in the planning stage.

---

**Prompt for generating the architecture design in the planning stage**

**[User]**

Your goal is to create a concise, usable, and complete software system design for reproducing the paper's method. Use appropriate open-source libraries and keep the overall architecture simple.

Based on the plan for reproducing the paper's main method, please design a concise, usable, and complete software system. Keep the architecture simple and make effective use of open-source libraries.

___

## Format Example
[CONTENT]
{
    "Implementation approach": "We will ... ,
    "File list": [
        "main.py",
        "dataset_loader.py",
        "model.py",
        "trainer.py",
        "evaluation.py"
    ],
    "Data structures and interfaces": "\nclassDiagram\n    class Main \n        +__init__()\n        +run_experiment()\n    \n class    DatasetLoader    \n        +__init__(config:        dict)\n        +load_data()    ->    Any\n    \n    class Model    \n        +__init__(params:    dict)\n        +forward(x:    Tensor) -> Tensor\n    \n    class Trainer \n        +__init__(model:    Model,    data:    Any)\n        +train()    ->    None\n    \n    class    Evaluation \n        +__init__(model: Model, data: Any)\n        +evaluate() -> dict\n    \n    Main --> DatasetLoader\n    Main --> Trainer\n    Main --> Evaluation\n    Trainer --> Model\n",
    "Program call flow": "\nsequenceDiagram\n    participant M as Main\n    participant DL as DatasetLoader\n    participant MD as Model\n    participant TR as Trainer\n    participant EV as Evaluation\n    M-»DL: load_data()\n    DL-»M: return dataset\n    M-»MD: initialize model()\n    M-»TR: train(model, dataset)\n    TR-»MD: forward(x)\n    MD-»TR: predictions\n    TR-»M: training complete\n    M-»EV: evaluate(model, dataset)\n    EV-»MD: forward(x)\n    MD-»EV: predictions\n    EV-»M: metrics\n",
    "Anything UNCLEAR": "Need clarification on the exact dataset format and any specialized hyperparameters."
}
[/CONTENT]

## Nodes: "<node>: <type> # <instruction>"
- Implementation approach: <class 'str'> # Summarize the chosen solution strategy.
- File list: typing.List[str] # Only need relative paths. ALWAYS write a main.py or app.py here.
- Data structures and interfaces: typing.Optional[str] # Use mermaid classDiagram code syntax, including classes, method(__init__ etc.) and functions with type annotations, CLEARLY MARK the RELATIONSHIPS between classes, and comply with PEP8 standards. The data structures SHOULD BE VERY DETAILED and the API should be comprehensive with a complete design.
- Program call flow: typing.Optional[str] # Use sequenceDiagram code syntax, COMPLETE and VERY DETAILED, using CLASSES AND API DEFINED ABOVE accurately, covering the CRUD AND INIT of each object, SYNTAX MUST BE CORRECT.
- Anything UNCLEAR: <class 'str'> # Mention ambiguities and ask for clarifications.

## Constraint
Format: output wrapped inside [CONTENT][/CONTENT] like the format example, nothing else.

## Action
Follow the instructions for the nodes, generate the output, and ensure it follows the format example.

Figure 15: Prompt for generating the architecture design in the planning stage. This prompt follows the previous prompt and response shown in Figure 14.

---

**Prompt for generating the logic design in the planning stage**

**[User]**
Your goal is break down tasks according to PRD/technical design, generate a task list, and analyze task dependencies.
You will break down tasks, analyze dependencies.
You outline a clear PRD/technical design for reproducing the paper's method and experiments.

Now, let's break down tasks according to PRD/technical design, generate a task list, and analyze task dependencies.
The Logic Analysis should not only consider the dependencies between files but also provide detailed descriptions to assist in writing the code needed to reproduce the paper.

---

## Format Example
[CONTENT]
```
{
    "Required packages": [
        "numpy==1.21.0",
        "torch==1.9.0"
    ],
    "Required Other language third-party packages": [
        "No third-party dependencies required"
    ],
    "Logic Analysis": [
        [
            "data_preprocessing.py",
            "DataPreprocessing class ........"
        ],
        [
            "trainer.py",
            "Trainer ....... "
        ],
        [
            "dataset_loader.py",
            "Handles loading and ........"
        ],
        [
            "model.py",
            "Defines the model ......."
        ],
        [
            "evaluation.py",
            "Evaluation class ........ "
        ],
        [
            "main.py",
            "Entry point ......."
        ]
    ],
    "Task list": [
        "dataset_loader.py",
        "model.py",
        "trainer.py",
        "evaluation.py",
        "main.py"
    ],
    "Full API spec": "openapi: 3.0.0 ...",
    "Shared Knowledge": "Both data_preprocessing.py and trainer.py share ........",
    "Anything UNCLEAR": "Clarification needed on recommended hardware configuration for large-scale experiments."
}
```

[/CONTENT]

## Nodes: "<node>: <type> # <instruction>"
- Required packages: typing.Optional[typing.List[str]] # Provide required third-party packages in requirements.txt format.(e.g., 'numpy==1.21.0').
- Required Other language third-party packages: typing.List[str] # List down packages required for non-Python languages. If none, specify "No third-party dependencies required".
- Logic Analysis: typing.List[typing.List[str]] # Provide a list of files with the classes/methods/functions to be implemented, including dependency analysis and imports. Include as much detailed description as possible.
- Task list: typing.List[str] # Break down the tasks into a list of filenames, prioritized based on dependency order. The task list must include the previously generated file list.
- Full API spec: <class 'str'> # Describe all APIs using OpenAPI 3.0 spec that may be used by both frontend and backend. If front-end and back-end communication is not required, leave it blank.
- Shared Knowledge: <class 'str'> # Detail any shared knowledge, like common utility functions or configuration variables.
- Anything UNCLEAR: <class 'str'> # Mention any unresolved questions or clarifications needed from the paper or project scope.

## Constraint
Format: output wrapped inside [CONTENT][/CONTENT] like the format example, nothing else.

## Action
Follow the node instructions above, generate your output accordingly, and ensure it follows the given format example.

---

Figure 16: Prompt for generating the logic design in the planning stage. This prompt follows the previous prompt and response shown in Figure 15.

**Prompt for generating the configuration file in the planning stage**

[User]
You write elegant, modular, and maintainable code. Adhere to Google-style guidelines.

Based on the paper, plan, design specified previously, follow the "Format Example" and generate the code.
Extract the training details from the above paper (e.g., learning rate, batch size, epochs, etc.), follow the "Format example" and generate the code. DO NOT FABRICATE DETAILS — only use what the paper provides.

You must write 'config.yaml'.

ATTENTION: Use '##' to SPLIT SECTIONS, not '#'. Your output format must follow the example below exactly.

-----

# Format Example
## Code: config.yaml
```yaml
## config.yaml
training:
    learning_rate: ...
    batch_size: ...
    epochs: ...
...
```

-----

## Code: config.yaml

Figure 17: Prompt for generating the configuration file in the planning stage. This prompt follows the previous prompt and response shown in Figure 16.

---

**Prompt for analysis**

**[System]**
You are an expert researcher, strategic analyzer and software engineer with a deep understanding of experimental design and reproducibility in scientific research.
You will receive a research paper in JSON format, an overview of the plan, a design in JSON format consisting of "Implementation approach", "File list", "Data structures and interfaces", and "Program call flow", followed by a task in JSON format that includes "Required packages", "Required other language third-party packages", "Logic Analysis", and "Task list", along with a configuration file named "config.yaml".

Your task is to conduct a comprehensive logic analysis to accurately reproduce the experiments and methodologies described in the research paper.
This analysis must align precisely with the paper's methodology, experimental setup, and evaluation criteria.

1. Align with the Paper: Your analysis must strictly follow the methods, datasets, model configurations, hyperparameters, and experimental setups described in the paper.
2. Be Clear and Structured: Present your analysis in a logical, well-organized, and actionable format that is easy to follow and implement.
3. Prioritize Efficiency: Optimize the analysis for clarity and practical implementation while ensuring fidelity to the original experiments.
4. Follow design: YOU MUST FOLLOW "Data structures and interfaces". DONT CHANGE ANY DESIGN. Do not use public member functions that do not exist in your design.
5. REFER TO CONFIGURATION: Always reference settings from the config.yaml file. Do not invent or assume any values—only use configurations explicitly provided.

**[User]**
# Context
## Paper
{The content of the paper in json format}

——

## Overview of the plan
{The content of the overall plan}

——

## Design
{The content of the architecture design}

——

## Task
{The content of the logic design}

——

## Configuration file
```yaml
{The content of the configuration file}
```
——

## Instruction
Conduct a Logic Analysis to assist in writing the code, based on the paper, the plan, the design, the task and the previously specified configuration file (config.yaml).
You DON'T need to provide the actual code yet; focus on a thorough, clear analysis.

Write the logic analysis in '{The name of the file to be generated"}', which is intended for '{Description of the file generated through the "Logic Analysis" step of the logic design.}'.

——

## Logic Analysis: {todo_file_name}

Figure 18: Prompt for analysis. {} indicate placeholders to be filled with the content described in the accompanying explanation. The prompt is presented to the LLM for each file, following the sequence defined in the logic design.

**Prompt for coding**

**[System]**
You are an expert researcher and software engineer with a deep understanding of experimental design and reproducibility in scientific research.
You will receive a research paper in JSON format, an overview of the plan, a Design in JSON format consisting of "Implementation approach", "File list", "Data structures and interfaces", and "Program call flow", followed by a Task in JSON format that includes "Required packages", "Required other language third-party packages", "Logic Analysis", and "Task list", along with a configuration file named "config.yaml".
Your task is to write code to reproduce the experiments and methodologies described in the paper.

The code you write must be elegant, modular, and maintainable, adhering to Google-style guidelines.
The code must strictly align with the paper's methodology, experimental setup, and evaluation metrics.
Write code with triple quoto.

**[User]**
# Context
## Paper
{The content of the paper in json format}

——

## Overview of the plan
{The content of the overall plan}

——

## Design
{The content of the architecture design}

——

## Task
{The content of the logic design}

——

## Configuration file
```yaml
{The content of the configuration file}
```

——

## Code Files
{The content of the code files generated in the previous step.}

——

# Format example
## Code: {todo_file_name}
```python
## todo_file_name
...
```

——

# Instruction
Based on the paper, plan, design, task and configuration file(config.yaml) specified previously, follow "Format example", write the code.

We have {done_file_lst}.
Next, you must write only the "{todo_file_name}".
1. Only One file: do your best to implement THIS ONLY ONE FILE.
2. COMPLETE CODE: Your code will be part of the entire project, so please implement complete, reliable, reusable code snippets.
3. Set default value: If there is any setting, ALWAYS SET A DEFAULT VALUE, ALWAYS USE STRONG TYPE AND EXPLICIT VARIABLE. AVOID circular import.
4. Follow design: YOU MUST FOLLOW "Data structures and interfaces". DONT CHANGE ANY DESIGN. Do not use public member functions that do not exist in your design.
5. CAREFULLY CHECK THAT YOU DONT MISS ANY NECESSARY CLASS/FUNCTION IN THIS FILE.
6. Before using a external variable/module, make sure you import it first.
7. Write out EVERY CODE DETAIL, DON'T LEAVE TODO.
8. REFER TO CONFIGURATION: you must use configuration from "config.yaml". DO NOT FABRICATE any configuration values.

{detailed_logic_analysis}

## Code: {todo_file_name}

Figure 19: Prompt for coding. {} indicate placeholders to be filled with the content described in the accompanying explanation. The prompt is presented to the LLM for each file, following the sequence defined in the logic design. Previously generated code files are accumulated and provided as part of the ## Code Files input.

**Prompt for model-based reference-based evaluation**

**[System]**
You will be given a research paper along with two corresponding code repositories: a gold repository and a target repository.

Your task is to compare the target repository against the gold repository, rate the target repository on one metric, and provide a critique highlighting key differences.

Please make sure you read and understand these instructions carefully. Keep this document open while reviewing, and refer to it as needed.

—

Evaluation Criteria:

Correctness (1-5): The quality of the target repository in accurately implementing the paper's concepts, methodology, and algorithms without logical errors, as compared to the gold repository. Additionally, provide a critique focusing on the completeness, accuracy, and implementation choices made in the target repository relative to the gold repository.

1: Very Poor. The target repository does not correctly implement the core concepts, methodology, or algorithms from the paper. Major logical errors or missing components are present, especially when compared to the gold repository.
2: Poor. The target repository attempts to implement the paper's concepts but contains significant mistakes or missing components, making the implementation incorrect when compared to the gold repository.
3: Fair. Some core components and concepts are correctly implemented in the target repository, but there are notable logical errors or inaccuracies compared to the gold repository.
4: Good. The target repository correctly implements the key components and methodology, with only minor inaccuracies or deviations from the gold repository.
5: Excellent. The target repository fully and accurately implements all relevant key components, methodology, and algorithms from the paper, matching the quality of the gold repository.

—

Evaluation Steps

1. Identify Key Aspects of the Paper: Carefully read the research paper to understand its core concepts, methodology, and algorithms. Pay close attention to the key aspects that are crucial for implementing the paper's results (e.g., specific algorithms, data preprocessing steps, evaluation protocols).

2. Analyze the Gold Repository: Examine the gold repository to understand how these key aspects have been implemented. Use the gold repository as a reference for how the paper's methodology should be translated into code. Note the completeness, accuracy, and design choices in the gold repository that faithfully represent the paper's concepts.

3. Examine the Target Repository: Analyze the target repository to assess how well it implements the key aspects of the paper. Reference the gold repository as a guide for understanding these key aspects in the target repository. Focus on whether the target repository's core logic, algorithms, and structure align with the methodology and experiments described in the paper.

4. Identify Logical Errors and Deviations: Check for logical errors, missing steps, or deviations from the paper's methodology. Note any incorrect representations, inconsistencies, or incomplete implementations that could affect the correctness of the target repository.

5. Provide a Critique: Consider both the completeness and accuracy of the implementation relative to the paper's goals and the gold repository's standard. You do not need to analyze minor details like logging functions, script organization, or documentation quality. Instead, concentrate on the correctness of the logic and implementation that ensures the core concepts from the paper are fully reflected in the target repository. For each mismatch or deviation in implementation, note down specific critiques comparing relevant functions in the target repository to the corresponding functions in the gold repository. Highlight incorrect logic, missing steps, or deviations that affect the correct implementation of the paper's methodology.

5. Assess the Correctness: Determine whether the target repository includes all the critical elements described in the paper and implemented in the gold repository. Identify missing components, significant deviations, or incorrect implementations that could affect the correctness of the target repository.

6. Assign a Score: Based on your evaluation, provide a critique and assign a correctness score from 1 to 5 for the target repository, reflecting how well it implements the key aspects of the paper refer to the gold repository. Include a detailed critique in the specified JSON format.

—

Severity Level:

Each identified critique will be assigned a severity level based on its impact on the correctness of the methodology implementation.

- High: Missing or incorrect implementation of the paper's core concepts, major loss functions, or experiment components that are fundamental to reproducing the paper's methodology.
- Example: The main algorithm is missing or fundamentally incorrect.
- Medium: Issues affecting training logic, data preprocessing, or other core functionalities that significantly impact performance but do not completely break the system.
- Example: Improper training loop structure, incorrect data augmentation, or missing essential components in data processing.
- Low: Errors in specific features that cause deviations from expected results but can be worked around with modifications. Any errors in the evaluation process belong to this category unless they impact the core concepts. These include minor issues like logging, error handling mechanisms, configuration settings, evaluation steps that do not alter the fundamental implementation and additional implementations not explicitly stated in the paper.
- Example: Suboptimal hyperparameter initialization, incorrect learning rate schedule, inaccuracies in evaluation metrics, using a different random seed, variations in batch processing, different weight initialization, issues in result logging or reporting, variations in evaluation dataset splits, improper error handling in non-critical steps, mismatches in secondary evaluation criteria, or additional implementation details not specified in the paper that do not interfere with core results.

---

**Prompt for model-based reference-based evaluation**

—

Example JSON format:
```json
{
    "critique_list": [
        {
            "gold_file_name": "preprocessing.py",
            "gold_func_name": "data_process",
            "target_file_name": "dataset.py",
            "target_func_name": "train_preprocess",
            "severity_level": "medium",
            "critique": "A critique of the target repository's file with reference to the gold repository."
        },
        {
            "gold_file_name": "utils.py",
            "gold_func_name": "calculate_metric",
            "target_file_name": "metric.py",
            "target_func_name": "f1_at_k"
            "severity_level": "low",
            "critique": "A critique of the target repository's file with reference to the gold repository."
        },
    ],
    "score": 2
}
```

—

Sample:

Research Paper:

{{The content of the paper}}

Gold Repository:

{{The gold repository, officially released by the authors, serves as the reference implementation.}}

Target Repository:

{{The generated repository, which serves as the target repository for evaluation.}}

—

Please provide critique of the target repository and a single numerical rating (1, 2, 3, 4, or 5) based on the quality of the sample, following the Example JSON format, without any additional commentary, formatting, or chattiness.

Figure 20: Prompt for model-based reference-based evaluation. {{}} indicate placeholders to be filled with the content described in the accompanying explanation.

**Prompt for model-based reference-free evaluation**

**[System]**

You will be given a research paper along with its corresponding code repository.

Your task is to rate the code repository on one metric and provide a critique highlighting key differences.

Please make sure you read and understand these instructions carefully. Keep this document open while reviewing, and refer to it as needed.

—

Evaluation Criteria:

Correctness (1-5): The quality of the repository in accurately implementing the paper's concepts, methodology, and algorithms without logical errors. Additionally, provide a critique focusing on the completeness, accuracy, and implementation choices made in the repository relative to the methodology and algorithms described in the paper.

1: Very Poor. The repository does not correctly implement the core concepts, methodology, or algorithms from the paper. Major logical errors or missing components are present.
2: Poor. The repository attempts to implement the paper's concepts but contains significant mistakes or missing components, making the implementation incorrect.
3: Fair. Some core components and concepts are correctly implemented, but there are notable logical errors or inaccuracies in the methodology.
4: Good. The repository correctly implements the key components and methodology, with only minor inaccuracies that do not significantly affect correctness.
5: Excellent. The repository fully and accurately implements all key components, methodology, and algorithms from the paper without logical errors.

—

Evaluation Steps

1. Identify Key Aspects of the Paper: Carefully read the paper to understand its core concepts, methodology, and algorithms. Pay close attention to key aspects crucial for implementing the paper's results (e.g., specific algorithms, data preprocessing steps, evaluation protocols).

2. Examine the Code Repository: Analyze the repository to determine how well it implements the key aspects of the paper. Focus on whether the repository's core logic, algorithms, and structure align with the methodology and experiments described in the paper.

3. Identify Logical Errors and Deviations: Check for logical errors, missing steps, or deviations from the paper's methodology. Note any incorrect representations, inconsistencies, or incomplete implementations that could affect the correctness of the repository.

4. Provide a Critique: Consider the completeness and accuracy of the implementation relative to the paper's goals. You do not need to analyze minor details like logging functions, script organization, or documentation quality. Instead, concentrate on the correctness of the logic and implementation to ensure the core concepts from the paper are fully reflected in the repository. For each identified issue, write a detailed critique specifying the affected files and functions in the repository. Highlight missing or incorrectly implemented steps that impact the correctness and alignment with the paper's methodology.

5. Assess Completeness and Accuracy: Evaluate the repository for its completeness and accuracy relative to the paper's methodology. Ensure that all critical components—such as data preprocessing, core algorithms, and evaluation steps—are implemented and consistent with the paper's descriptions.

6. Assign a Score: Based on your evaluation, provide a critique and assign a correctness score from 1 to 5 for the repository, reflecting how well it implements the key aspects of the paper. Include a detailed critique in the specified JSON format.

—

Severity Level:

Each identified critique will be assigned a severity level based on its impact on the correctness of the methodology implementation.

- High: Missing or incorrect implementation of the paper's core concepts, major loss functions, or experiment components that are fundamental to reproducing the paper's methodology.
- Example: The main algorithm is missing or fundamentally incorrect. - Medium: Issues affecting training logic, data preprocessing, or other core functionalities that significantly impact performance but do not completely break the system.
- Example: Improper training loop structure, incorrect data augmentation, or missing essential components in data processing.
- Low: Errors in specific features that cause deviations from expected results but can be worked around with modifications. Any errors in the evaluation process belong to this category unless they impact the core concepts. These include minor issues like logging, error handling mechanisms, configuration settings, evaluation steps that do not alter the fundamental implementation and additional implementations not explicitly stated in the paper.
- Example: Suboptimal hyperparameter initialization, incorrect learning rate schedule, inaccuracies in evaluation metrics, using a different random seed, variations in batch processing, different weight initialization, issues in result logging or reporting, variations in evaluation dataset splits, improper error handling in non-critical steps, mismatches in secondary evaluation criteria, or additional implementation details not specified in the paper that do not interfere with core results.

---

**Prompt for model-based reference-free evaluation**

---

Example JSON format:
```json
{
    "critique_list": [
        {
            "file_name": "dataset.py",
            "func_name": "train_preprocess",
            "severity_level": "medium",
            "critique": "A critique of the target repository's file."
        },
        {
            "file_name": "metrics.py",
            "func_name": "f1_at_k",
            "severity_level": "low",
            "critique": "A critique of the target repository's file."
        }
    ],
    "score": 2
}
```

---

Sample:

Research Paper:

{{The content of the paper}}

Code Repository:

{{The generated repository, which serves as the target repository for evaluation.}}

---

Please provide a critique list for the code repository and a single numerical rating (1, 2, 3, 4, or 5) based on the quality of the sample, following the Example JSON format, without any additional commentary, formatting, or chattiness.

Figure 21: Prompt for model-based reference-free evaluation. {{}} indicate placeholders to be filled with the content described in the accompanying explanation.

**Prompt for LLM-assisted debugging**

**[System]**
You are a highly capable code assistant specializing in debugging real-world code repositories. You will be provided with:
(1) a code repository (in part or in full), and
(2) one or more execution error messages generated during the execution of the repository.

Your objective is to debug the code so that it executes successfully.
This may involve identifying the root causes of the errors, modifying faulty logic or syntax, handling missing dependencies, or making other appropriate corrections.

Guidelines:
- Provide the exact lines or file changes needed to resolve the issue.
- When necessary, suggest best practices or improvements to prevent similar issues.
- Show only the modified lines using a unified diff format:

«««< SEARCH
original line
=======
corrected line
»»»> REPLACE

- If multiple fixes are needed, provide them sequentially with clear separation.
- If external dependencies or environment setups are required (e.g., packages, versions, file paths), specify them explicitly.

Constraints:
- Do not make speculative edits without justification.
- Do not assume access to an internet connection for installation or retrieval unless explicitly stated.
- Prioritize minimal and effective fixes that preserve the original intent of the code.
- Maintain the coding style and structure used in the original repository unless refactoring is necessary for correctness.

**[User]**
### Code Repository
{{codes}}

_

### Execution Error Messages
{{execution_error_msg}}

_

## Instruction
Now, you need to debug the above code so that it runs without errors. Identify the cause of the execution error and modify the code appropriately. Your output must follow the exact format as shown in the example below.

_

## Format Example
Filename: train.py
«««< SEARCH
result = model.predict(input_data)
=======
result = model(input_data)
»»»> REPLACE

_

## Answer

Figure 22: Prompt for LLM-assisted debugging. {{}} indicate placeholders to be filled with the content described in the accompanying explanation.

---

**Prompt for verifying overall planning**

**[System]**
You will be given a research paper and an accompanying overall reproduction plan.

Your task is to rate the plan on one metric and provide a critique highlighting key differences between the plan and what the paper actually requires.

Please make sure you read and understand these instructions carefully. Keep this document open while reviewing, and refer to it as needed.

—

Evaluation Criteria

Plan–Paper Alignment (1–5): How well the overall plan aligns with the paper's methodology, experimental setup, and evaluation metrics.
1: Very Poor. The plan is largely misaligned with the paper's goals and methods, omits critical components (datasets, algorithms, or evaluation), and shows major misunderstandings.
2: Poor. The plan attempts to follow the paper but has significant gaps (key experiments missing, wrong resource assumptions, unclear success criteria).
3: Fair. The plan covers several core needs but contains notable inaccuracies or omissions (partial experiments, vague milestones, unspecified risks/assumptions).
4: Good. The plan aligns with most paper requirements, has clear milestones and resources; only minor gaps or ambiguities remain.
5: Excellent. The plan fully aligns with the paper's methodology and experiments, specifies resources and risks precisely, and defines clear, measurable success criteria.

—

Evaluation Steps

1. Extract Paper Requirements:
Identify objectives, datasets, models/algorithms, and training/evaluation protocols needed for reproduction.
2. Map Requirements to Plan:
Check whether the plan includes corresponding milestones, deliverables, resource estimates (compute, data, libraries).
3. Assess Success Criteria:
Ensure the plan defines measurable outcomes tied to the paper's metrics and variance (e.g., seeds, confidence intervals).
4. Critique:
List concrete misalignments, missing items, and unrealistic assumptions; point to specific plan sections.
5. Score:
Provide a single 1–5 rating and a detailed critique in the specified JSON format.

—

Severity Level
- High: Missing core experiments, datasets, or objectives; success criteria not tied to paper metrics.
- Medium: Incomplete milestones/resources; unclear ablations; weak risk mitigation.
- Low: Minor ambiguity in timelines, non-critical tooling choices, formatting.

—

Example JSON format
```json
{
    "critique_list": [
        {
            "plan_section": "Milestones",
            "severity_level": "high",
            "critique": "No milestone for ablation studies described in Section 4 of the paper; plan skips required variant training."
        },
        {
            "plan_section": "Resources",
            "severity_level": "medium",
            "critique": "GPU estimate does not account for 3 seeds per experiment as required by the paper's evaluation."
        }
    ],
    "score": 3
}
```

—

Sample:
Research Paper: {{Paper}}
Overall Plan: {{Plan}}

—

Please provide a critique of the weaknesses in the overall plan and a single numerical rating (1, 2, 3, 4, or 5), following the Example JSON format, without any additional commentary, formatting, or chattiness.

Figure 23: Prompt for verification in overall planning. {{}} indicate placeholders to be filled with the content described in the accompanying explanation.

---

**Prompt for refining overall planning**

**[System]**
You are an expert researcher and strategic planner with a deep understanding of experimental design and reproducibility in scientific research.

You will receive a research paper (JSON format), the original overall plan, and an evaluation critique+score of that plan.

Your task is to revise and improve the overall plan based on the critique, ensuring it fully aligns with the paper.
This plan should align precisely with the paper's methodology, experimental setup, and evaluation metrics.

—

## Instructions:
1. Fix High/Medium Issues: Correct all critical omissions and misalignments from the critique.
2. Preserve Correct Elements: Keep valid, well-aligned parts of the original plan.
3. Add Completeness: Ensure all methods, datasets, experimental setups, and evaluation metrics from the paper are included.
4. Be Clear and Structured: Present the improved plan in a roadmap format with actionable steps.
5. Prioritize Efficiency: Optimize the plan for clarity and practical implementation while ensuring fidelity to the original experiments.
6. Highlight Changes: Provide a summary of the key changes you made relative to the critique.

—

## Format Example
[CONTENT]
```
{
     "summary_of_changes": [
          "Added ablation milestones that were missing",
          "Specified required GPU hours based on experiment scale",
          "Clarified success criteria tied to accuracy and F1 metrics"
     ],
     "improved_version": "«<Revised and detailed plan here»>"
}
```
[/CONTENT]

## Notes
1. We want to reproduce the method described in the attached paper.
2. The authors did not release any official code, so we have to plan our own implementation.
3. Before writing any Python code, please outline a comprehensive plan that covers:
- Key details from the paper's **Methodology**.
- Important aspects of **Experiments**, including dataset requirements, experimental settings, hyperparameters, or evaluation metrics.
4. The plan should be as **detailed and informative** as possible to help us write the final code later.

## Requirements
- You don't need to provide the actual code yet; focus on a **thorough, clear strategy**.
- If something is unclear from the paper, mention it explicitly.

## Action
The response should give us a strong roadmap, making it easier to write the code later.
Follow the instructions for the notes and requirements, generate the output, and ensure it follows the format example.

—

## Inputs:

Research Paper:
{{Paper}}

Original Overall Plan:
{{Plan}}

Critique+Score:
{{Critique}}

---

Figure 24: Prompt for refinement in overall planning. {{}} indicate placeholders to be filled with the content described in the accompanying explanation.

---

**Prompt for verifying architecture design**

**[System]**
You will be given a research paper and an architecture design consisting of Implementation approach, File list, Data structures and interfaces(classDiagram), Program call flow(sequenceDiagram) and Anything UNCLEAR intended to complete software system design for reproducing the paper's method.

Your task is to rate the architecture on one metric and provide a critique highlighting key differences between the diagrams and what the paper requires.

Please make sure you read and understand these instructions carefully. Keep this document open while reviewing, and refer to it as needed.

—

Evaluation Criteria

Architecture–Method Fidelity (1–5): How faithfully the architecture design — Implementation approach, File list, Data structures and interfaces (classDiagram), Program call flow (sequenceDiagram) — captures the paper's components, data/control flows, responsibilities, and key interfaces.

Section-specific indicators (used to inform the 1–5 rating):

- Implementation approach
- Faithfully reflects the paper's algorithmic pipeline, major assumptions, and training/evaluation protocols.
- Mentions all required optimizer/solver variants, loss terms, constraints, and data preprocessing the paper relies on.
- Notes reproducibility-critical details (random seeds, determinism settings, hardware/precision) when the paper depends on them.

- File list
- Provides a clear, minimal, and traceable mapping from paper sections to code modules.
- Encodes strategy/factory points for ablations (optimizers, model variants, datasets) without over-coupling.
- Separates concerns (I/O vs. training vs. evaluation vs. plotting) and anticipates extensibility.

- Data structures and interfaces (classDiagram)
- Defines interfaces that match the paper's abstractions (e.g., loss components, physics constraints, evaluation metrics).
- Shows inputs/outputs and typing consistent with the paper's notation (tensor shapes, units, domains).
- Exhibits low coupling/high cohesion; substitution of components (optimizers, backends) is possible without ripple changes.

- Program call flow (sequenceDiagram)
- Preserves the paper's control flow order (training → validation → testing; optimizer switching; line-search loops).
- Includes error/edge handling the paper requires (e.g., fallback when line search fails, early stopping, tolerance checks).
- Captures logging, checkpointing, and metric computation at the times the paper specifies.

1: Very Poor. Core algorithmic components or flows from the paper are missing or fundamentally wrong; responsibilities are misplaced.
2: Poor. Attempts the paper's structure but with major omissions (e.g., missing loss path, preprocessing stage, or evaluation path) or incorrect interactions.
3: Fair. Most major components exist, but interactions are partially incorrect or responsibilities are muddled (tight coupling, unclear interfaces).
4: Good. Components and interactions largely match the paper; minor omissions or coupling issues that don't block correctness.
5: Excellent. Diagrams accurately reflect all core components and flows, with clear interfaces, appropriate separation of concerns, and traceability to paper sections.

Evaluation Steps

1. Identify Core Components:
From the paper, list modules (data loader, model submodules, loss functions, trainers, evaluators) and key messages/flows.
- Implementation approach: Extract all algorithmic steps (data preprocessing, model construction, loss formulation, optimization schedule, evaluation protocols).
- File list: Map each paper section/subsection to a candidate module; mark where ablation knobs (e.g., optimizer choice) must exist.
- Data structures and interfaces: Enumerate the required classes/structs/functions and their signatures implied by the paper (input domains, tensor shapes, units).
- Program call flow: Outline the exact order of operations (including optimizer switching, line-search/inner loops, validation checkpoints, and plotting/metric export).

2. Assess Implementation Approach:
Check whether the description faithfully covers all algorithmic components from the paper (optimizers, loss terms, constraints, PDE formulations, evaluation metrics). Verify clarity on critical reproducibility details (hyperparameters, tolerance values, data handling).

3. Assess File List:
Judge whether files are sufficient, appropriately separated, and aligned with the paper's modular structure. Look for missing utility modules (e.g., configs, logging, checkpointing) or over-coupling between responsibilities.

4. Assess Data Structures and Interfaces (Class Diagrams):
Check class responsibilities, interfaces, cohesion/coupling, extensibility, and fidelity to the paper's abstractions. Confirm that class APIs expose exactly what the paper specifies (inputs, outputs, and typing).

5. Assess Program Call Flow (Sequence Diagrams):
Verify message order, sync/async boundaries, optimizer switching, error/edge handling, and inclusion of training/evaluation/validation paths. Confirm that evaluation and logging happen at the correct cadence.

---

**Prompt for verifying architecture design**

6. Critique:
Note missing components/relations, incorrect message ordering, poor modularity, or violation of core design principles that hinder faithful implementation.

For each identified weakness, provide a JSON entry that includes:
- section: One of Implementation approach, File list, Data structures and interfaces, Program call flow
- element: The concrete element under critique
- severity_level: high, medium, or low
- critique: A concise explanation of the issue

7. Score:
Provide a single 1–5 rating that reflects overall Architecture–Method Fidelity and a detailed critique in the specified JSON format.

—

Severity Level

- High: Missing/incorrect modeling of core algorithm modules or loss/evaluation flows; sequence order contradicts the paper's method.
- Medium: Over-coupling, unclear interfaces hindering ablations or reproducibility; partial flow omissions (e.g., missing validation loop).
- Low: Naming inconsistencies, minor UML notation issues, optional utilities misplaced.

—

Example JSON format
```json
{
    "critique_list": [
        {
            "section": "Implementation approach",
            "element": "NysNewton-CG details",
            "severity_level": "high",
            "critique": "Implementation approach lacks specifics on Nyström preconditioner update frequency and PCG tolerance, which are essential for faithful reproduction."
        },
        {
            "section": "File list",
            "element": "config.py",
            "severity_level": "medium",
            "critique": "No configuration file is listed; paper requires reproducibility across experiments with tunable hyperparameters."
        },
        {
            "section": "Data structures and interfaces",
            "element": "LossFunction",
            "severity_level": "high",
            "critique": "Loss components for PDE residuals and boundary/initial conditions are not represented as separate classes; paper emphasizes modularity for ablation studies."
        },
        {
            "section": "Program call flow",
            "element": "Evaluation ordering",
            "severity_level": "medium",
            "critique": "Evaluation occurs only at the end; the paper requires intermediate validation steps for monitoring convergence."
        }
    ],
    "score": 3
}
```

—

Sample:

Research Paper:
{{Paper}}

Architecture Design:
{{ArchitectureDesign}}

—

Please provide a critique of the weaknesses in the architecture design and a single numerical rating (1, 2, 3, 4, or 5), following the Example JSON format, without any additional commentary, formatting, or chattiness.

---

Figure 25: Prompt for verification in architecture design. {{}} indicate placeholders to be filled with the content described in the accompanying explanation.

---

**Prompt for refining architecture design**

**[System]**
You are an expert researcher and strategic planner with a deep understanding of experimental design and reproducibility in scientific research.

You will receive a research paper (JSON format), the overall plan, the original architecture design and an evaluation critique+score of that architecture design.

Your task is to revise and improve the software architecture design for reproducing the paper's method based on the critique, while keeping it aligned with both the paper and the overall plan.

This software architecture design design should align precisely with the paper's methodology, experimental setup, and evaluation metrics.
Keep the architecture simple and make effective use of open-source libraries.

—

## Instructions
1. Fix High/Medium Issues: Correct missing or mis-specified modules, incorrect sequence flows, or over-coupled class designs.
2. Trace to Plan/Paper: Ensure diagrams and modules reflect the methods and milestones in the paper + overall plan.
3. Keep Correct Parts: Retain any well-designed files, class structures, or flows.
4. Improve Clarity: Rewrite class diagrams (Mermaid syntax), sequence diagrams, and file lists with complete detail.
5. Highlight Changes: Provide a summary of what was fixed or added.

—

## Format Example
[CONTENT]
{
    "summary_of_changes": [
        "Separated DataLoader and TokenizerAdapter into distinct modules",
        "Added validation loop to sequence diagram",
        "Improved interface design for Evaluation class"
    ],
    "improved_version": {
        "Implementation approach": "We will ... ,
        "File list": [
            "main.py",
            "dataset_loader.py",
            "model.py",
            "trainer.py",
            "evaluation.py"
        ],
        "Data structures and interfaces": "\nclassDiagram\n class Main \n +__init__()\n +run_experiment()\n \n class DatasetLoader \n +__init__(config: dict)\n +load_data() -> Any\n class Model \n +__init__(params: dict)\n +forward(x: Tensor) -> Tensor \n \n class Trainer \n +__init__(model: Model, data: Any)\n +train() -> None\n \n class Evaluation \n +__init__(model: Model, data: Any)\n +evaluate() -> dict\n \n Main –> DatasetLoader\n Main –> Trainer\n Main –> Evaluation\n Trainer –> Model\n",
        "Program call flow": "\nsequenceDiagram\n participant M as Main\n participant DL as DatasetLoader\n participant MD as Model\n participant TR as Trainer\n participant EV as Evaluation\n M–»DL: load_data()\n DL–»M: return dataset\n M–»MD: initialize model()\n M–»TR: train(model, dataset)\n TR–»MD: forward(x)\n MD–»TR: predictions\n TR–»M: training complete\n M–»EV: evaluate(model, dataset)\n EV–»MD: forward(x)\n MD–»EV: predictions\n EV–»M: metrics\n",
        "Anything UNCLEAR": "Need clarification on the exact dataset format and any specialized hyperparameters."
    }
}
[/CONTENT]

## Nodes: "<node>: <type> # <instruction>"
- Implementation approach: <class 'str'> # Summarize the chosen solution strategy.
- File list: typing.List[str] # Only need relative paths. ALWAYS write a main.py or app.py here.
- Data structures and interfaces: typing.Optional[str] # Use mermaid classDiagram code syntax, including classes, method(__init__ etc.) and functions with type annotations, CLEARLY MARK the RELATIONSHIPS between classes, and comply with PEP8 standards. The data structures SHOULD BE VERY DETAILED and the API should be comprehensive with a complete design.
- Program call flow: typing.Optional[str] # Use sequenceDiagram code syntax, COMPLETE and VERY DETAILED, using CLASSES AND API DEFINED ABOVE accurately, covering the CRUD AND INIT of each object, SYNTAX MUST BE CORRECT.
- Anything UNCLEAR: <class 'str'> # Mention ambiguities and ask for clarifications.

## Constraint
Format: output wrapped inside [CONTENT][/CONTENT] like the format example, nothing else.

## Action
Follow the instructions for the nodes, generate the output, and ensure it follows the format example.

—

## Inputs:
Research Paper: {{Paper}}
Overall Plan: {{Plan}}
Original Architecture Design: {{ArchitectureDesign}}
Critique+Score: {{Critique}}

Figure 26: Prompt for refinement in architecture design. {{}} indicate placeholders to be filled with the content described in the accompanying explanation.

**Prompt for verifying logic design**

**[System]**
You will be given a research paper and a logic design describing the ordered sequence of files/modules to be generated (e.g., scaffolding, filenames, module boundaries, dependency order, build/run scripts).

Your task is to rate the logic design on one metric and provide a critique highlighting key differences between the proposed generation sequence and what the paper requires.

Please make sure you read and understand these instructions carefully. Keep this document open while reviewing, and refer to it as needed.

—

Evaluation Criteria

Executable Dependency Correctness (1–5): Whether the generation order and module boundaries produce a coherent, buildable system that correctly reflects the paper's pipeline (data → train → eval) and enables required experiments.

1: Very Poor. Order/boundaries prevent a successful build or omit essential artifacts; critical dependencies unresolved.
2: Poor. Major steps are out of order or missing (e.g., metrics defined after their use); build/run impossible without substantial rework.
3: Fair. Core path is present but with notable dependency leaks or circularity; buildable with non-trivial fixes.
4: Good. Mostly correct ordering and boundaries; minor leaks or script issues that don't block execution.
5: Excellent. Fully coherent generation sequence with clear dependencies, reproducible builds, and explicit hooks for experiments/ablations.

—

Evaluation Steps

1. Identify Required Pipeline:
Identify the main stages from the paper (e.g., preprocessing, model, training, evaluation) that must be reflected in the logic design.

2. Check Ordering & Boundaries:
Confirm that module ordering respects dependencies (e.g., data before training, training before evaluation) and avoids circular imports.

3. Reproducibility Hooks:
Verify configuration, seed control, CLI/entry points, and script orchestration match the paper's eval protocol.

4. Assess Logic Analysis:
Evaluate whether the logic analysis correctly captures the roles, dependencies, and data flow of each file/module.
- Look for missing modules, unclear roles, or mismatched dependencies.
- Check whether shared knowledge/configuration is properly integrated.

5. Assess Task List:
Ensure the listed files/modules fully cover the required pipeline and appear in an executable order.
- Flag if key scripts are missing, duplicated, or misaligned with the analysis.

6. Critique:
Identify misplaced steps, missing files, circular dependencies, or non-reproducible sequencing; reference specific steps/filenames. Summarize weaknesses and mismatches. Categorize by severity (High/Medium/Low) and reference specific sections (Logic Analysis or Task list).

7. Score:
Provide a single 1–5 rating and a detailed critique in the specified JSON format.

—

Severity Level

- High: Missing generation of core modules or ordering that makes the pipeline non-executable (e.g., trainer created before model/loss interfaces exist).
- Medium: Misordered secondary components (configs, metrics, dataset splits) that significantly hinder correct runs or evaluations.
- Low: Naming inconsistencies, minor script flags, optional packaging artifacts.

---

**Prompt for verifying logic design**

—

Example JSON format
```json

Example JSON format
{
    "critique_list": [
        {
            "section": "Logic Analysis",
            "step_ref": "evaluation.py",
            "severity_level": "high",
            "critique": "Evaluator script depends on metrics that are not defined before its use; imports would fail."
        },
        {
            "section": "Logic Analysis",
            "step_ref": "trainer.py",
            "severity_level": "medium",
            "critique": "Trainer references optimizer variants, but configuration hooks are not clearly defined."
        },
        {
            "section": "Task list",
            "step_ref": "main.py",
            "severity_level": "low",
            "critique": "Entrypoint is listed but lacks mention of configuration flags or seed injection for reproducibility."
        }
    ],
    "score": 4
}
```

—

Sample:

Research Paper:
{{Paper}}

Logic Design:
{{LogicDesign}}

—

Please provide a critique of the weaknesses in the logic design and a single numerical rating (1, 2, 3, 4, or 5), following the Example JSON format, without any additional commentary, formatting, or chattiness.

Figure 27: Prompt for verification in logic design. {{}} indicate placeholders to be filled with the content described in the accompanying explanation.

---

**Prompt for refining logic design**

**[System]**
You are an expert researcher and strategic planner with a deep understanding of experimental design and reproducibility in scientific research.

You will receive a research paper (JSON format), the overall plan, the architecture design, the original logic design and an evaluation critique+score of that logic design.

Your task is to revise and improve the logic design based on the critique, ensuring it is executable, complete, and aligned with both the paper, overall plan and architecture design.

The logic design breaks down tasks according to the PRD/technical design, generates a task list, and analyzes task dependencies.

The logic design outlines a clear PRD/technical plan for reproducing the paper's methods and experiments.

The "Logic Analysis" should not only consider the dependencies between files but also provide detailed descriptions to assist in writing the code needed to reproduce the paper.

–

## Instructions
1 .Fix High/Medium Issues: Correct misordered dependencies, missing files, or incomplete API specs.
2. Ensure Executability: Verify the dependency order supports a buildable and runnable system.
3. Align with Architecture: Ensure file breakdown matches the architecture's file list and APIs.
4. Highlight Changes: Provide a clear summary of modifications.

—

## Format Example
[CONTENT]
```
{
    "summary_of_changes": [
        "Moved metric definition before evaluator script in task list",
        "Expanded API spec to include ablation toggle endpoints",
        "Clarified shared config variables for Trainer and DataLoader"
    ],
    "improved_version": {
        "Required packages": [
            "numpy==1.21.0",
            "torch==1.9.0"
        ],
        "Required Other language third-party packages": [
            "No third-party dependencies required"
        ],
        "Logic Analysis": [
            [
                "data_preprocessing.py",
                "DataPreprocessing class ........"
            ],
            [
                "trainer.py",
                "Trainer ....... "
            ],
            [
                "dataset_loader.py",
                "Handles loading and ........"
            ],
            [
                "model.py",
                "Defines the model ......."
            ],
            [
                "evaluation.py",
                "Evaluation class ........ "
            ],
            [
                "main.py",
                "Entry point ......."
            ]
        ],
        "Task list": [
            "dataset_loader.py",
            "model.py",
            "trainer.py",
            "evaluation.py",
            "main.py"
        ],
        "Full API spec": "openapi: 3.0.0 ...",
        "Shared Knowledge": "Both data_preprocessing.py and trainer.py share ........",
        "Anything UNCLEAR": "Clarification needed on recommended hardware configuration for large-scale experiments."
    }
}
```
[/CONTENT]

**Prompt for refining logic design**

## Nodes: "<node>: <type> # <instruction>"
- Required packages: typing.Optional[typing.List[str]] # Provide required third-party packages in requirements.txt format.(e.g., 'numpy==1.21.0').
- Required Other language third-party packages: typing.List[str] # List down packages required for non-Python languages. If none, specify "No third-party dependencies required".
- Logic Analysis: typing.List[typing.List[str]] # Provide a list of files with the classes/methods/functions to be implemented, including dependency analysis and imports. Include as much detailed description as possible.
- Task list: typing.List[str] # Break down the tasks into a list of filenames, prioritized based on dependency order. The task list must include the previously generated file list.
- Full API spec: <class 'str'> # Describe all APIs using OpenAPI 3.0 spec that may be used by both frontend and backend. If front-end and back-end communication is not required, leave it blank.
- Shared Knowledge: <class 'str'> # Detail any shared knowledge, like common utility functions or configuration variables.
- Anything UNCLEAR: <class 'str'> # Mention any unresolved questions or clarifications needed from the paper or project scope.

## Constraint
Format: output wrapped inside [CONTENT][/CONTENT] like the format example, nothing else.

## Action
Follow the node instructions above, generate your output accordingly, and ensure it follows the given format example."""}]

—

## Inputs:

Research Paper:
{{Paper}}

Overall Plan:
{{Plan}}

Architecture Design:
{{ArchitectureDesign}}

Original Logic Design:
{{LogicDesign}}

Critique+Score:
{{Critique}}

Figure 28: Prompt for refinement in logic design. {{}} indicate placeholders to be filled with the content described in the accompanying explanation.

**Prompt for verifying the configuration file**

**[System]**
You will be given a research paper, an accompanying overall reproduction plan, an architecture design consisting of Implementation approach, File list, Data structures and interfaces(classDiagram), Program call flow(sequenceDiagram) and Anything UNCLEAR intended to complete software system design for reproducing the paper's method, a logic design describing the ordered sequence of files/modules to be generated (e.g., scaffolding, filenames, module boundaries, dependency order, build/run scripts) and a 'config.yaml' file generated from those artifacts.

Your task is to evaluate the quality of the 'config.yaml' file in supporting reproduction of the paper's experiments.

Please make sure you read and understand these instructions carefully. Keep this document open while reviewing, and refer to it as needed.

—

Evaluation Criteria

Configuration Fidelity (1–5): The extent to which the 'config.yaml' accurately reflects the paper's methodology, datasets, hyperparameters, and evaluation settings, while aligning with the planning artifacts.

1: Very Poor. The config omits or misrepresents critical settings (datasets, hyperparameters, evaluation parameters). Cannot reproduce the experiment.
2: Poor. Includes some relevant parameters but misses major components or sets them incorrectly; partial reproducibility at best.
3: Fair. Covers most essential parameters, but with gaps, inconsistencies, or unclear defaults. Requires manual correction.
4: Good. Mostly faithful and complete, with only minor ambiguities (e.g., default values, logging frequency). Reproducible with little adjustment.
5: Excellent. Fully specifies all required datasets, preprocessing, model parameters, training/evaluation settings, and reproducibility details (seeds, logging). Ready to run directly.

—

Evaluation Steps

1. Check Paper Alignment:
Extract required datasets, hyperparameters, evaluation protocols, and reproducibility factors from the paper.

2. Compare to Planning Artifacts:
Ensure 'config.yaml' contains entries consistent with the improved overall plan, architecture design, and logic design.

3. Evaluate Completeness:
Confirm inclusion of key sections:
- Dataset paths and preprocessing details
- Model hyperparameters (hidden size, learning rate, optimizer, etc.)
- Training/evaluation settings (batch size, epochs, metrics)
- Ablation/variant toggles if experiments require them
- Random seed and reproducibility parameters

4. Check Consistency:
Verify keys, structure, and naming match the architecture and logic design (file names, module references, etc.).

5. Critique:
Identify missing or inconsistent config fields, unclear values, or misaligned defaults.

6. Score:
Assign a score from 1–5 and output your critique in JSON format.

—

Severity Levels

- High: Missing/incorrect core parameters (datasets, learning rate, epochs, evaluation metrics).
- Medium: Incomplete experiment coverage (ablations missing, evaluation variants absent, inconsistent naming).
- Low: Formatting/naming issues, minor logging/debugging configs, optional parameters not critical to reproducibility.

—

**Prompt for verifying the configuration file**

Example JSON Output
```json
{
    "critique_list": [
        {
            "config_key": "dataset.path",
            "severity_level": "high",
            "critique": "Dataset path missing; cannot locate dataset specified in the paper."
        },
        {
            "config_key": "training.seed",
            "severity_level": "medium",
            "critique": "Random seed not set, reducing reproducibility across runs."
        },
        {
            "config_key": "logging.save_dir",
            "severity_level": "low",
            "critique": "Output directory not clearly defined; may default to an unintended location."
        }
    ],
    "score": 3
}
```

—

Sample:

Research Paper:
{{Paper}}

Overall Plan:
{{Plan}}

Architecture Design:
{{ArchitectureDesign}}

Logic Design:
{{LogicDesign}}

Config File:
{{ConfigYAML}}

—

Please provide a critique of the weaknesses in the 'config.yaml' file and a single numerical rating (1, 2, 3, 4, or 5), following the Example JSON format, without any additional commentary, formatting, or chattiness.

Figure 29: Prompt for verification in the configuration file. {{}} indicate placeholders to be filled with the content described in the accompanying explanation.

---

**Prompt for refining the configuration file**

**[System]**
You are an expert ML engineer and experiment reproducibility specialist.

You will receive a research paper (JSON format), the overall plan, the architecture design, the logic design, the original 'config.yaml' file and an evaluation critique+score of that 'config.yaml' file.

Your task is to revise and improve the 'config.yaml' so that it fully supports reproducing the paper's method based on the critique, ensuring it is executable, complete, and aligned with the paper, the overall plan, architecture design and logic design.

_

## Instructions
1. Fix High/Medium Issues: Correct missing dataset paths, hyperparameters, evaluation metrics, or other essential fields noted in the critique.
2. Preserve Correct Fields: Keep all valid and well-constructed config entries intact.
3. Ensure Completeness: Add all missing sections required by the paper:
- Dataset specifications
- Model hyperparameters
- Training settings
- Evaluation metrics and protocols
- Ablation/variant toggles if required
- Reproducibility controls (random seeds, checkpoints, logging)
4. Consistency: Ensure keys and structure match the architecture and logic design (file references, module naming).
5. Clarity: Use standard YAML conventions with clear hierarchical structure.
6. Highlight Changes: Provide a summary of what was changed relative to the critique.

—

## Format Example
[CONTENT]
{
    "summary_of_changes": [
        "Added dataset.path and preprocessing parameters",
        "Specified random seed for reproducibility",
        "Aligned optimizer settings with paper (AdamW, lr=3e-5)",
        "Included ablation toggles for baseline vs. variant experiments"
    ],
    "improved_version": "«<Full corrected 'config.yaml' here»>"
}
[/CONTENT]

—

## Inputs:

Research Paper:
{{Paper}}

Overall Plan:
{{Plan}}

Architecture Design:
{{ArchitectureDesign}}

Logic Design:
{{LogicDesign}}

Original Config File:
{{ConfigYAML}}

Critique+Score:
{{Critique}}

Figure 30: Prompt for refinement in the configuration file. {{}} indicate placeholders to be filled with the content described in the accompanying explanation.

---

**Prompt for verifying the analysis file**

**[System]**
You will be given a research paper in JSON format, an overview of the plan, a design in JSON format consisting of "Implementation approach", "File list", "Data structures and interfaces", and "Program call flow", followed by a task in JSON format that includes "Required packages", "Required other language third-party packages", "Logic Analysis", and "Task list", a configuration file named "config.yaml", along with an analysis file containing comprehensive logic analysis to accurately reproduce the experiments and methodologies described in the research paper. This analysis must align precisely with the paper's methodology, experimental setup, and evaluation criteria.

Your task is to evaluate the quality of the analysis file in preparing to implement the code, and how well it aligns with the paper's methodology and the planning artifacts.

—

Evaluation Criteria

Analysis Fidelity (1–5): The extent to which the analysis file clearly and correctly specifies the responsibilities, methods, and workflows required to reproduce the paper's experiments and methodologies.

1: Very Poor. The analysis is vague, missing core methods, or contradicts the paper/planning artifacts. Cannot guide implementation.

2: Poor. Contains partial method outlines but omits critical functionality (e.g., evaluation loop, config integration). Would mislead implementation.

3: Fair. Covers most key components, but lacks detail in method responsibilities or misorders dependencies. Usable with significant manual fixing.

4: Good. Clear and structured, with most responsibilities correctly assigned and aligned with the paper. Only minor omissions or ambiguities.

5: Excellent. Complete, precise, and executable outline. All methods and workflows are included, responsibilities are clear, and it directly enables faithful code implementation.

—

Evaluation Steps

1. Check Paper Alignment:
Verify that classes and methods in the analysis match the paper's methodology (datasets, training, evaluation, metrics).

2. Check Plan Consistency:
Ensure responsibilities match the overall plan, architecture design, logic design (naming, APIs, flows), and configuration file. The analysis file must follow "Data structures and interfaces" and do not use public member functions that do not exist in your design. Also, always reference settings from the config.yaml file. Do not invent or assume any values—only use configurations explicitly provided.

3. Check Completeness:
Confirm that the analysis covers the file's role in the overall experiment pipeline, including relevant aspects such as:
- Core orchestration or entry-point logic (if the file defines workflows, execution flow, or script-level commands)
- Dataset handling (loading, preprocessing, augmentation, batching)
- Model initialization (architectures, weights, optimizers, schedulers)
- Training loop and checkpoints (iteration structure, loss computation, saving/restoring models)
- Evaluation loop and metrics (validation/testing, performance measurement)
- Configuration and logging integration (hyperparameters, experiment tracking, reproducibility)
- Utility methods and shared functionality (helper functions, abstractions, or cross-module dependencies that support multiple parts of the codebase)

4. Check Clarity:
Evaluate whether the method steps are sufficiently detailed and logically ordered to be implemented directly. The analysis should present a logical, well-organized, and actionable format that is easy to follow and apply.

5. Critique:
List missing steps, unclear method responsibilities, or inconsistencies with prior planning artifacts.

6. Score:
Assign a single 1–5 score and provide critiques in JSON format.

—

Severity Levels

- High: Missing orchestration, dataset/model/training/eval flows, or analysis contradicts paper's methods.
- Medium: Incomplete detail on dependencies, unclear method responsibilities, or inconsistent naming compared to planning artifacts.
- Low: Minor formatting, naming clarity, or logging/debugging omissions.

—

**Prompt for verifying the analysis file**

Example JSON Output
```json
{
    "critique_list": [
        {
            "section": "conduct_training",
            "severity_level": "high",
            "critique": "Training method does not mention checkpoint saving/loading, which is required for reproducibility in the paper."
        },
        {
            "section": "initialize_model",
            "severity_level": "medium",
            "critique": "Model initialization does not specify tokenizer or embedding layer setup as described in the architecture design."
        },
        {
            "section": "setup_logging",
            "severity_level": "low",
            "critique": "Logging configuration is not aligned with the shared logging utilities outlined in the logic design."
        }
    ],
    "score": 3
}
```

—

Sample:

Research Paper:
{{Paper}}

Overall Plan:
{{Plan}}

Architecture Design:
{{ArchitectureDesign}}

Logic Design:
{{LogicDesign}}

Config File:
{{ConfigYAML}}

Analysis File:
{{AnalysisFile}}

—

Please provide a critique of the weaknesses in the analysis file and a single numerical rating (1, 2, 3, 4, or 5), following the Example JSON format, without any additional commentary, formatting, or chattiness.

Figure 31: Prompt for verification in the analysis file. {{}} indicate placeholders to be filled with the content described in the accompanying explanation.

---

**Prompt for refining the analysis file**

**[System]**
You are an expert researcher, strategic analyzer and software engineer with a deep understanding of experimental design and reproducibility in scientific research.

You will receive a research paper (JSON format), the overall plan, the architecture design, the logic design, a configuration file named 'config.yaml', the original analysis file and an evaluation critique+score of the analysis file.

Your task is to revise and improve the analysis file based on the critique and ensure that it aligns with the research paper, the overall plan, the architecture design, the logic design, and the configuration file.

This analysis must align precisely with the paper's methodology, experimental setup, and evaluation criteria.

The analysis must be conducted with absolute fidelity to the paper's methodology, ensuring that every element—from datasets and model configurations to hyperparameters and experimental setups—mirrors the original specification without deviation or assumption.

The presentation should be clear, logically structured, and actionable, allowing others to replicate or extend the work with ease.

The established architecture design of "Data structures and interfaces" must remain intact; under no circumstances should this design be altered, nor should functions outside those explicitly defined be introduced.

Every reference to experimental settings must be drawn directly from the config.yaml file, with no invented or inferred values permitted.

_

## Instructions
1. Fix High/Medium Issues: Resolve all critical omissions and contradictions noted in the critique (e.g., missing training/eval loops, incorrect method responsibilities, ignoring config.yaml values).

2. Preserve Correct Elements: Keep all valid, accurate, and consistent sections of the original analysis file.

3. Ensure Completeness: The improved analysis must cover the file's role in the experiment pipeline, including relevant aspects such as:
- Orchestration/entry-point logic
- Dataset handling
- Model initialization
- Training loop & checkpoints
- Evaluation loop & metrics
- Config and logging integration
- Utility methods and shared knowledge

4. Consistency:
- Match class/method names and APIs to those in the architecture design.
- Respect dependencies and order defined in the logic design.
- Always reference hyperparameters/settings from config.yaml — never invent values.

5. Clarity: Write method responsibilities and steps in a clear, logically ordered, and directly implementable way.

6. Highlight Changes: Provide a summary of the key changes relative to the critique.

—

## Format Example
[CONTENT]
{
    "summary_of_changes": "Added checkpoint saving/loading steps in training, aligned model initialization with architecture design and config.yaml, clarified logging setup to use shared utilities, expanded evaluation flow with metrics defined in config.yaml.",
    "improved_version": "«<Full improved analysis file here»>"
}
[/CONTENT]

—

## Inputs:

Research Paper: {{Paper}}

Overall Plan: {{Plan}}

Architecture Design: {{ArchitectureDesign}}

Logic Design: {{LogicDesign}}

Config File: {{ConfigYAML}}

Original Analysis File: {{AnalysisFile}}

Critique+Score: {{Critique}}

---

Figure 32: Prompt for refinement in the analysis file. {{}} indicate placeholders to be filled with the content described in the accompanying explanation.

---

**Prompt for addressing the environment in the coding stage**

**[System]**
You are an expert researcher and software engineer with a deep understanding of experimental design and reproducibility in scientific research. You will receive configuration file named "config.yaml", and implmented code repository. Your task is to write a Bash script that can run the given repository from scratch. The script should create and activate the required environment, install all dependencies, and include the commands needed to execute the main file or entry point. Make sure the script is self-contained and can be executed without any manual setup.

Write code with triple quoto.

**[User]**
# Context

## Configuration file
```yaml
{The content of the configuration file}
```

___

## Code Files
{The content of the generated repository}

___

# Format example
## Code: reproduce.sh
```bash
## reproduce.sh
...
```

___

# Instruction
Based on the code files, follow "Format example", write the code.

We have {done_file_lst}.
Next, you must write only the "reproduce.sh".

## Code: reproduce.sh

---

Figure 33: Prompt for addressing the environment in the coding stage. {{}} indicate placeholders to be filled with the content described in the accompanying explanation.

# E    EXAMPLES OUTPUT OF THE PLANNING PHASE

Below is a comprehensive roadmap that details the steps you need to follow to faithfully reproduce the experiments and methodologies described in "Attention Is All You Need." This plan outlines all the key components—from data preprocessing to the model architecture, training regimen, and evaluation metrics—providing a solid strategy before writing any Python code. You can later translate each section into modular code (e.g., in TensorFlow or PyTorch) while referring back to the detailed hyperparameters and settings mentioned in the paper.

---

1. OVERVIEW & KEY DETAILS

• The paper introduces the Transformer architecture—a purely attention-based encoder–decoder model that completely drops recurrence and convolution.
• There are two main experimental tracks:
 – Machine translation on WMT 2014 English–German (≈4.5M sentence pairs) and English–French (≈36M sentence pairs) datasets.
 – English constituency parsing on the WSJ portion of the Penn Treebank (with both small-data and semi-supervised settings).
• The model consists of stacked encoder and decoder layers (base configuration uses 6 layers each, "big" models use larger dimensions) with multi-head (8 heads) scaled dot-product attention, point-wise feed-forward networks, positional encodings (using sinusoids), residual connections, and layer normalization.
• The training uses the Adam optimizer with warmup and a custom learning rate schedule, extensive token-level batching, label smoothing ($\varepsilon = 0.1$), and dropout (typically 0.1, though one variant used 0.3 for English–French "big" models).

---

2. DATA PREPARATION & EXPERIMENTAL SETUP

A. Machine Translation Datasets:
 • English–German:
  – Data: WMT 2014 EN-DE dataset (≈4.5M sentence pairs).
  – Preprocessing:
   ○ Tokenize sentences.
   ○ Apply byte-pair encoding (BPE) with a shared source–target vocabulary of ~37,000 tokens.
   ○ Group/batch sentence pairs approximately by sequence length; target batch size ~25,000 tokens each for source and target.
 • English–French:
  – Data: WMT 2014 EN-FR dataset (≈36M sentence pairs).
  – Preprocessing:
   ○ Tokenize sentences.
   ○ Use word-piece segmentation (vocabulary size of 32,000 tokens).
   ○ Similar batching strategy based on approximate token counts.

B. Constituency Parsing:
 • Data: WSJ portion of the Penn Treebank.
  – For "WSJ only" experiments, ~40K training sentences (use a vocabulary of 16K tokens).
  – For semi-supervised settings, include additional high-confidence corpora (~17M sentences; use a vocabulary of 32K tokens).
 • Preprocess sentences to generate input representations and the corresponding target trees (likely represented as linearized bracketed trees).
 • For decoding, set maximum output length to "input length + 300."

Note:
 – Some specifics of tokenization and BPE/word-piece toolchain details are not spelled out in the paper; choose established open-source tools (e.g., SentencePiece, Subword-NMT) and document your choices.

---

3. MODEL ARCHITECTURE & MODULES

A. Embedding & Positional Encoding:
 • Embedding Layers:
  – Learn token embeddings for both the encoder (source) and decoder (target).
  – Use a common weight matrix (shared between input embeddings and the pre-softmax linear layer) with scaling factor $\sqrt{d\_model}$.
 • Positional Encoding (sinusoidal):
  – Implement positional encoding with:
    PE(pos, 2i) = sin(pos / 10000^(2i/d_model))
    PE(pos, 2i+1) = cos(pos / 10000^(2i/d_model))
  – This gives each embedding the same dimension as d_model (512 for the base model).

B. Encoder:
 • Consists of N = 6 identical layers.
 • Each layer contains:
  1. Multi-head self-attention sub-layer:
   – Use 8 attention heads.
   – For each head: d_k = d_v = d_model / 8 = 64.
   – Compute scaled dot-product attention (dot product divided by $\sqrt{d\_k}$).
  2. Position-wise fully connected feed-forward network:
   – Two linear transformations with a ReLU activation in between.
   – Dimensionality: Input/Output = 512; inner-layer (d_ff) = 2048.
 • Each sub-layer is wrapped with a residual connection and followed by layer normalization.

C. Decoder:
 • Also consists of N = 6 identical layers, with one additional sub-layer compared to the encoder.
 • Each layer includes:
  1. Masked multi-head self-attention sub-layer (mask future tokens to preserve auto-regressive property).
  2. Multi-head cross-attention sub-layer over the encoder output (encoder–decoder attention).
  3. Position-wise feed-forward network (as in encoder).
 • Residual connections and layer normalization are applied in each sub-layer.
 • The output embedding is offset by one position to ensure proper autoregressive modeling.

D. Attention Mechanism Details:
 • Scaled Dot-Product Attention:
  – Given query Q, keys K, and values V, compute attention weights as: softmax(QK^T / $\sqrt{d\_k}$) times V.
 • Multi-head attention:
  – Linearly project Q, K, V into h (8) subspaces using learned matrices (W_Q, W_K, W_V).
  – Apply scaled dot-product attention in each head.
  – Concatenate the outputs and project with W_O.
  – This allows the model to attend to different representation subspaces simultaneously.
 • Note: The paper describes slight modifications (masking in decoder, possible variations in head counts/dimensions); be sure to parameterize these so you can later experiment with variants.

E. Additional Components:
 • Softmax over shared vocabulary weights for prediction.
 • Dropout:
  – Apply dropout (P_drop = 0.1 for base models, and 0.1 or 0.3 variant noted for EN-FR "big" model) on:
   ○ Output of every sub-layer (after attention and feed-forward networks) before adding the residual connection.
   ○ Sum of embedding and positional encoding.
 • Label Smoothing:
  – Use ε_ls = 0.1 during training to soften the targets.

---

4. TRAINING PROCEDURE & HYPERPARAMETERS

A. Optimizer & Learning Rate Schedule:
 • Use the Adam optimizer with parameters:
  – β1 = 0.9, β2 = 0.98, ε = 1e-9.
 • Implement the learning rate schedule:
  – Learning rate = d_model^(-0.5) * min(step^(-0.5), step * warmup_steps^(-1.5))
  – Use warmup_steps = 4000.
  – This schedule increases the learning rate linearly in the warmup phase and then decays as an inverse square root of the step number.

B. Batch Formation:
 • Form batches by grouping sentence pairs with approximately 25,000 tokens in both source and target sequences.
 • Ensure that batches are dynamic and based on token counts rather than fixed sentence counts to use GPU memory efficiently.

Figure 34: Artifact from Step 1.1 (Overall Plan) in the planning stage of PaperCoder, generated during repository construction for the Transformer (Vaswani et al., 2017) (1/2).

C. Training Schedules (per experiment):
    1. Transformer Base Model:
        – Train for 100,000 steps (~12 hours on 8 P100 GPUs).
        – Checkpoint averaging: Average the last 5 checkpoints (recorded at 10-minute intervals).
    2. Transformer Big Model:
        – Configurations: Increase model dimensions (e.g., d_model = 1024, d_ff = 4096, possibly adjust the number of heads and dropout as specified in Table 3 "big" configuration).
        – Train for 300,000 steps (~3.5 days on 8 P100 GPUs).
        – Average the last 20 checkpoints.
    3. English Constituency Parsing:
        – Use a 4-layer Transformer with d_model = 1024.
        – For WSJ only: vocabulary of 16K tokens.
        – For the semi-supervised setting: vocabulary of 32K tokens.
        – Keep most hyperparameters the same as the translation base model.
        – Beam search adjustments: Use beam size = 21, length penalty $\alpha$ = 0.3, and set maximum output length as (input length + 300).

D. Regularization & Other Training Details:
    • Apply dropout at all prescribed points.
    • Use label smoothing consistently.
    • (Ambiguity Note:) The paper does not specify if gradient clipping was used—this is an implementation decision. Document your choice if you add gradient clipping.
    • Random seeds and consistent data shuffling should be set for reproducibility.

---

5. DECODING & EVALUATION

A. Decoding for Translation Tasks:
    • Use beam search with:
        – Beam size of 4.
        – Length penalty $\alpha$ = 0.6.
        – Maximum output length set to "input length + 50," with early termination if possible.
    • Evaluate BLEU scores on:
        – WMT 2013 newstest for development.
        – WMT 2014 newstest for final test performance.
    • Consider using standard BLEU evaluation tools (e.g., SacreBLEU) to ensure comparability with reported numbers.

B. Evaluation for Constituency Parsing:
    • Use the F1 score (WSJ 23 F1 is reported) as the primary metric.
    • Evaluate on the appropriate WSJ sections as outlined in the paper.
    • Compare against previous state-of-the-art results and baselines.

---

6. IMPLEMENTATION ORGANIZATION & ROADMAP

Step 1. Environment & Setup:
    • Choose your deep learning framework (e.g., TensorFlow or PyTorch). Document framework version and GPU requirements (8 NVIDIA P100 GPUs assumed).
    • Set up data pipelines for reading, tokenizing, and batching the datasets.
    • Prepare preprocessing scripts for BPE or word-piece segmentation.

Step 2. Module-wise Implementation:
    • Write modular code for:
        – Embeddings and positional encodings.
        – Scaled dot-product attention.
        – Multi-head attention module (with masking functionality for decoder).
        – Position-wise feed-forward layers.
        – Encoder layer: Assemble self-attention and feed-forward blocks with residual connections and layer normalization.
        – Decoder layer: Build a 3 sub-layer block (masked self-attention, encoder–decoder attention, and feed-forward) as described.
        – Output projection and softmax layers.
    • Ensure weight sharing between input embeddings and the pre-softmax projection.

Step 3. Training Loop & Learning Rate Schedule:
    • Integrate the optimizer and learning rate schedule (include warmup).
    • Implement batching by token count.
    • Include dropout and label smoothing in the loss calculation.
    • Set logging and checkpointing (including saving every 10 minutes or equivalent step intervals).

Step 4. Decoding & Inference:
    • Code the beam search decoding with parameters that can be adjusted (beam size, length penalty, max output length).
    • Implement early stopping during decoding as suggested.

Step 5. Checkpoint Averaging:
    • Write a script to average parameters from the last 5 (or 20 for big models) checkpoints for final evaluation.

Step 6. Experiment Tracking & Reproducibility:
    • Log training loss, validation BLEU/perplexity, and decoding outputs.
    • Record hyperparameters and random seeds.
    • Compare intermediate results (e.g., BLEU scores on dev set) to those reported in the paper.
    • Note any divergences and investigate potential causes (batching differences, tokenization differences, etc.).

Step 7. Constituency Parsing Experiments:
    • Adapt the Transformer implementation to work for constituency parsing by:
        – Adjusting the number of layers and hidden dimensions (4 layers, d_model = 1024).
        – Changing vocabulary sizes accordingly.
        – Tuning beam search parameters (beam size 21, length penalty 0.3, max output = input+300).
    • Evaluate the parsing F1 and compare with the numbers reported in the paper.

---

7. AMBIGUITIES & UNCERTAIN DETAILS

• Tokenization & BPE/Word-piece: The exact BPE merge operations and vocabulary processing details are not fully described. Choose standard tools and report your settings.
• Data Shuffling & Batching: The "approximate 25,000 tokens per direction" rule is somewhat open-ended. You may need to experiment to match the GPU memory usage described.
• Certain hyperparameter details for the "big" model variants (e.g., specific dropout differences for English–French) or additional tuning for parsing may require minor experimentation.
• Gradient clipping is not mentioned explicitly in the paper; decide if you want to add it for stabilization and document your choice.
• In constituency parsing, only a "small number of experiments" were done to choose dropout and beam size. Begin by using the translation hyperparameters and then adjust based on preliminary performance.
• Although the paper uses checkpoint averaging, details such as frequency of checkpoint saves (every 10 minutes) need to be mapped to your own compute environment.

---

8. FINAL REMARKS

• Document every step and ensure that any deviation from the paper's settings is fully recorded.
• Once all modules and training loops are implemented, run pilot experiments (e.g., on a subset of the data) to verify that your implementation (especially the attention mechanisms and learning rate schedule) behaves as expected.
• Compare intermediate logs (perplexity on training batches, dev set BLEU) with the paper's reported numbers, and refine your implementation if significant differences persist.

This roadmap should provide you with a solid foundation for translating the paper's methodology into robust code. Each block and step is directly traced back to the paper's sections (from model architecture to training and evaluation), ensuring that you preserve fidelity to the original experiment design while allowing room for minor adjustments if necessary.

Figure 35: Artifact from Step 1.1 (Overall Plan) in the planning stage of PaperCoder, generated during repository construction for the Transformer (Vaswani et al., 2017) (2/2).

```
----------------------------------------
[Implementation approach]
We will implement a simple, modular Transformer system in Python using open-source libraries such as PyTorch for model implementation and training, TorchText (or similar) for dataset management, and
SentencePiece/Subword-NMT for tokenization. Our system will separate concerns into distinct modules: dataset loading (preprocessing and batching), model definition (implementing Transformer
encoder-decoder with multi-head attention, positional encodings, feed-forward networks, residual connections, and layer normalization), training (with the Adam optimizer and a custom learning rate schedule
including warmup), and evaluation (using BLEU for translation and F1 for parsing). This design remains simple while leveraging established libraries to keep the code concise and maintainable.

----------------------------------------
[File list]
- main.py
- dataset_loader.py
- model.py
- trainer.py
- evaluation.py

----------------------------------------
[Data structures and interfaces]
classDiagram
    class Main {
        +__init__(config: dict)
        +run_experiment() -> None
    }
    class DatasetLoader {
        +__init__(config: dict)
        +load_data() -> dict
    }
    class TransformerModel {
        +__init__(params: dict)
        +forward(src: Tensor, tgt: Tensor) -> Tensor
    }
    class Encoder {
        +__init__(params: dict)
        +forward(x: Tensor) -> Tensor
    }
    class Decoder {
        +__init__(params: dict)
        +forward(x: Tensor, encoder_output: Tensor) -> Tensor
    }
    class MultiHeadAttention {
        +__init__(d_model: int, num_heads: int)
        +forward(query: Tensor, key: Tensor, value: Tensor, mask: Optional[Tensor]) -> Tensor
    }
    class FeedForward {
        +__init__(d_model: int, d_ff: int)
        +forward(x: Tensor) -> Tensor
    }
    class PositionalEncoding {
        +__init__(d_model: int, max_len: int)
        +forward(x: Tensor) -> Tensor
    }
    class Trainer {
        +__init__(model: TransformerModel, data: dict, config: dict)
        +train() -> None
        +save_checkpoint(path: str) -> None
        +load_checkpoint(path: str) -> None
    }
    class Evaluation {
        +__init__(model: TransformerModel, data: dict, config: dict)
        +evaluate() -> dict
    }
    Main --> DatasetLoader
    Main --> TransformerModel
    Main --> Trainer
    Main --> Evaluation
    TransformerModel o-- Encoder
    TransformerModel o-- Decoder
    Encoder --> MultiHeadAttention
    Decoder --> MultiHeadAttention
    Encoder --> FeedForward
    Decoder --> FeedForward
    Encoder --> PositionalEncoding
    Decoder --> PositionalEncoding
    Trainer --> TransformerModel
    Evaluation --> TransformerModel

----------------------------------------
[Program call flow]
sequenceDiagram
    participant M as Main
    participant DL as DatasetLoader
    participant TM as TransformerModel
    participant TR as Trainer
    participant EV as Evaluation

    M->>DL: __init__(config)
    M->>DL: load_data()
    DL-->>M: return {data: translation/parsing dataset}
    M->>TM: __init__(params)
    M->>TR: __init__(model=TM, data, config)
    M->>EV: __init__(model=TM, data, config)
    M->>TR: train()
    TR->>TM: forward(src, tgt) [during each batch]
    TM-->>TR: model predictions
    TR->>TR: save_checkpoint() periodically
    TR-->>M: training complete
    M->>EV: evaluate()
    EV->>TM: forward(src, tgt) [for inference decoding]
    TM-->>EV: predictions
    EV-->>M: evaluation metrics (BLEU / F1 scores)

----------------------------------------
[Anything UNCLEAR]
Clarification is needed on the exact dataset formats for both machine translation and constituency parsing (e.g., file structure, tokenization output). Additionally, details on whether to use gradient clipping,
precise checkpoint saving frequency, and any specialized hyperparameters (especially for the 'big' model configuration) would be helpful.
```

Figure 36: Artifact from Step 1.2 (Architecture Design) in the planning stage of PaperCoder, generated during repository construction for the Transformer (Vaswani et al., 2017).

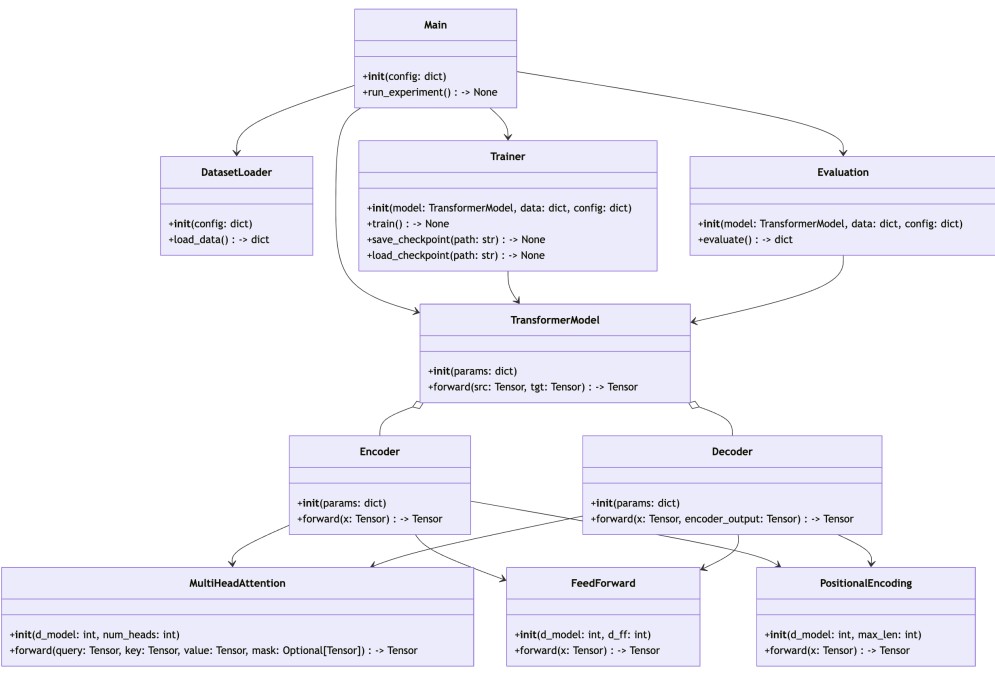

Figure 37: Class diagram visualization generated from Step 1.2 (Architecture Design) in the planning stage of PaperCoder, generated during repository construction for the Transformer (Vaswani et al., 2017).

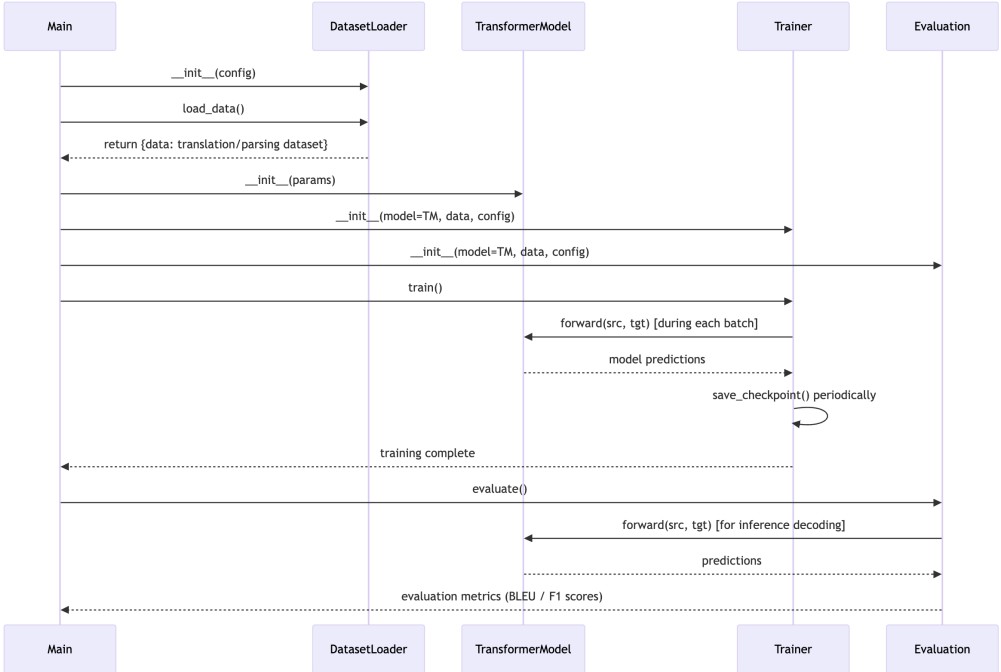

Figure 38: Program call flow visualization generated from Step 1.2 (Architecture Design) in the planning stage of PaperCoder, generated during repository construction for the Transformer (Vaswani et al., 2017).

```
----------------------------------------
[Required packages]
- numpy==1.21.0
- torch==1.9.0
- sentencepiece==0.1.96
- torchtext==0.10.0
- sacremoses==0.0.45

----------------------------------------
[Required Other language third-party packages]
- No third-party dependencies required

----------------------------------------
[Logic Analysis]
- ['dataset_loader.py', 'DatasetLoader class handles loading raw datasets for both machine translation and constituency parsing, including file I/O, tokenization (via SentencePiece or BPE), vocabulary
creation, and dynamic batching based on token counts. It imports torchtext and tokenization libraries, and provides outputs in a format consumable by the Trainer and Evaluation modules.']
- ['model.py', "Defines the TransformerModel class composed of Encoder and Decoder modules. Inside, implement subcomponents like Encoder, Decoder, MultiHeadAttention, FeedForward, and
PositionalEncoding. This module leverages PyTorch's nn.Module, and reflects the design details from the paper with attention mechanisms, residual connections, layer normalization, and shared embedding
layers. Dependencies include torch.nn and relevant utility functions."]
- ['trainer.py', 'Implements the Trainer class which sets up the training loop using the TransformerModel. The Trainer handles optimizer setup (Adam with custom hyperparameters and a learning rate schedule
with warmup), loss computation with label smoothing, batching, checkpoint saving/loading, and logging training statistics. It depends on both model.py and dataset_loader.py.']
- ['evaluation.py', 'Contains the Evaluation class responsible for running inference on the trained model, implementing beam search decoding, and computing evaluation metrics (BLEU for translation tasks
and F1 scores for constituency parsing). It may use external libraries like sacreBLEU for BLEU score calculation. This module depends on model.py and dataset_loader.py for data and model interfacing.']
- ['main.py', 'Acts as the entry point to the system. It reads the configuration (possibly from a config file or command-line arguments), instantiates DatasetLoader to load data, creates the TransformerModel,
and then instantiates the Trainer and Evaluation classes. The control flow manages calling the training loop followed by evaluation and metric reporting. It ties all modules together and handles overall
experiment orchestration.']

----------------------------------------
[Task list]
- dataset_loader.py
- model.py
- trainer.py
- evaluation.py
- main.py

----------------------------------------
[Full API spec]

----------------------------------------
[Shared Knowledge]
Shared utilities include configuration management, logging functions, and checkpoint management code that are common between dataset_loader.py and trainer.py. Hyperparameters and model constants
(e.g., d_model, num_heads, d_ff, warmup_steps, dropout rates) should be centralized in a common configuration module or dictionary to ensure consistency across modules.

----------------------------------------
[Anything UNCLEAR]
Clarification is needed on the recommended hardware configuration for large-scale experiments, the exact format of the datasets (file structure, tokenization details, data splits), the frequency of checkpoint
saving, and whether gradient clipping should be incorporated in the training loop.
```

Figure 39: Artifact from Step 1.3 (Logic Design) in the planning stage of PaperCoder, generated during repository construction for the Transformer (Vaswani et al., 2017).

```yaml
## config.yaml
training:
  optimizer: "Adam"
  beta1: 0.9
  beta2: 0.98
  epsilon: 1e-9
  warmup_steps: 4000
  total_steps: 100000   # Base model: 100K training steps (~12 hours on 8 P100 GPUs)
  learning_rate_schedule: "d_model^-0.5 * min(step^-0.5, step * warmup_steps^-1.5)"
  batch_tokens:
    source: 25000
    target: 25000
  dropout: 0.1
  label_smoothing: 0.1

model:
  num_layers: 6
  d_model: 512
  d_ff: 2048
  num_heads: 8
  d_k: 64
  d_v: 64
  shared_embedding: true

data:
  dataset: "WMT 2014 English-to-German"
  encoding: "byte-pair encoding"
  vocabulary_size: 37000

evaluation:
  beam_size: 4
  length_penalty: 0.6
  max_output_length: "input_length + 50"
```

Figure 40: Artifact from Step 1.4 (Configuration File) in the planning stage of PaperCoder, generated during repository construction for the Transformer (Vaswani et al., 2017).

Table 20: List of ICLR 2024 papers used in our Paper2CodeBench benchmark. We evaluate each paper using the model-based, reference-free setting, with `gpt-4o-2024-11-20` as the evaluation model.

| Paper | Source | Score |
|---|---|---|
| Generative Judge for Evaluating Alignment | Poster | 4 |
| Distributional Preference Learning: Understanding and Accounting for Hidden Context in RLHF | Poster | 4 |
| Inherently Interpretable Time Series Classification via Multiple Instance Learning | Oral | 3.9 |
| iTransformer: Inverted Transformers Are Effective for Time Series Forecasting | Oral | 3.9 |
| Tell Your Model Where to Attend: Post-hoc Attention Steering for LLMs | Poster | 3.9 |
| Knowledge Distillation Based on Transformed Teacher Matching | Poster | 3.9 |
| Meaning Representations from Trajectories in Autoregressive Models | Poster | 3.8 |
| A Simple Interpretable Transformer for Fine-Grained Image Classification and Analysis | Poster | 3.8 |
| VDC: Versatile Data Cleanser based on Visual-Linguistic Inconsistency by Multimodal Large Language Models | Poster | 3.8 |
| Vocos: Closing the gap between time-domain and Fourier-based neural vocoders for high-quality audio synthesis | Poster | 3.8 |
| SliceGPT: Compress Large Language Models by Deleting Rows and Columns | Poster | 3.8 |
| Beyond Accuracy: Evaluating Self-Consistency of Code Large Language Models with IdentityChain | Poster | 3.8 |
| Guiding Masked Representation Learning to Capture Spatio-Temporal Relationship of Electrocardiogram | Poster | 3.8 |
| Social Reward: Evaluating and Enhancing Generative AI through Million-User Feedback from an Online Creative Community | Oral | 3.7 |
| Language Model Detectors Are Easily Optimized Against | Poster | 3.7 |
| Improving protein optimization with smoothed fitness landscapes | Poster | 3.7 |
| SparseFormer: Sparse Visual Recognition via Limited Latent Tokens | Poster | 3.7 |
| AutoVP: An Automated Visual Prompting Framework and Benchmark | Poster | 3.7 |
| Hierarchical Context Merging: Better Long Context Understanding for Pretrained LLMs | Poster | 3.7 |
| SEABO: A Simple Search-Based Method for Offline Imitation Learning | Poster | 3.7 |
| OpenChat: Advancing Open-source Language Models with Mixed-Quality Data | Poster | 3.7 |
| Rethinking The Uniformity Metric in Self-Supervised Learning | Poster | 3.7 |
| VONet: Unsupervised Video Object Learning With Parallel U-Net Attention and Object-wise Sequential VAE | Poster | 3.6 |
| Efficient Backpropagation with Variance-Controlled Adaptive Sampling | Poster | 3.6 |
| Structuring Representation Geometry with Rotationally Equivariant Contrastive Learning | Poster | 3.6 |
| ControlVideo: Training-free Controllable Text-to-Video Generation | Poster | 3.6 |
| Context-Aware Meta-Learning | Poster | 3.6 |
| RECOMBINER: Robust and Enhanced Compression with Bayesian Implicit Neural Representations | Poster | 3.6 |
| Peering Through Preferences: Unraveling Feedback Acquisition for Aligning Large Language Models | Poster | 3.6 |
| Modulate Your Spectrum in Self-Supervised Learning | Poster | 3.6 |

Table 21: List of ICML 2024 papers used in our Paper2CodeBench benchmark. We evaluate each paper using the model-based, reference-free setting, with `gpt-4o-2024-11-20` as the evaluation model.

| Paper | Source | Score |
|---|---|---|
| SAMformer: Unlocking the Potential of Transformers in Time Series Forecasting with Sharpness-Aware Minimization and Channel-Wise Attention | Oral | 4 |
| Autoformalizing Euclidean Geometry | Poster | 4 |
| Recurrent Distance Filtering for Graph Representation Learning | Poster | 4 |
| CosPGD: an efficient white-box adversarial attack for pixel-wise prediction tasks | Poster | 3.9 |
| Token-level Direct Preference Optimization | Poster | 3.9 |
| BayOTIDE: Bayesian Online Multivariate Time Series Imputation with Functional Decomposition | Oral | 3.8 |
| CurBench: Curriculum Learning Benchmark | Poster | 3.8 |
| Exploring the Low-Pass Filtering Behavior in Image Super-Resolution | Poster | 3.8 |
| Towards Efficient Exact Optimization of Language Model Alignment | Poster | 3.7 |
| On the Effectiveness of Supervision in Asymmetric Non-Contrastive Learning | Poster | 3.7 |
| Drug Discovery with Dynamic Goal-aware Fragments | Poster | 3.7 |
| Fool Your (Vision and) Language Model With Embarrassingly Simple Permutations | Poster | 3.7 |
| Image Restoration Through Generalized Ornstein-Uhlenbeck Bridge | Poster | 3.7 |
| Timer: Generative Pre-trained Transformers Are Large Time Series Models | Poster | 3.7 |
| Mitigating Oversmoothing Through Reverse Process of GNNs for Heterophilic Graphs | Poster | 3.7 |
| Scribble-Supervised Semantic Segmentation with Prototype-based Feature Augmentation | Poster | 3.7 |
| ConvNet vs Transformer, Supervised vs CLIP: Beyond ImageNet Accuracy | Poster | 3.7 |
| CLIF: Complementary Leaky Integrate-and-Fire Neuron for Spiking Neural Networks | Oral | 3.6 |
| FiT: Flexible Vision Transformer for Diffusion Model | Oral | 3.6 |
| Decomposing Uncertainty for Large Language Models through Input Clarification Ensembling | Oral | 3.6 |
| SparseTSF: Modeling Long-term Time Series Forecasting with *1k* Parameters | Oral | 3.6 |
| Sample-specific Masks for Visual Reprogramming-based Prompting | Oral | 3.6 |
| Boundary Exploration for Bayesian Optimization With Unknown Physical Constraints | Poster | 3.6 |
| Listwise Reward Estimation for Offline Preference-based Reinforcement Learning | Poster | 3.6 |
| Graph Distillation with Eigenbasis Matching | Poster | 3.6 |
| Temporal Spiking Neural Networks with Synaptic Delay for Graph Reasoning | Poster | 3.6 |
| Position: Quo Vadis, Unsupervised Time Series Anomaly Detection? | Poster | 3.6 |
| Neural SPH: Improved Neural Modeling of Lagrangian Fluid Dynamics | Poster | 3.6 |
| Self-Play Fine-Tuning Converts Weak Language Models to Strong Language Models | Poster | 3.6 |
| Unveiling and Harnessing Hidden Attention Sinks: Enhancing Large Language Models without Training through Attention Calibration | Poster | 3.6 |

Table 22: List of NeurIPS 2024 papers used in our Paper2CodeBench benchmark. We evaluate each paper using the model-based, reference-free setting, with `gpt-4o-2024-11-20` as the evaluation model.

| Paper | Source | Score |
|---|---|---|
| PACE: marrying generalization in PArameter-efficient fine-tuning with Consistency rEgularization | Oral | 4 |
| The Road Less Scheduled | Oral | 4 |
| G-Retriever: Retrieval-Augmented Generation for Textual Graph Understanding and Question Answering | Poster | 4 |
| Binarized Diffusion Model for Image Super-Resolution | Poster | 4 |
| Learning to Predict Structural Vibrations | Poster | 4 |
| Attack-Aware Noise Calibration for Differential Privacy | Poster | 4 |
| Make Your LLM Fully Utilize the Context | Poster | 3.9 |
| Smoothed Energy Guidance: Guiding Diffusion Models with Reduced Energy Curvature of Attention | Poster | 3.9 |
| Sm: enhanced localization in Multiple Instance Learning for medical imaging classification | Poster | 3.9 |
| AutoTimes: Autoregressive Time Series Forecasters via Large Language Models | Poster | 3.9 |
| End-to-End Ontology Learning with Large Language Models | Poster | 3.8 |
| Scaling transformer neural networks for skillful and reliable medium-range weather forecasting | Poster | 3.8 |
| Autoregressive Image Generation without Vector Quantization | Oral | 3.7 |
| Adaptive Randomized Smoothing: Certified Adversarial Robustness for Multi-Step Defences | Oral | 3.7 |
| Generalizable Person Re-identification via Balancing Alignment and Uniformity | Poster | 3.7 |
| Universal Neural Functionals | Poster | 3.7 |
| Are Self-Attentions Effective for Time Series Forecasting? | Poster | 3.7 |
| xMIL: Insightful Explanations for Multiple Instance Learning in Histopathology | Poster | 3.7 |
| Leveraging Environment Interaction for Automated PDDL Translation and Planning with Large Language Models | Poster | 3.7 |
| Task-Agnostic Machine Learning-Assisted Inference | Poster | 3.7 |
| Make Continual Learning Stronger via C-Flat | Poster | 3.7 |
| DARG: Dynamic Evaluation of Large Language Models via Adaptive Reasoning Graph | Poster | 3.7 |
| AsyncDiff: Parallelizing Diffusion Models by Asynchronous Denoising | Poster | 3.7 |
| You Only Look Around: Learning Illumination Invariant Feature for Low-light Object Detection | Poster | 3.6 |
| MutaPLM: Protein Language Modeling for Mutation Explanation and Engineering | Poster | 3.6 |
| Advancing Training Efficiency of Deep Spiking Neural Networks through Rate-based Backpropagation | Poster | 3.6 |
| Improved off-policy training of diffusion samplers | Poster | 3.6 |
| Navigating the Effect of Parametrization for Dimensionality Reduction | Poster | 3.6 |
| Long-Range Feedback Spiking Network Captures Dynamic and Static Representations of the Visual Cortex under Movie Stimuli | Poster | 3.6 |
| InfLLM: Training-Free Long-Context Extrapolation for LLMs with an Efficient Context Memory | Poster | 3.6 |

Table 23: List of papers used in human evaluation. We evaluate the official repository of each paper, released by the authors, using the model-based reference-free setting with `gpt-4o-2024-11-20` as the evaluation model.

| RepoName | Paper | Score |
|---|---|---|
| VideoICL | VideoICL: Confidence-based Iterative In-context Learning for Out-of-Distribution Video Understanding | 2.6 |
| MuDI | Identity Decoupling for Multi-Subject Personalization of Text-to-Image Models | 3.3 |
| KALMV | Knowledge-Augmented Language Model Verification | 3.3 |
| sea-attention | SEA: Sparse Linear Attention with Estimated Attention Mask | 2.7 |
| HarmAug | HarmAug: Effective Data Augmentation for Knowledge Distillation of Safety Guard Models | 3.0 |
| GruM | Graph Generation with Diffusion Mixture | 3.7 |
| Adaptive-RAG | Adaptive-RAG: Learning to Adapt Retrieval-Augmented Large Language Models through Question Complexity | 2.7 |
| SoT | Sketch-of-Thought: Efficient LLM Reasoning with Adaptive Cognitive-Inspired Sketching | 4.0 |
| Mol-LLaMA | Mol-LLaMA: Towards General Understanding of Molecules in Large Molecular Language Model | 3.5 |
| judge_code_efficiency | Rethinking Code Refinement: Learning to Judge Code Efficiency | 3.1 |
| KARD | Knowledge-Augmented Reasoning Distillation for Small Language Models in Knowledge-Intensive Tasks | 3.2 |
| COINCIDE_code | Concept-skill Transferability-based Data Selection for Large Vision-Language Models | 3.0 |
| Janus | Aligning to thousands of preferences via system message generalization | 3.5 |
| N/A | Silent Branding Attack: Trigger-free Data Poisoning Attack on Text-to-Image Diffusion Models | N/A |
| VideoRAG | VideoRAG: Retrieval-Augmented Generation over Video Corpus | 3.0 |
| RADA | Retrieval-augmented data augmentation for low-resource domain tasks | 3.0 |
| STELLA_code | STELLA: Continual Audio-Video Pre-training with Spatio-Temporal Localized Alignment | 3.3 |
| prometheus-vision | Prometheus-vision: Vision-language model as a judge for fine-grained evaluation | 3.1 |
| CoLoR | Efficient Long Context Language Model Retrieval with Compression | 3.0 |
| Volcano | Volcano: Mitigating Multimodal Hallucination through Self-Feedback Guided Revision | 3.2 |
| N/A | T1: Tool-integrated Self-verification for Test-time Compute Scaling in Small Language Models | N/A |

Table 24: List of papers used in executability analysis.

| Repo Name | Paper |
|---|---|
| CoLoR | Efficient Long Context Language Model Retrieval with Compression |
| cognitive-behaviors | Cognitive Behaviors that Enable Self-Improving Reasoners, or, Four Habits of Highly Effective STaRs |
| RADA | Retrieval-Augmented Data Augmentation for Low-Resource Domain Tasks |
| Self-Instruct | Self-Instruct: Aligning Language Models with Self-Generated Instructions |
| G-EVAL | G-Eval: NLG Evaluation using GPT-4 with Better Human Alignment |

Name:
Paper:
Github:

[General]

**1**. If someone wants to reproduce the methods and experiments in your paper, which components would they need to implement? Please break down into the following sections: **(1) data processing, (2) method (e.g., model training or main pipeline), and (3) evaluation.**

*For example, in* Self-Instruct *(TLDR; Self-Instruct, a framework for improving the instruction-following capabilities of language models by bootstrapping off their own generations.):*

Data Processing

- N/A

Method (e.g., Model training or Main pipeline)

1. Instruction Generation
2. Classification Task Identification
3. Instance Generation
4. Filtering

Evaluation

- Training the model using the generated synthetic data via our methods
- Evaluating the trained model

| Your Answer |
| --- |
| Data Processing |
| Method (e.g., Model training or Main pipeline) |
| Evaluation |

Figure 41: Human Evaluation Guideline (1/3)

[Comparison]

**2.** Given a set of repositories, which one is the most helpful for reproducibility—that is, which one best re-implements the methods and experiments as intended by the paper?

Please review the provided repositories (Group 1: repo1–repo4, Group 2: repo5–repo7, Group 3: repo8–repo10) and rank them based on how well they are implemented.

It is worth noting that the same repository may appear more than once between repo1 and repo10; this is not an error.

(Optional things: Feel free to leave a comment explaining why you ranked them that way)

[Group1: repo1-repo4]

| | |
|---|---|
| 1st | |
| 2nd | |
| 3rd | |
| 4th | |

[Group2: repo5-repo7]

| | |
|---|---|
| 1st | |
| 2nd | |
| 3rd | |

[Group3: repo8-repo10]

| | |
|---|---|
| 1st | |
| 2nd | |
| 3rd | |

Among the top-ranked repositories in each group, which one do you think is the best? If the repositories are the same, you can select any of them. Please briefly explain your reason.

[All: repo1-repo10]

| | |
|---|---|
| 1st | |
| Reason | |

Figure 42: Human Evaluation Guideline (2/3)

[Detailed Analysis about the 1st Repository]

**3.** Do you think the first-ranked repository you chose would make it easier to reproduce the paper's methods and experiments than starting from scratch?

| Yes | |
|---|---|
| No | |

If you selected 'No', please briefly explain why. Otherwise, you may leave this blank.

| Reason for No | |
|---|---|

**4.** Based on the key components you mentioned in question 1, how well does the **"repo10"** repository support them?
 Please check one of the following for each component:
 **(o = fully implemented, △ = partially implemented, x = not implemented)**
If you select △ or ×, please briefly explain your reason.

*Example: Self-Instruct (TLDR; Self-Instruct, a framework for improving the instruction-following capabilities of language models by bootstrapping off their own generations.)*

Data Processing

- N/A

Method (e.g., Model training or Main pipeline)

1. Instruction Generation (o)
2. Classification Task Identification (o)
3. Instance Generation (△) : *They don't implement output-first and input-first separately.*
4. Filtering (△) : *They only implemented it using the ROUGE-L-based filter, not with the exact same input-output pairs.*

Evaluation

- Training the model using the generated synthetic data via our methods (o)
- Evaluating the trained model (x): *They only provided the training code.*

| Your Answer |
|---|
| Data Processing |
| Method (e.g., Model training or Main pipeline) |
| Evaluation |

Figure 43: Human Evaluation Guideline (3/3)

