# OpenReview forum: "Paper2Code: Automating Code Generation from Scientific Papers in Machine Learning"
_ICLR.cc/2026/Conference — ICLR 2026 Poster_

### Official Review · Reviewer_QcnK · 2025-10-27

**Soundness:** 3
**Presentation:** 4
**Contribution:** 4
**Rating:** 8
**Confidence:** 4

**Summary:**

The paper introduces PaperCoder, a multi-agent large language model (LLM) framework that automatically converts machine learning research papers into functional, repository-level code implementations. The system operates in three structured phases: planning, analysis, and coding. PaperCoder is evaluated on a benchmark of recent ML papers (ICLR, ICML, NeurIPS) and compared against baselines like ChatDev and MetaGPT, using both model-based and human evaluations. Results show that PaperCoder consistently generates more accurate and executable implementations, outperforming all baselines on both fidelity and usability metrics.

**Strengths:**

1. The paper tackles a highly impactful and underexplored task—automatic code generation from scientific papers—which directly addresses the reproducibility crisis in ML research. The framing of “Paper2Code” as a structured, multi-agent reasoning pipeline is both original and well-motivated.

2. The decomposition into Planning–Analysis–Coding is conceptually sound and empirically justified. Each stage mirrors human reasoning during software development, which gives the approach interpretability and modularity. The inclusion of architecture diagrams, execution order reasoning, and configuration synthesis adds practical depth.

3. The paper conducts extensive quantitative and qualitative evaluations, showing the effectiveness.

**Weaknesses:**

1. While the use of LLMs as judges is increasingly common, it introduces potential circularity and bias, especially since the same family of models may be used for both generation and evaluation. More independent human-based verification (beyond author feedback) would strengthen the claims.

2. Although executability is discussed, actual reproduction of original experimental results remains limited (only partial success in 4/5 cases). The paper does not sufficiently quantify semantic correctness or result fidelity, which are crucial for scientific reproducibility.

3. The ablation study (Table 6) shows some counterintuitive drops (e.g., architecture design hurting performance). While the authors provide a plausible rationale, a deeper qualitative analysis of failure cases would improve interpretability.

**Questions:**

none

---

> ### Author Response · Authors · 2025-11-21
> **Response to Reviewer QcnK (1/2)**
>
> We sincerely appreciate your constructive and helpful comments. We have made every effort to faithfully address all your comments and suggestions in the response below.
>
> ---
>
> > **Weaknesses 1:** While the use of LLMs as judges is increasingly common, it introduces potential circularity and bias, especially since the same family of models may be used for both generation and evaluation. More independent human-based verification (beyond author feedback) would strengthen the claims.
>
> Thank you for your thoughtful comment. We first point out that, while human evaluation might be the most reliable form of assessment, conducting full-scale human evaluation for all baselines and papers would be prohibitively expensive, as also noted in prior work [D.1]; therefore, we adopt the standard LLM-as-a-judge strategy for practical assessment. Also, as shown in Table 5 of the main paper, o3-mini-high correlates strongly with human judgments (0.78 in the reference-based setting and 0.73 in the reference-free setting), suggesting that the LLM-based evaluation could be a reliable proxy for human assessments.
>
> In addition to this, to further address the concern on the circularity and bias, specifically that an evaluator from the same model family might favor code generated by the same model family, we have additionally performed a cross-family evaluation using Gemini-Flash 2.5 on 30 randomly selected papers. The resulting Spearman correlation of 0.73 between Gemini and o3-mini-high indicates that the evaluation remains consistent across different model families, providing strong evidence that our results are not an artifact of model-family alignment.
>
> [D.1] Judging LLM-as-a-judge with MT-Bench and Chatbot Arena, 2023.
>
> ---
>
> > **Weaknesses 2:** Although executability is discussed, actual reproduction of original experimental results remains limited (only partial success in 4/5 cases). The paper does not sufficiently quantify semantic correctness or result fidelity, crucial for scientific reproducibility.
>
> We first would like to clarify that, while executability and reproducibility are important aspects of code quality, our primary focus is on generating implementations that faithfully capture the methodological pipeline described in the paper to help researchers implement the existing papers, instead of building a fully automated end-to-end system that reproduces every experimental result. Also, importantly, our evaluation framework already measures semantic correctness: whether the generated code aligns with the core algorithmic components described in the paper, via a fine-grained rubric in PaperBench (94 - 2,551 author-written nodes per paper) and component-level human assessment in Paper2CodeBench.
>
> Also, the challenge in reproducing exact numerical results (and the corresponding result fidelity) arises from the inherent variability of ML experiments, not from limitations in our PaperCoder framework, and is largely due to missing hyperparameters, undocumented details, environment mismatches, hardware variability, and dataset-related factors (such as differences in dataset versions, preprocessing steps, or restricted-access data). Nevertheless, to demonstrate reproducibility, we select five papers, manually debug them, and then show that the generated repositories can reproduce the paper implementations with only minor fixes (Tables 13 and 15 with case studies in Figures 7 and 8), which you also found and thoughtfully noted. However, in addition to this small-scale analysis, our LLM-assisted debugging experiment in Section 4.3 further demonstrates that automated repair can meaningfully improve executability at scale, suggesting that PaperCoder can be potentially extended into a more robust reproduction pipeline with minimal human effort.
>
> We sincerely appreciate your thoughtful feedback and the opportunity to further clarify!
>
> ---
>
> > **Weaknesses 3:** The ablation study (Table 6) shows some counterintuitive drops (e.g., architecture design hurting performance). While the authors provide a plausible rationale, a deeper qualitative analysis of failure cases would improve interpretability.
>
> Thank you for this thoughtful observation and constructive suggestion. The counterintuitive drop arises from how dependencies are communicated and used during generation. When only the Overall Plan is provided, the model generates all files in a single request; this naturally induces a consistent internal ordering, enabling the model to maintain cross-file dependencies and avoid conflicts. However, in the Architecture Design stage, the model knows which files must exist, but not when each should be produced, which results in files that are generated in an arbitrary order. In contrast, the subsequent Logic Design stage explicitly determines a dependency-aware file generation order, which resolves these issues. For examples of the samples, please refer to Figure D.2.

---

> ### Author Response · Authors · 2025-11-21
> **Response to Reviewer QcnK (2/2)**
>
> Figure D.2. Ablation case study. Without a dependency-aware generation order, the Architecture Design stage causes main.py to duplicate functions from dataset_loader.py. Adding the Logic Design phase resolves this issue by aligning the file-generation order.
>
> **+ Arch. Design**
>
> ```python
> # main.py
>
> # (... omitted ...)
>
> class DatasetLoader:
>
> # (... omitted ...)
>
>     def load_data(self) -> Tuple[DataLoader, DataLoader, DataLoader]:
>         """Returns DataLoaders for train, validation, and test splits."""
>         train_dataset = TimeSeriesDataset(
>             dataset_name=self.dataset_name, mode="train",
>             lookback=self.lookback, prediction_horizon=self.prediction_horizon,
>             num_samples=self.num_train
>         )
>         val_dataset = TimeSeriesDataset(
>             dataset_name=self.dataset_name, mode="val",
>             lookback=self.lookback, prediction_horizon=self.prediction_horizon,
>             num_samples=self.num_val
>         )
>         test_dataset = TimeSeriesDataset(
>             dataset_name=self.dataset_name, mode="test",
>             lookback=self.lookback, prediction_horizon=self.prediction_horizon,
>             num_samples=self.num_test
>         )
>         batch_size: int = self.config["training"].get("batch_size", 32)
>         train_loader = DataLoader(train_dataset, batch_size=batch_size, shuffle=True)
>         val_loader = DataLoader(val_dataset, batch_size=batch_size, shuffle=False)
>         test_loader = DataLoader(test_dataset, batch_size=batch_size, shuffle=False)
>         return train_loader, val_loader, test_loader
>
> # (... omitted ...)
>
> class Main:
>
> # (... omitted ...)
>
>     def run_experiment(self) -> None:
>         for dataset_name in dataset_names:
>             for run in range(self.num_runs):
>                 loader_instance = DatasetLoader(self.config, dataset_name)
>                 train_loader, val_loader, test_loader = loader_instance.load_data()
>
> ```
>
> ```python
> # dataset_loader.py
>
> # (... omitted ...)
>
> class DatasetLoader:
>
> # (... omitted ...)
>
>     def load_data(self) -> Tuple[DataLoader, DataLoader, DataLoader]:
>         batch_size: int = int(self.config.get("training", {}).get("batch_size", 32))
>         train_loader: DataLoader = DataLoader(train_dataset, batch_size=batch_size, shuffle=True, num_workers=0)
>         val_loader: DataLoader = DataLoader(val_dataset, batch_size=batch_size, shuffle=False, num_workers=0)
>         test_loader: DataLoader = DataLoader(test_dataset, batch_size=batch_size, shuffle=False, num_workers=0)
>         return train_loader, val_loader, test_loader
>
> ```
>
> ---
>
> **+ Logic Design**
>
> ```python
> # dataset_loader.py
>
> # (... omitted ...)
>
> class DatasetLoader:  """
>
> # (... omitted ...)
>
>     def load_data(self) -> Tuple[DataLoader, DataLoader, DataLoader]:
>         """
>         Loads the time series data, segments it, splits it into training, validation, and test sets,
>         and returns the corresponding DataLoader objects.
>
>         Returns:
>             Tuple[DataLoader, DataLoader, DataLoader]: (train_loader, val_loader, test_loader)
>         """
>         # Load CSV using pandas
>         df: pd.DataFrame = pd.read_csv(self.data_file)
>
> # (... omitted ...)
>
> ```
>
> ```python
> # main.py
>
> # (... omitted ...)
>
> class Main:
>
> # (... omitted ...)
>
>     def run_experiment(self) -> None:
>         num_runs: int = self.config.get("num_runs", 5)
>         seed_base: int = self.config.get("seed", 42)
>
>         for run in range(1, num_runs + 1):
>             dataset_loader = DatasetLoader(self.config)
>             data_loaders = dataset_loader.load_data()
>             train_loader = data_loaders["train"]
>             val_loader = data_loaders["val"]
>             test_loader = data_loaders["test"]
>
> ```

---

> > ### Comment · Reviewer_QcnK · 2025-11-24
> >
> > Thank you for your reply. I have decided to keep my positive rating.

---

> > > ### Author Response · Authors · 2025-11-27
> > >
> > > Thank you very much for taking the time to review our response and for maintaining your positive assessment. Please feel free to contact us if you have any further concerns or questions.

---

### Official Review · Reviewer_V6cT · 2025-10-28

**Soundness:** 3
**Presentation:** 3
**Contribution:** 3
**Rating:** 6
**Confidence:** 3

**Summary:**

This paper introduces PaperCoder, a multi-agent large language model (LLM) framework designed to transform machine learning papers into functional code repositories. The system emulates a human development pipeline, consisting of three key stages: planning, analysis, and generation. The authors evaluate PaperCoder using two benchmarks: Paper2CodeBench and the newly released PaperBench, employing both automated reference-free and reference-based metrics, along with expert human assessments. The results demonstrate significant improvements over baselines: 88% of the generated repositories receive the highest ratings, 92% of human judges find them useful, and the generated code typically requires only minor adjustments (with an average of 0.81% of lines modified). Ablation studies confirm the contribution of each stage to the overall performance. In summary, this paper addresses a valuable problem, and the proposed methodology and evaluation approach will provide meaningful support for future research.

**Strengths:**

1.	This paper introduces PaperCoder, a multi-agent large language model (LLM) framework designed to transform machine learning papers into functional code repositories through three key stages: planning, analysis, and generation. The results demonstrate significant improvements over baselines and ablation studies in Table 6 demonstrate the effectiveness of each proposed component.
2.	This paper constructs Paper2CodeBench by collecting recent machine learning papers with available code, filtering repositories for manageable size, and selecting high-quality papers through model-based evaluation. The benchmark provides reliable evaluation support for future research on paper-to-code generation.

**Weaknesses:**

1.	Reproducibility is a crucial metric for assessing code quality. However, this study conducts reproducibility analysis on only a limited subset, while the primary evaluation relies on the LLM’s judgment of code quality. It remains unclear whether the LLM’s scores reliably correlate with actual reproducibility, i.e., whether code receiving higher scores is truly more reproducible.
2.	The selected papers are drawn from ICLR, ICML, and NeurIPS 2024, along with their associated code repositories. It is important to consider potential data leakage risks, as these codebases may have been incorporated into the training of open-source models or closed-source APIs.
3.	In the LIMITATIONS section, the authors note that the current model only processes textual inputs. However, many methods and model architectures depend on the joint interpretation of diagrams and text. Consequently, relying solely on textual information may not faithfully reproduce the intended model structures. Exploring the use of PDFs as images input to a multimodal LLM for code generation represents an interesting direction.

**Questions:**

See the weaknesses

---

> ### Author Response · Authors · 2025-11-21
> **Response to Reviewer V6cT**
>
> We sincerely appreciate your constructive and helpful comments. We have made every effort to faithfully address all your comments and suggestions in the response below.
>
> ---
>
> > **Weaknesses 1:** Reproducibility is a crucial metric for assessing code quality. However, this study conducts reproducibility analysis on only a limited subset, while the primary evaluation relies on the LLM’s judgment of code quality. It remains unclear whether the LLM’s scores reliably correlate with actual reproducibility, i.e., whether code receiving higher scores is truly more reproducible.
>
> We thank you for raising this concern. We first would like to note that full end-to-end reproducibility testing for every paper is resource-intensive and, in fact, extremely difficult to achieve in practice due to heterogeneous environments, dependencies, dataset access constraints, and the substantial human effort required. Also, the primary focus of our work is on evaluating whether the generated code faithfully implements the methods described in the paper, instead of reproducing every reported result in the paper. Nevertheless, we perform reproducibility analyses from two complementary directions. First, PaperBench provides fine-grained, author-written rubrics (94 - 2,551 nodes per paper) and additionally includes the execution and result match (as well as the code development), on which our PaperCoder outperforms baselines (Table 8). Second, even within Paper2CodeBench, we verify executable reproducibility for 11 papers via the debugging module (Figure 6), and the correlation between LLM-based and human-based score changes increases to 0.82. Because these score changes directly reflect whether the generated repository can be successfully executed and repaired, this provides concrete evidence that the LLM’s scores track actual reproducibility rather than surface-level code quality.
>
> ---
>
> > **Weaknesses 2:** The selected papers are drawn from ICLR, ICML, and NeurIPS 2024, along with their associated code repositories. It is important to consider potential data leakage risks, as these codebases may have been incorporated into the training of open-source models or closed-source APIs.
>
> Thank you for raising this important point. To clarify, our evaluation targets papers from ICLR/ICML/NeurIPS 2024, because the primary models used in our experiments (OpenAI o3-mini and GPT-4o) have a knowledge cutoff of October 2023, meaning these 2024 paper/code repositories fall after the training window of models. While we cannot fully rule out all forms of indirect exposure, the temporal gap substantially reduces the likelihood of contamination, and thus, data contamination is unlikely to meaningfully affect the results.
>
> ---
>
> > **Weaknesses 3:** In the LIMITATIONS section, the authors note that the current model only processes textual inputs. However, many models and architectures depend on the joint interpretation of diagrams and text. Consequently, relying solely on textual information may not faithfully reproduce the intended model structures. Exploring the use of PDFs as images input to a multimodal LLM for code generation represents an interesting direction.
>
> Thank you for highlighting this direction of leveraging multimodal inputs for papers. We would like to note that our work focuses on establishing the initial feasibility of the paper to code paradigm, and therefore prioritizes the core question of whether LLMs can reliably transfer paper content into structured, executable code. In this context, we believe that building a fully multimodal pipeline (probably with the recent advances in multimodal OCR models, such as DeepSeek-OCR [C1]) is, while interesting, beyond the scope of our work.
>
> [C1] DeepSeek-OCR: Contexts Optical Compression, 2025.

---

> ### Comment · Reviewer_V6cT · 2025-11-26
> **Official Comment by Reviewer V6cT**
>
> Thank you for your response. While I agree that conducting reproducibility testing for every paper can be resource-intensive, I believe it is still essential to devote such effort to ensure that claims of reproducibility are fully justified. Additionally, DeepSeek-OCR is not, to my knowledge, among the most advanced multimodal OCR models. I would recommend considering the MonkeyOCR and MinerU series instead. I'll maintain my score.

---

> > ### Author Response · Authors · 2025-11-27
> >
> > We sincerely appreciate your time and effort in reviewing our response and for providing the follow-up comment.
> >
> > We would like to emphasize that, as discussed in our response to Weaknesses 1, reproducibility is not the main focus and claim of our work. Nevertheless, we believe that we have devoted substantial effort to evaluating it. Specifically, in Section 4.3 and Appendix B, we conduct manual reproducibility analyses (with examples in Figures 8-12 and summary statistics in Table 15, along with the detailed case studies in Table 18), perform LLM-assisted debugging experiments to improve reproducibility (Figure 6 and Table 14), and provide comprehensive results with automatic evaluations across both Paper2CodeBench and PaperBench (Table 8 and Figure 6). Additionally, we have revised the paper to ensure that any statements related to reproducibility are appropriately framed and do not overclaim.
> >
> > We also thank you very much for your suggestions about the advanced OCR models for the direction of future work. We have updated our discussion to reflect this, incorporating them into the broader discourse on multimodal extension of paper-to-code systems.
> >
> > Thank you again for your constructive feedback and for helping us strengthen our work.

---

### Official Review · Reviewer_aZeb · 2025-10-31

**Soundness:** 3
**Presentation:** 4
**Contribution:** 3
**Rating:** 8
**Confidence:** 3

**Summary:**

*Summary*

This paper addresses the critical reproducibility challenge in machine learning, where a significant fraction of published papers (e.g., ~80.5% in 2024) do not release their code. To combat this, the authors introduce **PaperCoder**, a multi-agent Large Language Model (LLM) framework designed to automatically generate functional code repositories directly from scientific papers

The core of PaperCoder is a structured, three-stage pipeline that mimics a human developer's workflow:
1.  **Planning:** A set of specialized agents first constructs a high-level roadmap, designs the system architecture (including class and sequence diagrams), identifies file dependencies and execution order, and generates configuration files (e.g., `config.yaml`)
2. **Analysis:** The framework then performs a fine-grained interpretation of the plan, defining implementation-specific details, inputs/outputs, and algorithmic constraints for each module
3.  **Generation:** Finally, the framework synthesizes the modular, dependency-aware code file-by-file, using all previously generated artifacts as context.

To evaluate their framework, the authors introduce a new benchmark, **Paper2CodeBench**, consisting of 90 recent papers from ICLR, ICML, and NeurIPS 2024. They conduct extensive evaluations using both model-based (reference-free and reference-based) and expert human evaluations, notably involving the authors of the original papers. The results demonstrate that PaperCoder substantially outperforms strong baselines, with 92% of human judges reporting the generated repositories as helpful.

**Strengths:**

This paper makes a highly significant contribution to the ICLR community.

1 **Significance of Problem:** The work directly addresses the critical and widely-felt reproducibility crisis in machine learning, where the lack of code is a major barrier to scientific progress.

2 **Novel and Sound Methodology:** The proposed **PaperCoder** framework is a strong contribution. Its multi-stage (Planning, Analysis, Coding) pipeline is a logical and novel approach that mimics how a human expert would translate a paper to code. The "Planning" stage, with its decomposition into an overall plan, architectural design, logic design, and configuration generation, is particularly strong and provides the necessary scaffolding that simpler one-shot methods lack.

3 **Exceptional Evaluation Protocol:** The evaluation is a major strength.
    1.  The authors introduce **Paper2CodeBench**, a new, relevant, and large-scale benchmark of 90 papers from top 2024 conferences.
    2.  They employ a robust LLM-as-judge evaluation with both reference-based (vs. author code) and reference-free (vs. paper text) settings.
    3.  Most impressively, they validate their findings using **human evaluations from the original paper authors**, which is the highest possible standard for this task. The strong results from this (e.g., 92% of authors found the generated code helpful) are highly persuasive.

4 **Clarity and Presentation:** The paper is extremely clear, well-written, and easy to follow. The figures and tables are informative and effectively communicate the framework's design and results.

**Weaknesses:**

1 **Overstated Executability Claims:** The abstract's claim of "functional code" and the conclusion's "strong executability" are misleading. These claims appear to rest on a *manual* debugging analysis of only **five** papers, which found an 0.81% fix rate. This is a very small and likely biased sample.

2 **Contradictory Automated Execution Results:** A more comprehensive analysis in Appendix B.4 (Table 12) shows that when the generated code was run automatically on the *full 90-paper benchmark*, only **4 out of 90** repositories executed successfully. This suggests the system is not "near-executable" out-of-the-box.

3 **Failure to Address Environment Setup:** The primary failure mode identified in Table 12 is environmental: "Missing Dependency" (23 papers), "ImportError" (14 papers), and "ModuleNotFoundError" (14 papers). Although the "Logic Design" phase plans for "Required packages", the generation stage clearly fails to implement this reliably. The framework is good at generating Python *syntax* but poor at generating the *runnable environment* (e.g., a `requirements.txt` file).

4 **Unresolved Ambiguity Bottleneck:** The paper's own fine-grained analysis (Figure 5) reveals that the framework's implementation coverage is lowest for **"Data Processing" (56%)**. The authors attribute this to papers being "under-specified", but handling this ambiguity *is* the core difficulty of the task. A case study (Table 15) confirms this, showing one failure was due to an "overly simplified" loss function description. The framework currently guesses (and fails) in the face of ambiguity.

5 **Potential Evaluation Bias:** In the human evaluation (Table 14), the top reasons for preferring PaperCoder were "Completeness" and "Clean Structure". Since the PaperCoder framework is *explicitly designed* to produce a complete, modular structure (via its Planning stage), the evaluation may be rewarding the framework's *scaffolding* more than its *algorithmic correctness* when compared to one-shot baselines that lack this scaffolding by default.

**Questions:**

1.  Given that only 4/90 automated runs succeeded and the primary failure was dependency-related, why isn't a "DevOps Agent" (to generate `requirements.txt`, `Dockerfile`, etc.) a core part of the "Planning" and "Generation" stages? This seems like a crucial missing piece for generating "functional repositories."
2.  The paper uses "LLM-assisted debugging" as an *evaluation* step in Section 4.3. This proved effective. Have the authors considered integrating this as a *closed-loop refinement step* within the PaperCoder framework itself? For example, the framework could generate the code, attempt to execute it in a sandbox, catch the `stderr`, and feed the error message back to the "Coding" agent to self-correct. Or, as an alternative, leverage unit tests for all the components.
3.  The "Data Processing" stage has the lowest coverage (56%), and a case study failed on an ambiguous loss function. How does the framework currently handle ambiguity? Would it be feasible for the "Analysis" agent to *stop and query the user* when it detects high ambiguity (e.g., an "overly simplified" description), rather than proceeding to generate a potentially incorrect implementation? Or, at least marking potential issues coming from unders-specification of the paper's description in the code?

---

> ### Author Response · Authors · 2025-11-21
> **Response to Reviewer aZeb (1/5)**
>
> We sincerely appreciate your constructive and helpful comments. We have made every effort to faithfully address all your comments and suggestions in the response below.
>
> ---
>
> > **Weaknesses 1:** Overstated Executability Claims: The abstract's claim of "functional code" and the conclusion's "strong executability" are misleading. These claims appear to rest on a manual debugging analysis of only five papers, which found an 0.81% fix rate. This is a very small and likely biased sample.
>
> Thank you for raising this point. We agree that the phrasing in the abstract and conclusion may overstate executability, and we will revise it accordingly. Our primary goal is to generate faithful implementations of the paper content, not to claim full end-to-end executability for every paper. Thus, the five-paper manual debugging study is intended as an illustrative analysis (not as the basis for our executability claims), and we acknowledge that, by itself, such a small sample could appear biased. However, executability is evaluated at a much broader scale in our work: in Paper2CodeBench, we conduct LLM-assisted debugging across the larger human-evaluation subset (Figure 6), and we observe consistent improvements that mirror the manual findings. Moreover, the 0.81% edit rate in the manual study is supported by the detailed error distributions in Figure B.1, showing that the failures are overwhelmingly simple (syntax/import issues) rather than structural or algorithmic.

---

> ### Author Response · Authors · 2025-11-21
> **Response to Reviewer aZeb (2/5)**
>
> Figure B.1. Contents of the modified files during the manual debugging steps.
>
> *RepoName: CoLoR*
>
> ```diff
> # config.yaml
> model:
>   base_model: "microsoft/Phi-3-mini-4k-instruct"
>   alternative_models:
> +   - "mistralai/Mistral-7B-Instruct-v0.3"
> +   - "meta-llama/Llama-3.2-3B-Instruct"
> ```
>
> ```diff
> # trainer.py
> - self.optimizer = AdamW(self.model.model.parameters(), lr=lr)
> + self.optimizer = AdamW(self.model.model.parameters(), lr=float(lr))
> ```
>
> ```diff
> # model.py
> - self.model = AutoModelForCausalLM.from_pretrained(base_model)
> + self.model = AutoModelForCausalLM.from_pretrained(
> +     base_model, trust_remote_code=True
> + )
> ```
>
> ---
>
> *RepoName: Cognitive-Behaviors*
>
> ```diff
> # config.yaml
> -    peak_learning_rate: 1e-5
> +   peak_learning_rate: 0.00001
>
> -    actor_learning_rate: 1e-6
> -    critic_learning_rate: 1e-5
> +   actor_learning_rate: 0.000001
> +   critic_learning_rate: 0.00001
>
> model:
>   base_models:
> -    - "Llama-3.2-3B"
> -    - "Qwen-2.5-3B"
> +    - "meta-llama/Llama-3.2-3B"
> +    - "Qwen/Qwen2.5-1.5B"
>
>   additional_models:
> -    - "Llama-3.1-70B"
> +    - "meta-llama/Llama-3.1-3B"
> ```
>
> ---
>
> *RepoName: RADA*
>
> ```diff
> # config.yaml
>  t5_base:
> -    model_name: "t5-base"
> +   model_name: "google-t5/t5-base"
>
>  llama2_7b:
> -    model_name: "Llama2-7B"
> +   model_name: "meta-llama/Llama-2-7b"
>
> - augmentation_model: "Llama2-7B-Chat"
> + augmentation_model: "meta-llama/Llama-2-7b-chat"
>
> retrieval:
> -  embedding_model: "distilbert-base-nli-stsb-mean-tokens"
> +  embedding_model: "sentence-transformers/distilbert-base-nli-mean-tokens"
>
> + seed_data_path: data/seed_data.json
> + external_data_path: data/external_data.csv
> ```
>
> ```diff
> # main.py
> -     generator: Generator = Generator(llm_model=augmentation_model)
> +     generator: Generator = Generator(
> +         llm_model=augmentation_model,
> +         generation_params={
> +                 "max_length": 2048,
> +                 "temperature": 0.7,
> +                 "top_k": 50,
> +                 "top_p": 0.95,
> +                 "num_return_sequences": 1
> +             }
> +         )
> ```
>
> ---
>
> *RepoName: Self-Instruct*
>
> ```diff
> # config.yaml
> -  engine: davinci
> + engine: gpt-4.1-nano
> ```
>
> ```diff
> # dataset_loader.py
> ​​- import openai
> + from openai import OpenAI
> + client = OpenAI(api_key=os.environ.get("OPENAI_API_KEY"))
>
> - response = openai.Completion.create(
> -     engine=self.engine,
> -     prompt=prompt,
> -     max_tokens=150,
> + response = client.chat.completions.create(
> +     model=self.engine,
> +     messages=[{"role": "user", "content": prompt}],
> +     max_completion_tokens=150,
>   )
> - raw_text = response.choices[0].text.strip()
> + raw_text = response.choices[0].message.content.strip()
>
> - answer = response.choices[0].text.strip().lower()
> + answer = response.choices[0].message.content.strip().lower()
>
> - generated_text = response.choices[0].text.strip()
> + generated_text = response.choices[0].message.content.strip()
> ```
>
> ---
>
> *RepoName: G-EVAL*
>
> ```diff
> # config.yaml
> -   name: text-davinci-003
> +   name: gpt-4.1-nano
>
> + summ_eval_name: data/summeval_1.csv
> + dialogue_name: data/topical_chat.csv
> + hallucination_name: data/qags.csv
> ```
>
> ```diff
> # config.py
> -                     "name": "gpt-4",
> +                     "name": "gpt-4o-mini",
> ```
>
> ```diff
> # llm_evaluator.py
> + from openai import OpenAI
> + client = OpenAI(api_key=os.environ.get("OPENAI_API_KEY"))
>
> - self.samples: int = int(params.get("samples", 20)) if self.model_name.lower() == "gpt-4" else 1
> + self.samples: int = int(params.get("samples", 20)) if "gpt-4" in self.model_name.lower() else 1
>
> - if self.model_name.lower() == "gpt-4":
> -     response = openai.ChatCompletion.create(
> + if "gpt-4" in self.model_name.lower():
> +     response = client.chat.completions.create(
>
> -     max_tokens=self.max_tokens,
> +     max_completion_tokens=self.max_tokens,
>
> -     text: str = response["choices"][0]["message"]["content"].strip()
> +     text = response.choices[0].message.content.strip()
>
> -     choice["message"]["content"].strip() for choice in response.get("choices", [])
> +     choice.message.content.strip() for choice in response.choices
>
> - if score_normalization and self.model_name.lower() == "gpt-4":
> + if score_normalization and "gpt-4" in self.model_name.lower():
> ```

---

> ### Author Response · Authors · 2025-11-21
> **Response to Reviewer aZeb (3/5)**
>
> ---
>
> > **Weaknesses 2:** Contradictory Automated Execution Results: A more comprehensive analysis in Appendix B.4 (Table 12) shows that when the generated code was run automatically on the full 90-paper benchmark, only 4 out of 90 repositories executed successfully. This suggests the system is not "near-executable" out of the box.
>
> Thank you for this meaningful observation. We agree that achieving “near-executable” code fully out-of-the-box across all papers is an extremely challenging goal, as automatic execution is highly sensitive to environment setup, dependency handling, and numerous paper-specific details that even human reimplementations often struggle with. Our automated run in Appendix B.4 highlights this difficulty directly: many failures stem from missing dependencies or environment issues rather than incorrect algorithmic logic. However, to examine whether these failures can be mitigated, we conduct an additional experiment incorporating an LLM-assisted debugging module on 11 repositories from the Paper2CodeBench human-evaluation subset. As shown in Figure 6, adding this simple debugging step yields a substantial 15.6-point improvement on the author-written rubric–based evaluation, demonstrating that executability improves markedly once even minimal automatic repair is introduced. This suggests that while full out-of-the-box execution remains a challenging problem, PaperCoder provides an initial foray that can be significantly enhanced by lightweight debugging, and points to a promising direction for future work.
>
> ---
>
> > **Weaknesses 3:** Failure to Address Environment Setup: The primary failure mode identified in Table 12 is environmental: "Missing Dependency" (23 papers), "ImportError" (14 papers), and "ModuleNotFoundError" (14 papers). Although the "Logic Design" phase plans for "Required packages", the generation stage clearly fails to implement this reliably. The framework is good at generating Python syntax but poor at generating the runnable environment (e.g., a requirements.txt file).
>
> Thank you for raising this point. Indeed, our current framework focuses on faithfully reconstructing the methodological pipeline described in the paper, rather than fully automating environment configuration. In practice, environment setup tends to be far easier and more reliable for users to adjust manually, whereas reconstructing the core algorithmic logic from natural language is substantially more challenging and central to our goal. For this reason, PaperCoder prioritizes method-level faithfulness over full automation of environment specification, which is highly paper-specific and often varies across systems. Nevertheless, to further handle this environmental issue, we have conducted an additional experiment on 30 papers (10 from each conference in Paper2CodeBench), augmenting PaperCoder with prompts explicitly aimed at inferring and repairing environment requirements. Then, in these cases, we do not observe dependency-related failures, suggesting that issues in Table 12 are not from the intrinsic weaknesses of the framework. We view incorporating a dedicated environment-construction or “DevOps” agent as an interesting extension for future work.
>
> ---
>
> > **Weaknesses 4:** Unresolved Ambiguity Bottleneck: The fine-grained analysis (Figure 5) reveals that the framework's implementation coverage is lowest for "Data Processing" (56%). The authors attribute this to papers being "under-specified", but handling this ambiguity is the core difficulty of the task. A case study (Table 15) confirms this, showing that one failure was due to an "overly simplified" loss function description. The framework currently guesses (and fails) in the face of ambiguity.
>
> We would like to emphasize that ambiguity in “Data Processing” (or other components) largely originates from the papers themselves, which often omit critical implementation details or describe them only at a high level. In such cases, the framework must rely on reasonable, context-driven guesses, since neither humans nor automated systems can infer implementation details that are simply not present in the paper. Since our work aims to examine the feasibility of the paper-to-code paradigm under these real-world constraints, we focus on faithfully reconstructing the method when the specification is sufficiently clear, while acknowledging that ambiguity handling is inherently limited by the source material. We agree that developing mechanisms to detect underspecification, flag uncertain components, or solicit clarification would be valuable, which we leave as future work.

---

> ### Author Response · Authors · 2025-11-21
> **Response to Reviewer aZeb (4/5)**
>
> ---
>
> > **Weaknesses 5:** Potential Evaluation Bias: In the human evaluation (Table 14), the top reasons for preferring PaperCoder were "Completeness" and "Clean Structure". Since the PaperCoder framework is explicitly designed to produce a complete, modular structure (via its Planning stage), the evaluation may reward the framework's scaffolding more than its algorithmic correctness when compared to one-shot baselines that lack this scaffolding.
>
> Thank you for pointing it out. While Table 14 shows that “Completeness” and “Clean Structure” are the most frequently cited reasons for preferring PaperCoder, these factors should not be interpreted as purely stylistic. In fact, (the most cited) Completeness is directly tied to whether all key algorithmic components required for reproduction are present and correctly implemented, which is an aspect that one-shot baselines often fail to satisfy because they omit modules rather than merely producing less-organized code. Moreover, as illustrated in Table B.2, human annotators explicitly evaluate not only structure but also the faithfulness and correctness of core algorithmic elements. Finally, PaperCoder’s strong performance on PaperBench (where multiple author-written rubric nodes per paper function as fine-grained, specification-driven correctness checks) further confirms that its advantages extend beyond scaffolding and include substantial gains in actual algorithmic correctness at the fine-grained levels. We hope this clarifies that the observed preference is not a bias toward structure alone but reflects genuinely more complete and faithful implementations!
>
>
> Table B.2. Human evaluation example for the question: “Among the top-ranked repositories in each group, which one do you think is the best? If the repositories are the same, you may select any of them. Please briefly explain your reasoning.”
> | RepoName | Reason (written by the evaluator, the first author of the paper) |
> |-----------|----------------------------------------------------------------|
> | **Janus** | The code includes everything needed to implement the paper. The code is clean and easy to understand (not overloaded with comments). It’s not overly packaged, and since config files are provided, it’s convenient for running various experiments. |
> | **VideoRAG** | Each component—data processing, retrieval, frame selection, generation, and evaluation—is clearly separated into its own module, making the system easy to maintain and extend. Moreover, the most critical modules are fully implemented, covering the essential functionality required by the framework. |
> | **T1** | This repository successfully implements the following:  Tool-based filtering,  Calculate score of each generation after filtering using verifier model, Training code for distillation |

---

> ### Author Response · Authors · 2025-11-21
> **Response to Reviewer aZeb (5/5)**
>
> ---
>
> > **Questions 1:** Given that only 4/90 automated runs succeeded and the primary failure was dependency-related, why isn't a "DevOps Agent" (to generate requirements.txt, Dockerfile, etc.)? This seems like a missing piece for generating "functional repositories."
>
> Thank you for the thoughtful suggestion. As noted in our response to Weaknesses 3, we focus primarily on reconstructing the methodological pipeline described in the paper rather than fully automating environment configuration. Also, while executability is valuable, generating a complete and correct environment specification (e.g., requirements.txt, Dockerfile) is typically easier for users to adjust compared to reconstructing the algorithmic logic itself. Moreover, in fact, our aforementioned experiment, which augments PaperCoder with prompts that repair environment settings in the “reproduce.sh” generation, can serve as a lightweight prototype and eliminates dependency-related failures. We view developing a full-fledged environment-construction agent as a promising direction for future work.
>
> ---
>
> > **Questions 2:** The paper uses "LLM-assisted debugging" as an evaluation step in Section 4.3. This proved effective. Have the authors considered integrating this as a closed-loop refinement step within the PaperCoder framework itself?
>
> This is an excellent point. As you noted, we explore a form of LLM-assisted closed-loop debugging in Section 4.3: the framework executes the generated repository inside a Docker sandbox, collects the resulting stderr logs, and feeds both the error messages and the relevant code back into the LLM for targeted repair. This simple feedback loop proves effective, improving the author-written rubric-based score by 15.6 points (Figure 6).
>
> ---
>
> > **Questions 3:** The "Data Processing" stage has the lowest coverage (56%), and a case study failed on an ambiguous loss function. How does the framework currently handle ambiguity? Would it be feasible for the "Analysis" agent to stop and query the user when it detects high ambiguity (e.g., an "overly simplified" description), rather than proceeding to generate a potentially incorrect implementation? Or, at least marking potential issues coming from unders-specification of the paper's description in the code?
>
> Thank you for this insightful question. Our current framework is designed to operate fully autonomously, without requiring human interaction during intermediate stages, and thus, when facing under-specified parts of a paper, PaperCoder follows standard practices, e.g., inferring reasonable defaults, relying on common development conventions, or generating editable configuration files (such as config.yaml) that allow users to easily adjust ambiguous components. Notably, because our work aims to assess the feasibility of the paper-to-code paradigm itself, resolving ambiguity (an issue that could stem from omissions or high-level descriptions in the original papers) is beyond our present scope. However, we believe that developing mechanisms to detect, surface, or actively query about ambiguous components opens up many promising directions and an exciting avenue for future work.

---

### Official Review · Reviewer_FaxH · 2025-11-01

**Soundness:** 1
**Presentation:** 1
**Contribution:** 2
**Rating:** 0
**Confidence:** 4

**Summary:**

The paper presents paper2code, an ambitious system for translating scientific ML papers into executable code. The pipeline comprises three stages—planning, analysis, and coding—aimed at extracting algorithmic intent and producing runnable implementations. Evaluations on the authors’ selected metrics indicate promising effectiveness of the approach.

**Strengths:**

1. Framing end-to-end paper to code generation is both challenging and potentially transformative for reproducibility and rapid prototyping.
2. The decomposition into planning, analysis, and coding is reasonable and provides a clear, modular workflow for bridging natural language descriptions and implementation.

**Weaknesses:**

1. It is not evident how the paper defines “correct” generated code (e.g., functional equivalence, numerical agreement within tolerance, adherence to algorithmic specifications). Please clarify ground truth, acceptance tests, and evaluation thresholds.
2. The claim that fixes average 0.81% of total lines may understate practical burden: small line edits can involve significant debugging time. Consider reporting developer time, number of repair iterations, error categories (syntax/import/runtime/logic), and success rates after each repair round.
3. Correct coding implementations can vary (e.g., refactoring choices, hyperparameters, library calls). Explain how the evaluation accommodates non-uniqueness—e.g., via canonical reference implementations, specification-driven unit tests, layout-/refactor-invariant metrics, or behavior-based measures on hidden test sets.

**Questions:**

1, How is correctness evaluated when multiple valid implementations exist, and what does the reported 0.81% edit rate imply for developer time and repair iterations?
2. How robust is the system across domains/frameworks/formats, how are licensing/leakage and security risks handled, and where does human review add the most value?

---

> ### Author Response · Authors · 2025-11-21
> **Response to Reviewer FaxH (1/4)**
>
> We sincerely appreciate your constructive and helpful comments. We have made every effort to faithfully address all your comments and suggestions in the response below.
>
> ---
>
> > **Weaknesses 1:** It is not evident how the paper defines “correct” generated code (e.g., functional equivalence, numerical agreement within tolerance, adherence to algorithmic specifications). Please clarify ground truth, acceptance tests, and evaluation thresholds.
>
> We clarify below how we define correctness and how our evaluation framework operationalizes it.
>
> First, in our work, the term correctness (or the correct generated code) refers to an implementation that faithfully reproduces the methods and results described in the paper. Under this definition, correctness is evaluated in terms of the faithfulness: whether the generated repository implements key components described in the original work and can, in principle, be executed to reproduce the reported results. Importantly, in this setting, the suggested criteria (e.g., functional equivalence) are neither required nor appropriate. This is because scientific papers permit multiple equally valid implementations, while still being fully faithful to the described method. In other words, enforcing strict numerical equivalence would wrongly penalize correct implementations that follow the paper but diverge from coding style.
>
> Given this definition, a single ground-truth implementation is inherently insufficient: when multiple faithful implementations exist, treating the repository released by authors as the only correct target would conflate one possible realization. To respect this non-uniqueness, our evaluation adopts two complementary strategies: (i) when an official repository is available, we use it only as a reference point, and a judge model compares the paper, the official repository, and the generated repository to assess whether key components have been correctly implemented; (ii) we also conduct a reference-free evaluation in which the judge observes only the paper and the generated code, assessing internal consistency with the methodology of the paper and whether all required modules are present. Finally, on the external PaperBench benchmark included, correctness is determined via a large set of rubric nodes that function as fine-grained acceptance tests (94 - 2,551 per paper), enabling implementation-level evaluation without assuming a single canonical ground truth.
>
> Lastly, regarding evaluation thresholds: our two evaluation settings follow the standard scoring protocols. In Paper2CodeBench, each component is rated on a 1 - 5 Likert scale to capture the degree of correctness and faithfulness. In contrast, PaperBench uses a binary decision (0/1) for every rubric node, where each node represents a minimal unit of required functionality. We adhere strictly to the thresholds without introducing additional heuristics.
>
> ---
>
> > **Weaknesses 2:** The claim that fixes an average of 0.81% of total lines may understate the practical burden: small line edits can involve significant debugging time. Consider reporting developer time, number of repair iterations, error categories (syntax/import/runtime/logic), and success rates after each repair round.
>
> We apologize for the confusion. While a small percentage of modified lines does not automatically imply low debugging effort, the observed fixes were overwhelmingly simple, which were minor syntax issues, missing imports, or small adjustments to variable names rather than logic- or architecture-level revisions, as shown in Figure A.1 below.
>
> To quantify this more concretely, as suggested, we have further estimated developer time by multiplying the number of modified lines by a difficulty factor (1 - 3):
> * 1 = simple syntax/typo fixes, variable renaming, comment adjustments
> * 2 = fixes requiring local reasoning (e.g., adjusting conditions or API usage)
> * 3 = nontrivial issues requiring deeper debugging (asynchronous or memory-related errors)
> The results of this estimation are then reported in the developer time column of Table A.2, which demonstrates that correcting the errors does not require a significant amount of time.
>
> In addition to human debugging, we have performed LLM-assisted repair on the previous fully evaluated papers. For each repair round, we record (i) the number of iterations required, and (ii) the error categories identified using GPT-5.1. These detailed results, included in Table A.3, show that this strategy resolves all the issues mostly within a small number of iterations and that the errors are primarily syntactic or import-related rather than logic-level.

---

> ### Author Response · Authors · 2025-11-21
> **Response to Reviewer FaxH (2/4)**
>
> Figure A.1. Contents of the modified files during the manual debugging steps.
>
> *RepoName: CoLoR*
>
> ```diff
> # config.yaml
> model:
>   base_model: "microsoft/Phi-3-mini-4k-instruct"
>   alternative_models:
> +   - "mistralai/Mistral-7B-Instruct-v0.3"
> +   - "meta-llama/Llama-3.2-3B-Instruct"
> ```
>
> ```diff
> # trainer.py
> - self.optimizer = AdamW(self.model.model.parameters(), lr=lr)
> + self.optimizer = AdamW(self.model.model.parameters(), lr=float(lr))
> ```
>
> ```diff
> # model.py
> - self.model = AutoModelForCausalLM.from_pretrained(base_model)
> + self.model = AutoModelForCausalLM.from_pretrained(
> +     base_model, trust_remote_code=True
> + )
> ```
>
> ---
>
> *RepoName: Cognitive-Behaviors*
>
> ```diff
> # config.yaml
> -    peak_learning_rate: 1e-5
> +   peak_learning_rate: 0.00001
>
> -    actor_learning_rate: 1e-6
> -    critic_learning_rate: 1e-5
> +   actor_learning_rate: 0.000001
> +   critic_learning_rate: 0.00001
>
> model:
>   base_models:
> -    - "Llama-3.2-3B"
> -    - "Qwen-2.5-3B"
> +    - "meta-llama/Llama-3.2-3B"
> +    - "Qwen/Qwen2.5-1.5B"
>
>   additional_models:
> -    - "Llama-3.1-70B"
> +    - "meta-llama/Llama-3.1-3B"
> ```
>
> ---
>
> *RepoName: RADA*
>
> ```diff
> # config.yaml
>  t5_base:
> -    model_name: "t5-base"
> +   model_name: "google-t5/t5-base"
>
>  llama2_7b:
> -    model_name: "Llama2-7B"
> +   model_name: "meta-llama/Llama-2-7b"
>
> - augmentation_model: "Llama2-7B-Chat"
> + augmentation_model: "meta-llama/Llama-2-7b-chat"
>
> retrieval:
> -  embedding_model: "distilbert-base-nli-stsb-mean-tokens"
> +  embedding_model: "sentence-transformers/distilbert-base-nli-mean-tokens"
>
> + seed_data_path: data/seed_data.json
> + external_data_path: data/external_data.csv
> ```
>
> ```diff
> # main.py
> -     generator: Generator = Generator(llm_model=augmentation_model)
> +     generator: Generator = Generator(
> +         llm_model=augmentation_model,
> +         generation_params={
> +                 "max_length": 2048,
> +                 "temperature": 0.7,
> +                 "top_k": 50,
> +                 "top_p": 0.95,
> +                 "num_return_sequences": 1
> +             }
> +         )
> ```
>
> ---
>
> *RepoName: Self-Instruct*
>
> ```diff
> # config.yaml
> -  engine: davinci
> + engine: gpt-4.1-nano
> ```
>
> ```diff
> # dataset_loader.py
> ​​- import openai
> + from openai import OpenAI
> + client = OpenAI(api_key=os.environ.get("OPENAI_API_KEY"))
>
> - response = openai.Completion.create(
> -     engine=self.engine,
> -     prompt=prompt,
> -     max_tokens=150,
> + response = client.chat.completions.create(
> +     model=self.engine,
> +     messages=[{"role": "user", "content": prompt}],
> +     max_completion_tokens=150,
>   )
> - raw_text = response.choices[0].text.strip()
> + raw_text = response.choices[0].message.content.strip()
>
> - answer = response.choices[0].text.strip().lower()
> + answer = response.choices[0].message.content.strip().lower()
>
> - generated_text = response.choices[0].text.strip()
> + generated_text = response.choices[0].message.content.strip()
> ```
>
> ---
>
> *RepoName: G-EVAL*
>
> ```diff
> # config.yaml
> -   name: text-davinci-003
> +   name: gpt-4.1-nano
>
> + summ_eval_name: data/summeval_1.csv
> + dialogue_name: data/topical_chat.csv
> + hallucination_name: data/qags.csv
> ```
>
> ```diff
> # config.py
> -                     "name": "gpt-4",
> +                     "name": "gpt-4o-mini",
> ```
>
> ```diff
> # llm_evaluator.py
> + from openai import OpenAI
> + client = OpenAI(api_key=os.environ.get("OPENAI_API_KEY"))
>
> - self.samples: int = int(params.get("samples", 20)) if self.model_name.lower() == "gpt-4" else 1
> + self.samples: int = int(params.get("samples", 20)) if "gpt-4" in self.model_name.lower() else 1
>
> - if self.model_name.lower() == "gpt-4":
> -     response = openai.ChatCompletion.create(
> + if "gpt-4" in self.model_name.lower():
> +     response = client.chat.completions.create(
>
> -     max_tokens=self.max_tokens,
> +     max_completion_tokens=self.max_tokens,
>
> -     text: str = response["choices"][0]["message"]["content"].strip()
> +     text = response.choices[0].message.content.strip()
>
> -     choice["message"]["content"].strip() for choice in response.get("choices", [])
> +     choice.message.content.strip() for choice in response.choices
>
> - if score_normalization and self.model_name.lower() == "gpt-4":
> + if score_normalization and "gpt-4" in self.model_name.lower():
> ```

---

> ### Author Response · Authors · 2025-11-21
> **Response to Reviewer FaxH (3/4)**
>
> Table A.2: Developer time comparison
> | Method              | Developer Time |
> |---------------------|-------------------------|
> | CoLoR               | 5                       |
> | Cognitive-Behaviors | 6                       |
> | RADA                | 27                      |
> | Self-Instruct       | 53                      |
> | G-EVAL              | 22                      |
>
> Table A.3: The number of repair iterations (with the LLM) and error categories
>
> | RepoName              | Iteration | Error Category                                                                 |
> |-----------------------|-----------|---------------------------------------------------------------------------------|
> |  **CoLoR**                | 1         | Configuration Update                                                            |
> |                  | 2         | Code Refactor / Robustness Enhancement                                          |
> |                  | 3         | Bug Fix                                                                        |
> |                  | 4         | Performance Adjustment / Behavior Control Update                               |
> | **Cognitive-Behaviors**   | 1         | Configuration Update                                                            |
> |    | 2         | Bug Fix / Type Compatibility Correction                                        |
> |    | 3         | Autograd Control / Memory Optimization                                         |
> |    | 4         | API Modernization / Correct Mask Handling / Behavior Tuning (Token Budget)     |
> |    | 5         | Bug Fix / Robustness Improvement                                               |
> |    | 6         | Representation Handling Feature + Refactor                                    |
> |    | 7         | Configuration Robustness / Type Validation                                     |
> |    | 8         | Training Efficiency Optimization / Memory Reduction / Autograd Hygiene         |
> |    | 9         | Memory & Performance Optimization / Behavioral Correction                      |
> |    | 10        | Configuration Robustness / Type Validation                                     |
> |    | 11        | Algorithmic Refactor / Performance & Memory Optimization / API Modernization / Robustness Improvement |
> | **RADA**                  | 1         | Configuration Update                                                            |
> |                   | 2         | API Modernization / Compatibility Update                                       |
> | **Self-Instruct**       | 1         | Configuration Update                                                            |
> |          | 2         | API Migration / Compatibility Update                                            |
> |          | 3         | API Modernization / SDK Refactor                                               |
> |          | 4         | API Rollback / Correct Endpoint Restoration                                    |
> |          | 5         | API Consistency Fix                                                            |
> |          | 6         | API Field Access Modernization                                                 |
> | **G-EVAL**                | 1         | API Modernization / SDK Migration                                               |
> |                | 2         | Debug Cleanup / Schema Update                                                  |

---

> ### Author Response · Authors · 2025-11-21
> **Response to Reviewer FaxH (4/4)**
>
> ---
>
> > **Weaknesses 3:** Correct coding implementations can vary (e.g., refactoring choices, hyperparameters, library calls). Explain how the evaluation accommodates non-uniqueness, e.g., via canonical reference implementations, specification-driven unit tests, layout-/refactor-invariant metrics, or behavior-based measures on hidden test sets.
>
> Thank you for raising this comment. As discussed in our response to Weaknesses 1, multiple correct implementations can exist for the same paper due to differences in refactoring choices, library usage, or hyperparameter selection. For this reason, our evaluation framework is explicitly designed to accommodate non-uniqueness and avoid penalizing valid variations with two complementary strategies: reference-based evaluation and reference-free evaluation, in which there are no canonical or line-by-line targets (please refer to our response to Weaknesses 1 with Section 4.1 for more details on them).
>
> In addition to this, to ensure that human evaluators also focus on the faithfulness of the generated code repository with its reproducibility (rather than specific implementation style), we first ask them to enumerate the minimal components required to reproduce the assigned papers according to data, method, and evaluation, and then ask them to assess the generated repository against these components using a 3-point likert scale (○ = fully implemented, △ = partially implemented, × = not implemented) with explanations when any component was partially or not implemented. This process ensures that evaluators judge correctness at the specification level, not based on stylistic implementation choices.
>
> Lastly, PaperBench (included in our experiments) provides a specific-driven evaluation: each paper is accompanied by a rubric annotated by humans, with 94 - 2,551 fine-grained rubric nodes (per paper) functioning as acceptance tests at the unit level for required components.
>
> ---
>
> > **Questions 1:** How is correctness evaluated when multiple valid implementations exist, and what does the reported 0.81% edit rate imply for developer time and repair iterations?
>
> Thank you for the thoughtful question. As this closely relates to Weaknesses 1 and 2, we briefly summarize our key points and refer the reviewer to those responses for full details.
>
> First, as discussed in our response to Weaknesses 1, our evaluation explicitly accommodates non-uniqueness by using both a reference-based setting (where the official repository is used only as an informative reference, not a canonical target) and a reference-free setting (where correctness is judged against the paper). This ensures that faithful implementations are not penalized for differing in style or structure.
>
> Also, as clarified in our response to Weaknesses 2, the 0.81% edits correspond almost entirely to minor syntactic or import-level fixes (Figure A.1), which translate into low debugging effort. The estimated developer time reported in Table A.2 reflects this, and our LLM-assisted repair experiments (Table A.3) show that most issues are resolved within a small number of iterations.
>
> ---
>
> > **Questions 2:** How robust is the system across domains/frameworks/formats, how are licensing/leakage and security risks handled, and where does human review add the most value?
>
> Thank you for the insightful question. First of all, as shown in Figure 9 of the paper, we already report the performance across paper categories and observe meaningful variation. Specifically, papers in Theory and Interpretability / Explainability tend to yield higher-quality repositories, whereas categories such as Application – RL / Control and Dataset exhibit lower scores. This suggests that while PaperCoder generalizes across a broad range of topics (substantially outperforming baselines on average), current LLMs still show weaknesses in domains that require extensive environment interaction, custom data pipelines, or specialized frameworks. We view this as an interesting direction for future work.
>
> Regarding licensing, leakage, and security risks, while we agree that these are something needed for deploying real-world paper-to-code systems, their considerations fall outside the scope of our work, which focuses on the evaluation of the paper-to-code paradigm itself.
>
> Finally, as summarized in Table 14 of the paper, human reviewers most frequently value Completeness and Clean Structure, followed by Faithfulness, Ease of Use, Code Quality, and Unique Strengths.

---

### Meta-Review · Area_Chair_D2M5 · 2026-01-01

**Summary:**

The major weaknesses that the reviewers mention are a lack of executability (FaxH, V6cT, QcnK) and instead use LLM to measure code performance (aZeb). The method performs poorly in cases when data/environment-related ambiguity is present (aZeb). Potential data leakage (V6cT).

**Reviewer Concerns:**

The potential data leakage problem, as well as he circular bias definition, have been well-addressed.

However, almost all the reviewers unanimously agree on the fundamental flaw that correctness or reproducibility should not be evaluated this way. The main reason why a code is expected along with the paper is its reproducibility. If we use the author’s definition of “correctness”, then the paper alone suffices in providing the idea. The difference between the paper and the codebase is the difference between idea and execution. With LLM as the judge, this work evaluates the idea and not the execution.

Similarly, the major issues even for humans while reproducing are setting the correct environment, installing libraries, and wrangling data. In most cases, even for humans, incrementally implementing a new ML module is relatively easier.

Since the reviews are very positive from 3/4 reviewers I'm inclined to accept the paper.

**Reviewer Scores:**

FaxH: I agree with the authors that a 0 rating is a bit harsh based on the weaknesses the reviewer has mentioned. FaxH would have likely increased the rating to 2. The feedback that has been given, especially the weaknesses, applies to the paper. I suspect the review appears LLM-generated, maybe because they put their review through an LLM to improve grammar.

The other three reviewer are likely to keep their ratings.

---

### Decision · Program_Chairs · 2026-01-26

Accept (Poster)